# Inhibition of fatty acid oxidation enables heart regeneration in adult mice

Xiang Li[1], Fan Wu[1], Stefan Günther[1], Mario Looso[1], Carsten Kuenne[1], Ting Zhang[1], Marion Wiesnet[1], Stephan Klatt[2], Sven Zukunft[2], Ingrid Fleming[2], Gernot Poschet[3], Astrid Wietelmann[1], Ann Atzberger[1], Michael Potente[4,5,7], Xuejun Yuan[1,6 ✉] & Thomas Braun[1,6 ✉]

Postnatal maturation of cardiomyocytes is characterized by a metabolic switch from glycolysis to fatty acid oxidation, chromatin reconfiguration and exit from the cell cycle, instating a barrier for adult heart regeneration[1,2]. Here, to explore whether metabolic reprogramming can overcome this barrier and enable heart regeneration, we abrogate fatty acid oxidation in cardiomyocytes by inactivation of *Cpt1b*. We find that disablement of fatty acid oxidation in cardiomyocytes improves resistance to hypoxia and stimulates cardiomyocyte proliferation, allowing heart regeneration after ischaemia–reperfusion injury. Metabolic studies reveal profound changes in energy metabolism and accumulation of α-ketoglutarate in *Cpt1b*-mutant cardiomyocytes, leading to activation of the α-ketoglutarate-dependent lysine demethylase KDM5 (ref. 3). Activated KDM5 demethylates broad H3K4me3 domains in genes that drive cardiomyocyte maturation, lowering their transcription levels and shifting cardiomyocytes into a less mature state, thereby promoting proliferation. We conclude that metabolic maturation shapes the epigenetic landscape of cardiomyocytes, creating a roadblock for further cell divisions. Reversal of this process allows repair of damaged hearts.

Shortly after birth, energy metabolism in cardiomyocytes (CMs) shifts from glycolytic to oxidative metabolism, resulting in major rearrangements in mitochondrial homoeostasis, cellular architecture and electrophysiological properties, among others[1]. Most CMs terminally withdraw from the cell cycle during the early postnatal period as part of the maturation program and undergo hypertrophic growth together with changes in cell–cell and cell–extracellular matrix interactions[2]. Postnatal maturation of CMs together with oxidative DNA damage, due to high metabolic activity, creates a natural barrier against CM cell division[4–6].

Metabolic and structural maturation of CMs are tightly intertwined. Reversal of either of the two processes may re-establish some proliferative capacity in CMs[7]. Enforcement of anaerobic metabolism by expression of pyruvate kinase isoenzyme 2 (*Pkm2*) mRNA, deletion of pyruvate dehydrogenase 4 (*Pdk4*) and inhibition of succinate dehydrogenase by malonate treatment promote CM cell proliferation[4,8–11]. However, the mechanisms linking metabolic rewiring with transcriptional and structural alterations, limiting proliferation of adult CMs, are not well understood.

Changes in the concentration of different metabolites, which serve as essential cofactors or substrates for various chromatin modifiers, potentially couple metabolic processes to the chromatin landscape and gene activity. α-ketoglutarate (αKG), a central intermediate of the Krebs cycle, is such a metabolite and is required for the activity

of several dioxygenases, including JmjC-domain-containing histone lysine demethylases (KDMs)[12]. αKG-dependent KDMs play pivotal roles in cardiac development, maturation of CMs and heart function by regulating levels of H3K4me3 and H3K9me3 histone modifications[13]. H3K4me3 is a classical marker of active promoters, and numerous cell identity genes in different tissues and organs, including the heart, are characterized by broad domains of H3K4me3. Such domains can extend for more than 40 kilobases and often correlate with high levels of gene expression and increased transcription elongation[14–17]. By contrast, narrow domains of H3K4me3 are usually found in promoters of typical housekeeping genes[14,18]. Although αKG is well known as a central metabolic fuel and for its function in numerous signalling pathways[19,20], information about the role of αKG in chromatin reconfiguration is scarce[21] and a potential role of αKG in promoting heart regeneration has not been investigated so far.

## Hyperplasia in *Cpt1b*-deficient hearts

Analysis of RNA-sequencing (RNA-seq) datasets revealed reduced expression levels of several key genes of glycolysis and cell cycle progression in the course of CM maturation during the first week after birth, whereas genes related to fatty acid oxidation (FAO) and the Krebs cycle were upregulated (Extended Data Fig. 1a). FAO-related genes that were upregulated included the muscle-specific isoform of carnitine

[1]Department of Cardiac Development and Remodeling, Max Planck Institute for Heart and Lung Research, Bad Nauheim, Germany. [2]Institute for Vascular Signaling, Centre for Molecular Medicine, Goethe-University, Frankfurt am Main, Germany. [3]Metabolomics Core Technology Platform, Centre for Organismal Studies (COS), Heidelberg University, Heidelberg, Germany. [4]Angiogenesis and Metabolism Laboratory, Max Planck Institute for Heart and Lung Research, Bad Nauheim, Germany. [5]Max Delbrück Center for Molecular Medicine, Helmholtz Association of German Research Centres, Berlin, Germany. [6]Instituto de Investigacion en Biomedicina de Buenos Aires (IBioBA) - CONICET - Partner Institute of the Max Planck Society, Buenos Aires, Argentina. [7]Present address: Berlin Institute of Health, Charité-Universitätsmedizin Berlin, Berlin, Germany. ✉e-mail: Xuejun.Yuan@mpi-bn.mpg.de; Thomas.Braun@mpi-bn.mpg.de

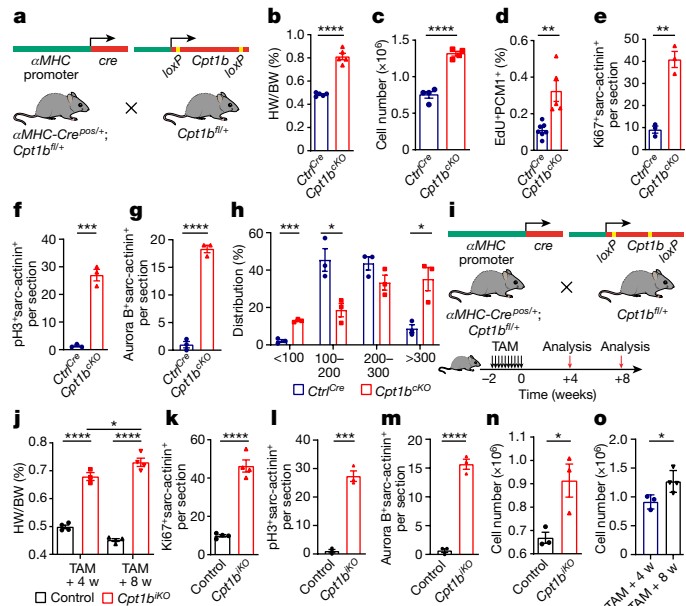

**Fig. 1 | Inactivation of *Cpt1b* induces hyperplastic and hypertrophic growth of CMs. a**, Generation of *Cpt1b^cKO* mice. **b**, HW/BW ratio of 10-week-old *Ctrl^Cre* and *Cpt1b^cKO* mice (*n* = 5 per genotype). **c**, Quantification of CMs in adult *Ctrl^Cre* and *Cpt1b^cKO* hearts (*n* = 4 per genotype). **d**, EdU incorporation in PCM1^+ cardiac nuclei from *Ctrl^Cre* hearts (*n* = 7) and *Cpt1b^cKO* hearts (*n* = 5) by FACS analysis. **e–g**, Quantification of Ki67^+ (**e**), pH3^+ (**f**), aurora B^+ (**g**) and sarcomeric (sarc)-actinin^+ CMs on heart sections from *Ctrl^Cre* and *Cpt1b^cKO* mice (*n* = 3 per genotype). **h**, Distribution of CM cross-section area (μm²) in *Ctrl^Cre* and *Cpt1b^cKO* mice (*n* = 3 each). **i**, Strategy to generate *Cpt1b^iKO* mice and experimental outline. **j**, HW/BW ratio of control and *Cpt1b^iKO* mice, 4 and 8 weeks (w) after TAM injection (control, 4 and 8 weeks after TAM, and *Cpt1b^iKO*, 8 weeks after TAM, *n* = 4 per group; *Cpt1b^iKO*, 4 weeks after TAM, *n* = 3). **k–m**, Quantification of Ki67^+ (**k**), pH3^+ (**l**), aurora B^+ (**m**) and sarc-actinin^+ CMs on heart sections from control and *Cpt1b^iKO* mice (*n* = 4 for **k**; *n* = 3 for **l**,**m**), 4 weeks after completion of TAM treatment. **n**, Quantification of CMs in control and *Cpt1b^iKO* hearts 4 weeks after TAM injection (*n* = 3 each). **o**, Quantification of CMs in *Cpt1b^iKO* hearts 4 weeks (*n* = 3) and 8 weeks (*n* = 4) after TAM injection. Error bars represent mean ± s.e.m. *n* numbers refer to individual mice. Two-tailed, unpaired Student *t*-tests were used for statistical analysis of data in **b–h**,**k–o**. One-way analysis of variance (ANOVA) with Tukey tests was used for correction of multiple comparisons in **j**. *$P < 0.05$, **$P < 0.01$, ***$P < 0.001$, ****$P < 0.0001$.

palmitoyltransferase *Cpt1b* but not the ubiquitously expressed *Cpt1a* isoform, which are required for mitochondrial uptake of fatty acids and subsequent FAO (Extended Data Fig. 1b). This observation is in line with previous studies reporting developmental changes in the use of cardiac energy substrates and a correlation between increased FAO and cell cycle withdrawal during CM maturation[22,23]. To explore a potential role of FAO in CM maturation and termination of proliferation, we inhibited the activity of CPT1 in CMs from neonatal mice at postnatal days 0–1 (P0–1) by treatment with etomoxir. Etomoxir treatment resulted in enhanced incorporation of 5-ethynyl-2′-deoxyuridine (EdU), increased numbers of Ki67- and pH3(Ser10)-positive cells, elevated protein levels of cyclin E1 and strong re-expression of *Nppa*, *Nppb* and *Acta1*, markers that are highly expressed at embryonic and fetal stages but are downregulated after birth[24] (Extended Data Fig. 1c–f).

To determine the role of FAO for regulation of CM proliferation and growth in vivo, we generated *αMHC-Cre^pos/+;Cpt1b^fl/fl* mice (hereafter referred to as *Cpt1b^cKO*), in which the *Cpt1b* gene is specifically inactivated in CMs at embryonic stages. Inactivation of the *Cpt1b* gene was efficient without a compensatory increase in the level of *Cpt1a* expression (Fig. 1a and Extended Data Fig. 1g,h). Body weight (BW), heart weight (HW), HW/BW ratios and morphology of *Cpt1b^cKO* hearts were

not changed compared to those of *αMHC-Cre^pos/+;Cpt1b^+/+* control mice (hereafter referred to as *Ctrl^Cre*) at P7 (Extended Data Fig. 1i,j). However, heart size, HW and HW/BW ratios but not BW increased in 10-week-old *Cpt1b^cKO* mice owing to concentric growth of the myocardium (Fig. 1b and Extended Data Fig. 1k–o). Notably, we observed a doubling of the absolute numbers of CMs in *Cpt1b^cKO* hearts, which was accompanied by markedly higher numbers of PCM1^+EdU^+ CMs and of Ki67^+, pH3^+ and aurora B^+ CMs in *Cpt1b*-mutant hearts at 10 weeks of age (Fig. 1c–g and Extended Data Fig. 2a–e). In addition, we noted a relatively modest increase of CM surface area in *Cpt1b^cKO* mice, suggesting combined hyperplastic and hypertrophic growth of the myocardium.

A more detailed morphometric evaluation revealed a marked increase of the CM population with the smallest cell size (<100 μm²), suggesting formation of new CMs in response to *Cpt1b* inactivation. The percentage of medium-sized CMs (100–300 μm²) decreased, whereas the number of larger CMs (>300 μm²) increased significantly, reflecting the overall increase of CM surface area (Fig. 1h). The relatively modest increase of CM surface area, in contrast to the massive increase in CM numbers in *Cpt1b^cKO* hearts, suggests that the increased heart size is mainly caused by hyperplastic growth. Of note, no signs of pathological cardiac hypertrophy such as myocardial fibrosis or cardiac dysfunction were evident in *Cpt1b^cKO* mice (Extended Data Fig. 2f–h). The data indicate that loss of *Cpt1b* stimulates proliferation of CMs, which is normally terminated in neonatal mouse hearts during the first week after birth.

## *Cpt1b*-deficient CMs are less mature

To examine whether inactivation of *Cpt1b* in fully differentiated adult CMs causes them to exit the post-mitotic state, we generated tamoxifen (TAM)-inducible *αMHC-MerCreMer^pos/+;Cpt1b^fl/fl* mice (hereafter referred to as *Cpt1b^iKO*; Fig. 1i and Extended Data Fig. 2i). As for *Cpt1b^cKO* mice, the HW and HW/BW ratios but not the BW of *Cpt1b^iKO* mice increased markedly compared to those of *αMHC-MerCreMer^pos/+;Cpt1b^+/+* control mice 4 and 8 weeks after TAM injections (Fig. 1j and Extended Data Fig. 2j). Likewise, we did not observe fibrosis or impaired cardiac function, although *Cpt1b^iKO* mice hearts showed a pronounced cardiomegaly and an increase of CM surface area (Extended Data Fig. 2k–o). Notably, the number of Ki67^+, pH3^+ and aurora B^+ CMs and the total number of CMs per heart were markedly elevated 4 weeks after TAM injection (Fig. 1k–n and Extended Data Fig. 2p–r). We also detected a further increase in the absolute number of CMs per heart and in HW at 8 weeks compared to 4 weeks after TAM administration, indicating that induction of CM proliferation by inactivation of *Cpt1b* is a lasting and continuous process (Fig. 1o and Extended Data Fig. 2j).

We next carried out RNA-seq analysis of CMs isolated from control, *Cpt1b^cKO* and *Cpt1b^iKO* mice and identified 1,513 and 1,432 differentially expressed genes (*P* value < 0.05) in *Cpt1b^cKO* and *Cpt1b^iKO* hearts, respectively. Gene Ontology (GO) term enrichment analysis recognized several upregulated genes associated with fatty acid and lipid metabolic processes in both mutant mouse lines, most likely reflecting an attempt to compensate for abrogation of FAO[4,5]. In both mutant mouse lines, we also detected strong upregulation of genes associated with cell cycle progression and reduced expression levels of genes involved in maturation and contraction, concomitant with upregulation of dedifferentiation markers (Extended Data Fig. 3a–c). Downregulated genes were mainly involved in cardiac muscle cell differentiation, heart development and sarcomere organization, suggesting a less mature state of *Cpt1b*-deficient CMs (Extended Data Fig. 3a–c). This notion was further confirmed by electron microscopy analysis and quantification of sarcomere density through immunofluorescence staining, uncovering reduced density of sarcomeres and irregular positioning of mitochondria, respectively (Extended Data Fig. 3d,e). The reduction of sarcomere density was particularly evident in the centre of CMs, close to the nuclei (Extended Data Fig. 3e). In addition, analysis by

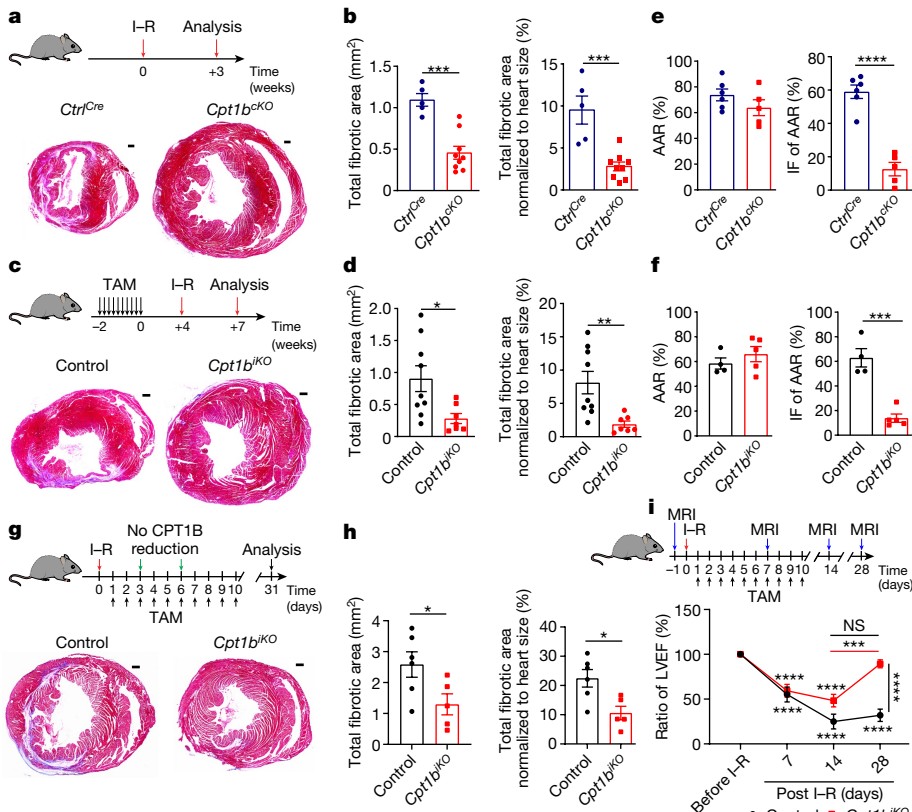

**Fig. 2 | *Cpt1b*-mediated abrogation of FAO protects from I–R damage and enables heart regeneration. a–d**, Trichrome staining (**a**,**c**) and quantification of fibrosis (scar area; **b**,**d**) on heart sections from *Ctrl^Cre* mice (*n* = 5) and *Cpt1b^cKO* mice (*n* = 9) (**a**,**b**), and control mice (*n* = 9) and *Cpt1b^iKO* mice (*n* = 7) (**c**,**d**), 3 weeks after I–R injury. I–R surgery was carried out using 7-week-old (**a**) or 14-week-old (**c**) mice, 4 weeks after completion of TAM treatment. Scale bars, 300 μm. **e**,**f**, AAR (left) and infarct area (IF) of AAR (right) in *Ctrl^Cre* mice (*n* = 6) and *Cpt1b^cKO* mice (*n* = 5) (**e**) and in control mice (*n* = 4) and *Cpt1b^iKO* mice (*n* = 5) (**f**) 24 h after I–R injury. **g**,**h**, Trichrome staining (**g**) and quantification (**h**) of fibrosis on heart sections from control mice (*n* = 6) and *Cpt1b^iKO* mice (*n* = 5), 31 days after I–R injury. The I–R injury was carried out 1 day before initiation of *Cpt1b* deletion. Scale bars, 300 μm. **i**, Magnetic resonance imaging-based assessment of heart function in control and *Cpt1b^iKO* mice before and 7, 14 and 28 days after I–R surgery (before and 7 days after I–R, *n* = 6 each; 14 days after I–R, *n* = 5 each; 28 days after I–R, control *n* = 4, *Cpt1b^iKO* *n* = 5). LVEF, left ventricle ejection fraction. Asterisks representing *P* values in **i** refer to differences between individual measurements and the measurement before I–R. Asterisks representing *P* values above lines refer to measurements connected by the lines. Error bars represent mean ± s.e.m. *n* numbers refer to individual mice. Two-tailed, unpaired Student *t*-tests were used for statistical analysis in **b**,**d**–**f**,**h**. Two-way ANOVA with Tukey tests was used for correction of multiple comparisons in **i**. NS, not significant; *$P < 0.05$; **$P < 0.01$; ***$P < 0.001$; ****$P < 0.0001$.

quantitative PCR (qPCR) with reverse transcription demonstrated reduced expression levels of mature CM-specific genes involved in Ca²⁺ signalling, glucose transport (*Slc2a4*) and lactate production (*Ldha*), but increased expression levels of dedifferentiation markers (*Nppa* and *Acta1*; Extended Data Fig. 3f). Integrative analysis of RNA-seq datasets from *Cpt1b^cKO* and *Cpt1b^iKO* CMs disclosed 172 and 299 genes that were jointly upregulated or downregulated in both mutant strains (Extended Data Fig. 3g). Genes involved in heart development and cell differentiation dominated in the group of downregulated genes (Extended Data Fig. 3h).

Of note, we observed activation of the anti-apoptotic HIF1A signalling pathway indicated by increased expression levels of *Hif1α* and its target gene *Bcl2*. *Egln3*, a suppressor of HIF1A, was downregulated, as were the pro-apoptotic genes *Bnip3* and *Bcl2l11*, which corresponds to the absence of TUNEL⁺ CMs in *Cpt1b^cKO* hearts (Extended Data Figs. 3c and 4a,b). Unexpectedly, we did not detect differences in the levels of reactive oxygen species between control and *Cpt1b*-knockout hearts, indicating that inhibition of FAO does not attenuate generation of reactive oxygen species (Extended Data Fig. 4c). However, the percentage of γH2A.X⁺ cardiac nuclei dropped markedly in *Cpt1b*-deficient hearts compared to control hearts, suggesting that inhibition of FAO reduces DNA damage irrespective of reactive oxygen species or stimulates DNA repair (Extended Data Fig. 4d). Taken together, the data indicate that inhibition of FAO initiates a cascade of events converting adult CMs into a more immature state that enables CM proliferation and to a lesser extent favours hypertrophy.

## FAO blockade enables heart regeneration

To investigate whether *Cpt1b* inactivation and the ensuing proliferation of CMs enables heart regeneration, we subjected *Cpt1b^cKO* and *Cpt1b^iKO* mice to ischaemia–reperfusion (I–R) injury, a model that closely mimics the situation in human patients receiving a stent for revascularization of an obstructed coronary artery. I–R-induced scars were virtually absent in *Cpt1b^cKO* and *Cpt1b^iKO* mice after 3 weeks compared to control animals, although the area at risk (AAR) was similar in both mutant hearts 24 h after I–R surgery (Fig. 2a–f, Extended Data Fig. 5a,b and Supplementary Information). The nearly complete prevention of scar formation was accompanied by the appearance of numerous small, round-shaped Ki67⁺ as well as aurora B⁺ CMs, specifically in the border zone of *Cpt1b*-deficient hearts, 72 h following I–R injury (Extended Data Fig. 5c), suggesting that the proliferative potential gained by abrogation of FAO allows preexisting CMs to re-enter the cell cycle and contribute to heart regeneration.

We detected a reduced infarct area within the AAR in both *Cpt1b^cKO* and *Cpt1b^iKO* hearts 24 h after I–R injury and better cardiac contraction

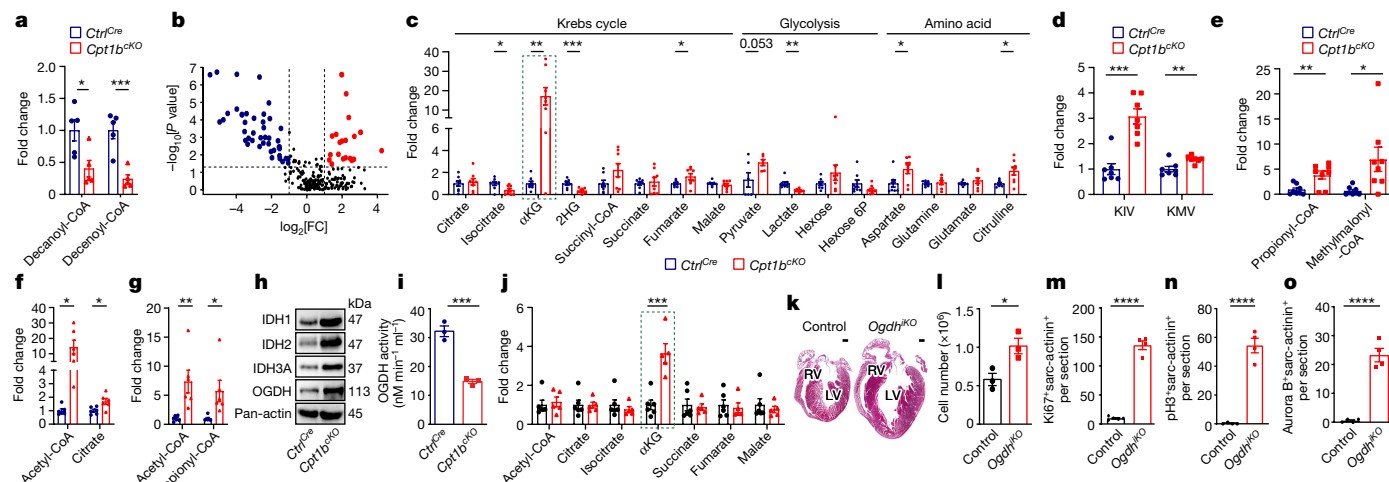

**Fig. 3 | Impeded mitochondrial import of fatty acid rewires the metabolism of CMs and boosts αKG levels. a**, Metabolic flux assay of *Ctrl^Cre* and *Cpt1b^cKO* hearts perfused with [^13C]palmitate (*n* = 5 each). **b**, Volcano plot showing different metabolites in *Ctrl^Cre* and *Cpt1b^cKO* hearts (*n* = 8 each). Symbol colors indicate increased (red), decreased (blue) or unchanged (black) metabolites in *Cpt1b^cKO* CMs. Dashed horizontal line indicates the threshold of changed metabolites with a *P* value of < 0.05; dashed vertical lines indicate a fold change of >2 or <0.5. **c**, Quantification of metabolites associated with the Krebs cycle and glycolysis, as well as amino acids, in *Cpt1b^cKO* hearts (pyruvate, *n* = 6 each; all other metabolites, *n* = 8 each). Dashed outline indicates the change in αKG. **d,e**, Quantification of BCAA catabolism-associated metabolites (*n* = 7 for **d**; *n* = 8 for **e**). KIV, α-ketoisovalerate; KMV, α-keto-β-methylvalerate. **f,g**, Metabolic flux assay of *Ctrl^Cre* and *Cpt1b^cKO* hearts perfused with [^13C]glucose (**f**) or [^13C] isoleucine (**g**) (*n* = 6 each). **h**, Western blot analysis of enzymes involved in αKG generation and catabolism (*n* = 3 each). Pan-actin was used as a loading control. For gel source data, see Supplementary Information. **i**, Enzymatic activity of OGDH in CMs from adult *Ctrl^Cre* and *Cpt1b^cKO* mice (*n* = 3 each). **j**, Comparison of Krebs cycle metabolites in CMs from control hearts (*n* = 6) and *Ogdh^iKO* hearts (*n* = 5). Dashed outline indicates the change in αKG **k**, Trichrome staining of heart sections from control and *Ogdh^iKO* mice, 3–4 weeks after termination of TAM treatment for *Ogdh* deletion. RV, right ventricle; LV, left ventricle. Scale bars, 500 μm. **l**, Quantification of CMs in adult control and *Ogdh^iKO* hearts (*n* = 3 each). **m–o**, Quantification of Ki67^+ (**m**), pH3^+ (**n**), aurora B^+ (**o**) and sarc-actinin^+ CMs on heart sections from control and *Ogdh^iKO* mice (**m**, control *n* = 5; *Ogdh^iKO* *n* = 4; **n** and **o**, *n* = 4 each). Error bar represents mean ± s.e.m. *n* numbers refer to individual mice. Two-tailed, unpaired Student *t*-tests were used for statistical analysis in **a,c–g,i,j,l–o**. \**P* < 0.05, \*\**P* < 0.01, \*\*\**P* < 0.001, \*\*\*\**P* < 0.0001.

in *Cpt1b^cKO* mice 48 h after I–R injury, suggesting that inhibition of FAO also confers protection against I–R damage (Fig. 2e,f and Extended Data Fig. 5d,e). In line with these findings, the rate of cell death was significantly reduced in *Cpt1b*-deficient CMs compared to control CMs, as measured by uptake of ethidium homodimer 1 after exposure to 1% O_2 (Extended Data Fig. 5f,g). To determine the contribution of CM proliferation to heart regeneration after I–R damage and to exclude enhanced protection as the main cause of the absence of scar formation, we initiated *Cpt1b* deletion 1 day after applying the I–R injury. Initiation of *Cpt1b* deletion by injection of TAM did not reduce CPT1B protein levels at 3 and 6 days after I–R injury, corresponding to 2 and 5 days after the first TAM injection. CPT1B protein levels were significantly reduced only 11 days after I–R injury (corresponding to 10 days after the first TAM injection) excluding the possibility that cardioprotection can occur owing to the absence CPT1B (Extended Data Fig. 5h). Histological analysis revealed a strong reduction of the fibrotic scar area in *Cpt1b^iKO* animals compared to control animals. Notably, we also observed a major recovery of cardiac functions 4 weeks after I–R surgery, reaching nearly the same level as before the injury (Fig. 2g–i, Supplementary Information and Supplementary Video), which unequivocally demonstrates that *Cpt1b* inactivation facilitates cardiac regeneration. We conclude that both enhanced CM proliferation and cardioprotection contribute to reduced scar formation after coronary occlusion in mice deficient for FAO.

## *Cpt1b* loss increases αKG levels in CMs

Next we investigated how blockage of FAO reprograms the metabolism of CMs. Metabolic flux and Seahorse assays demonstrated that long-chain fatty acid utilization is efficiently inhibited in *Cpt1b^cKO* CMs without evident compensatory usage of CPT1-independent medium- or short-chain fatty acids (Fig. 3a and Extended Data Fig. 6a). Furthermore,

targeted metabolome analysis revealed that levels of acyl-carnitines derived from medium-chain and long-chain fatty acids are strongly reduced in *Cpt1b*-deficient CMs, whereas intracellular levels of free carnitine were markedly elevated (Fig. 3b and Extended Data Fig. 6b,c). Unexpectedly, intracellular levels of acetyl-CoA and most metabolites of the Krebs cycle, such as citrate, succinate and malate, were not significantly altered, suggesting that CMs use different forms of energy production in the absence of FAO (Fig. 3c and Extended Data Fig. 6d). We assume that increased glucose oxidation compensates to some extent for the loss of FAO in *Cpt1b*-deficient CMs to generate acetyl-CoA, as the expression level of PDH1A but not that of its inhibitor PDK4 increases. Likewise, the expression levels of ACSS1 and ACSS2 that generate acetyl-CoA from acetate remained unchanged (Extended Data Fig. 6e).

Another potential source of energy in *Cpt1b*-depleted CMs is amino acids. We detected a major enrichment of metabolites involved in amino acid turnover, including ammonia recycling and urea cycle (Extended Data Fig. 6f). The levels of several amino acids whose intermediates (for example, acetyl-CoA, succinyl-CoA and fumarate) replenish the Krebs cycle were increased as well (Fig. 3c and Extended Data Fig. 6g,h). Likewise, we found a marked elevation of the levels of metabolites from branched chain amino acid (BCAA) catabolism in *Cpt1b*-deficient hearts, including α-ketoisovalerate and α-keto-β-methylvalerate. The levels of propionyl-CoA and methylmalonyl-CoA, which serve as precursors for succinyl-CoA, were also increased (Fig. 3d,e). Thus, we reason that enhanced glucose oxidation and BCAA catabolism in *Cpt1b*-deficient CMs efficiently compensate for impaired metabolic flux of fatty acid-derived acetyl-CoA into the Krebs cycle in *Cpt1b*-deficient CMs. This conclusion is also supported by the increased metabolic flux rate of ^13C-labelled glucose and isoleucine to metabolites of the Krebs cycle or BCAA catabolism, respectively. Additional evidence comes from Seahorse analysis of *Cpt1b*-deficient CMs, indicating more efficient

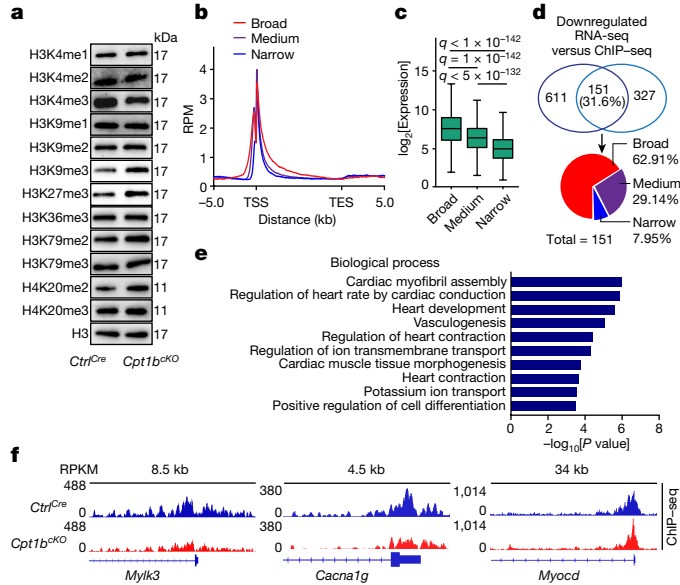

**Fig. 4 | Increased αKG levels induce H3K4me3 demethylation and decrease expression of CM maturity genes. a**, Western blot analysis of histone methylation modifications using FACS-isolated CM nuclei from $Ctrl^{Cre}$ and $Cpt1b^{cKO}$ mice ($n = 3$ mice each). Histone 3 (H3) was used as a loading control. For gel source data, see Supplementary Information. **b**, Coverage plots of H3K4me3 ChIP–seq signals within genes in three different groups categorized according to the breadth of H3K4me3 peaks in $Ctrl^{Cre}$ CMs. The 25% of genes with the broadest peaks were placed in the 'broad' group, the 25% of genes with the narrowest peaks were placed in the 'narrow' group, and the 'medium' group contains all other genes (25%–75%). RPM, reads per million mapped reads; TSS, transcription start site; TES, transcription end site; kb, kilobases. **c**, Box plots showing DESeq-normalized expression levels of genes with broad, medium and narrow H3K4me3 peaks in $Ctrl^{Cre}$ CMs. The box plot displays data from minimum to maximum, the median (centre line), 25th (bottom line) and 75th (top line) percentiles. One-way ANOVA analysis with multiple testing correction. The false discovery rate was controlled by using the two-stage step-up method of Benjamini, Krieger and Yekutieli. Broad, $n = 2,387$ genes; medium, $n = 4,775$ genes; narrow, $n = 2,387$ genes. **d**, Venn diagram showing the overlap between genes with reduced expression and genes with reduced H3K4me3 deposition in $Cpt1b^{cKO}$ compared to $Ctrl^{Cre}$ CMs. Distribution of overlapping peaks in the broad, medium and narrow groups is shown in the pie chart. **e**, Analysis of top GO terms from overlapping differentially expressed genes using the David tool ($n = 3$ mice each). **f**, Genome browser snapshots demonstrating reduced H3K4me3 deposition of representative genes associated with maturation of CMs in $Cpt1b^{cKO}$ CMs (normalized to mapped reads). RPKM, reads per kilobase per million mapped reads.

utilization of pyruvate, BCAAs and glucose compared to control CMs (Fig. 3f,g and Extended Data Fig. 6i,j).

Notably, we detected a marked, nearly 20-fold, increase of αKG in CMs after inactivation of *Cpt1b* (Fig. 3c). A direct conversion of glutamine and glutamate to αKG seems unlikely to account for the accumulation of αKG, as neither the concentration of glutamine nor that of glutamate declined (Fig. 3c). Instead, we observed significantly reduced isocitrate levels, along with an increase of the levels of IDH1, IDH2 and IDH3A, which catalyse conversion of isocitrate to αKG (Fig. 3c,h and Extended Data Fig. 6k). In addition, the enzymatic activity of OGDH, the key component of the αKG dehydrogenase complex converting αKG to succinyl-CoA was reduced in *Cpt1b*-deficient CMs, despite an increase in protein concentrations of OGDH complex components (Fig. 3h,i and Extended Data Fig. 6l). The reduction of the enzymatic activity of OGDH is most likely caused by the well-known inhibitory effects of intermediate metabolites from BCAA catabolism, the levels of which are increased in the *Cpt1b* mutants[25,26]. Inactivation of *Cpt1b* and αKG accumulation did not alter protein levels of other Krebs cycle

enzymes or mitochondrial DNA content (Extended Data Fig. 6m,n), indicating that FAO-independent functions of mitochondria were not compromised. Taken together, the results indicate that elevated synthesis and reduced metabolization synergize to accumulate αKG in CMs after abrogation of FAO.

We did not observe significant differences in the concentrations of diacylglycerols, triacylglycerols and very long-chain acyl-CoA in *Cpt1b*-deficient compared to control CMs (Extended Data Fig. 7a–d). We reason that the strong decline of *Lpl* expression in *Cpt1b*-deficient CMs prevents aberrant myocardial lipid accumulation and its sequelae (Extended Data Fig. 7e). To further substantiate the causality between αKG accumulation and CM reprogramming, we inactivated *Ogdh* in adult CMs using *αMHC-MerCreMer$^{pos/+}$Ogdh$^{fl/fl}$* (*Ogdh$^{iKO}$*) mice and *αMHC-MerCreMer$^{pos/+}$Ogdh$^{+/+}$* as controls, which—as expected—results in substantial accumulation of αKG (Fig. 3j and Extended Data Fig. 7f–h). Notably, CM-specific inactivation of *Ogdh* essentially recapitulated the phenotype observed in *Cpt1b*-knockout animals, including the massive increase in CM numbers, increased proliferation of CMs and reactivation of genes characteristic of immature CMs (Fig. 3k–o and Extended Data Fig. 7i–m). However, in contrast to *Cpt1b*-deficient mice, *Ogdh* mutants die between 5 and 8 weeks after gene inactivation, which makes them unsuitable for cardiac regeneration studies.

## Reduced H3K4me3 in CM identity genes

αKG is an essential cofactor for histone demethylases. Therefore, we wondered whether accumulation of αKG in *Cpt1b*-deficient CMs affects histone methylation. Analysis of different histone lysine methylation modifications using cardiac nuclei sorted by fluorescence-activated cell sorting (FACS) revealed only a reduction in the level of H3K4me3, whereas the levels of 11 other histone lysine methylation modifications were not reduced. By contrast, the levels of heterochromatin markers such as H3K9me3, H3K27me3 and H4K20me2 were markedly increased after inactivation of *Cpt1b*, probably owing to secondary effects associated with the shift of CMs to a more immature state (Fig. 4a and Extended Data Fig. 8a). Consistently, H3K4me3 levels are decreased in *Ogdh*-deficient CMs, which also exhibited increased αKG levels (Extended Data Fig. 8b). Expression levels of enzymes determining H3K4me3 levels, including methyltransferases (that is, *Kmt2* family, *Ash1l*, *Prdm9* and *Smyd* family) and αKG-dependent H3K4me3 demethylases (that is, *Kdm5a-d*, *Kdm2b* and *Riox1*), did not change after inactivation of *Cpt1b*. Neither did we detect a change in the intracellular levels of *S*-adenosyl methionine, the methyl-group donor for histone methylation. Therefore, we conclude that αKG boosts the activity of a H3K4me3-specific demethylase, eventually leading to reduction of H3K4me3 levels (Extended Data Fig. 8c,d).

Chromatin immunoprecipitation and sequencing (ChIP–seq) analysis of H3K4me3 using FACS-sorted CM nuclei to map gene regions with altered H3K4me3 identified 467 genes with increased and 478 genes with reduced H3K4me3 peaks in *Cpt1b*-deficient CMs compared to control CMs. Genes with increased H3K4me3 peaks were mainly associated with long-chain fatty acid transport and lipid and fatty acid metabolic processes, reflecting a potential feedback mechanism in response to abrogation of FAO. Notably, genes with reduced H3K4me3 peaks were mainly CM identity genes, involved in pathways regulating cardiac maturation and function (Extended Data Fig. 8e,f). As the breadth of H3K4me3 peaks is tightly linked to transcriptional strength of cell identity genes in various cell types and tissues, including the heart[14,27,28], we categorized genes into three groups according to the width of the H3K4me3 peaks. Genes showing the 25% broadest peaks exhibited the highest transcriptional activity and were preferentially associated with CM maturation and function (Fig. 4b,c and Extended Data Fig. 8g). Integrated analysis of RNA-seq and H3K4me3 ChIP–seq datasets indicated that 31.6% (151) of genes with reduced H3K4me3 peaks were transcriptionally repressed. A total of 95 genes (62.91%)

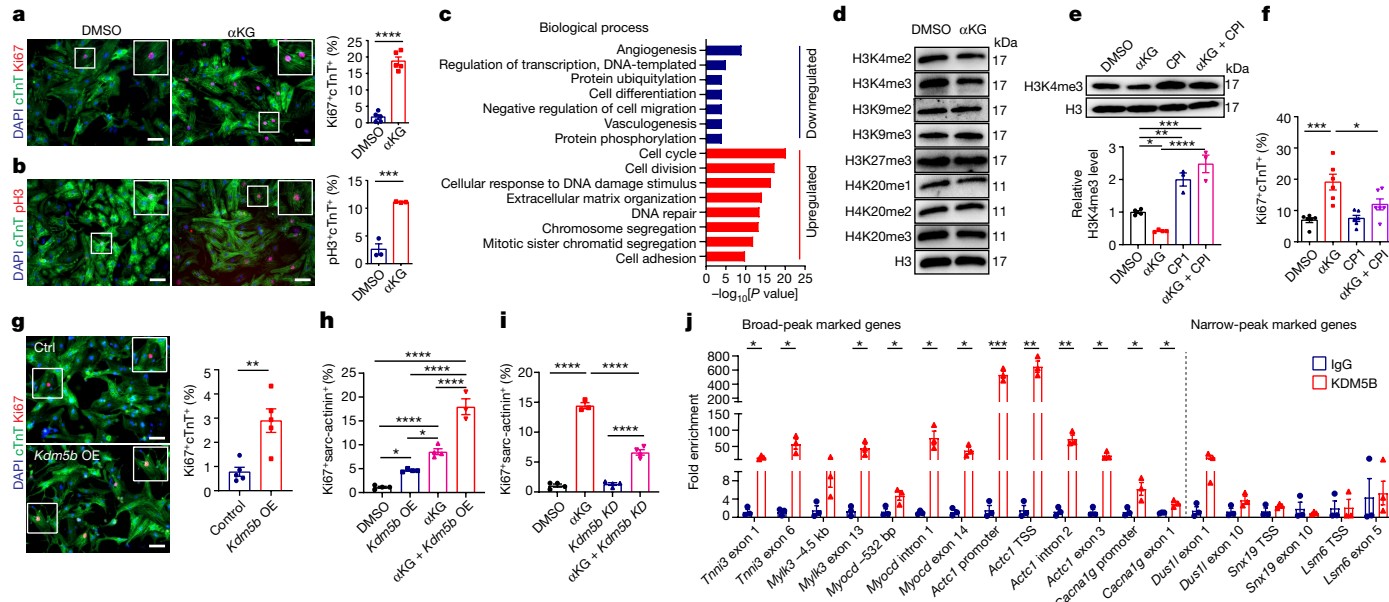

**Fig. 5 | Accumulation of αKG stimulates KDM5 activity, attenuates maturation and enhances proliferation of CMs. a,b,** Immunofluorescence images and quantification of Ki67 (*n* = 5 for **a**) and pH3(Ser10) (*n* = 3 for **b**) signals in cTnT⁺ neonatal CMs (P0–1) after 4-day culture in the presence of dimethylsulfoxide (DMSO) or (cell permeable) αKG. Top right corner, enlargement of area outlined in main image. Scale bars, 50 μm. **c,** GO-term enrichment of differentially expressed genes in DMSO- and αKG-treated neonatal CMs. Upregulated genes are in red, and downregulated genes are in blue. **d,** Western blot analysis of histone methylation modifications in P0–1 neonatal CMs treated with DMSO or αKG. H3 was used as a loading control. **e,** Top panels: western blot analysis of H3K4me3 in P0–1 CMs after 3-day culture in the presence of DMSO, αKG, CPI and αKG combined with CPI. H3 was used as a loading control. Lower panel: quantification of H3K4me3 levels (DMSO and αKG, *n* = 4 each; CPI and αKG + CPI, *n* = 3 each). **f,** Quantification of Ki67 signals in cTnT⁺ P0–1 CMs treated with DMSO, αKG, CPI and αKG combined with CPI (*n* = 6 each). **g,** Immunofluorescence micrographs and quantification of Ki67 signals in cTnT⁺ P0–1 CMs after lentiviral transduction of *Kdm5b* (*n* = 5 each). OE, overexpression. Top right corner, enlargement of area outlined in main image. Scale bars, 50 μm. **h,** Quantification of Ki67 signals in sarc-actinin⁺ neonatal CMs (P0–1), 4 days after lentiviral transduction of *Kdm5b* with and without αKG (DMSO, αKG, *Kdm5b* OE, *n* = 4; αKG + *Kdm5b* OE, *n* = 3). **i,** Quantification of Ki67 signals in sarc-actinin⁺ neonatal CMs (P0–1), 4 days after knockdown (KD) of *Kdm5b*, with and without αKG (DMSO, αKG + *Kdm5b* KD, *Kdm5b* KD, *n* = 4; αKG, *n* = 3). **j,** ChIP–qPCR of KDM5B in genes with broad and narrow H3K4me3 peaks in P0–1 CMs (*n* = 3). bp, base pairs. Error bars show mean ± s.e.m. Two-tailed, unpaired Student *t*-tests were used for statistical analysis in **a,b,g,i**. One-way ANOVA with Tukey tests was used for correction of multiple comparisons in **e,f,h,i**. *$P < 0.05$, **$P < 0.01$, ***$P < 0.001$, ****$P < 0.0001$. *n* numbers refer to independent experiments. For gel source data in **d,e**, see Supplementary Information.

in this group contained broad H3K4me3 peaks and 44 genes (29.14%) contained medium H3K4me3 peaks. By contrast, only a few genes with narrow H3K4me3 peaks (7.95%) showed reduced expression in *Cpt1b*-deficient CMs (Fig. 4d). Reduction of H3K4me3 deposition in genes with both reduced expression and reduced H3K4me3 peaks was found to be more pronounced inside gene bodies compared to transcription start site regions (Extended Data Fig. 8h). Strikingly, such genes were mostly associated with cardiac muscle contraction, CM differentiation and maturation, including *Mylk3*, *Cacna1g* and *Myocd* (Fig. 4e,f). We validated these findings by ChIP–qPCR, demonstrating that H3K4me3 enrichment was broadly diminished in both promoters and gene bodies of genes associated with CM maturation, whereas H3K9me3 and H3K27me3 levels at promoters remained unchanged (Extended Data Fig. 8i–k). Taking these findings together, we conclude that accumulation of αKG activates a H3K4me3-specific demethylase, which subsequently erases H3K4me3 within genes required for cardiac differentiation and maturation, thereby converting *Cpt1b*-deficient CMs into a more immature, proliferation-competent state.

## Activated KDM5 stimulates proliferation

To confirm that accumulation of αKG is a critical signal to prevent CM maturation and enable proliferation, we treated P0–1 neonatal CMs for 4 days with cell-permeable αKG. We observed a marked increase of Ki67⁺cTnT⁺ and pH3(Ser10)⁺cTnT⁺ CMs, consistent with RNA-seq data indicating enrichment of GO terms related to cell divisions in

αKG-treated cells (Fig. 5a–c). Furthermore, the level of expression of *Nppa*, *Nppb*, *Acta1* and *Myh7*, genes that are predominantly expressed during fetal and neonatal stages, was strongly elevated, whereas the level of expression of genes associated with CM maturation and differentiation, including *Tnni3*, *Mylk3* and *Myocd*, was reduced (Fig. 5c and Extended Data Fig. 9a). Treatment with αKG also diminished H3K4me3 but did not affect H3K9 and H4K20 methylation (Fig. 5d and Extended Data Fig. 9b). In line with this finding, overexpression of enzymes in CMs that increase αKG levels, such as those encoded by *Idh3b* and *Idh3g*, stimulated CM proliferation, markedly reduced H3K4me3 deposition and lowered mRNA levels of cardiac maturation-related genes (Extended Data Fig. 9c–h). Treatment of neonatal CMs with CPI-455, a specific inhibitor of the αKG-dependent H3K4 demethylase KDM5, or cell-permeable R2HG, a competitive inhibitor of αKG[29], prevented enhanced cell cycle activity induced by αKG or overexpression of *Idh3b* or *Idh3g* (Fig. 5e,f and Extended Data Fig. 9g–k). As expected, treatment with CPI-455 altered the expression of cardiac maturation-related genes accordingly (Extended Data Fig. 9l).

Pharmacological inhibition of αKG-dependent effects by the KDM5 inhibitor CPI-455 suggested that effects of αKG on the chromatin and on gene expression are relayed by members of the *Kdm5* gene family. This hypothesis was confirmed by overexpression of *Kdm5b*, which strongly reduced H3K4me3 levels and markedly increased CM cell cycle activity when combined with αKG treatment (Fig. 5g,h and Extended Data Fig. 10a–d). *Kdm5b* plays important roles during heart development, and its expression progressively declines during CM

maturation[30]. *Kdm5b* knockdown mediated by short interfering RNA efficiently antagonized pro-proliferative effects of αKG, fully restoring reduced H3K4me3 levels that were lowered by αKG treatment, and thereby increasing expression of several key genes associated with CM maturation (Fig. 5i and Extended Data Fig. 10e–g). To validate potential targets of KDM5B, we carried out ChIP–qPCR experiments. We found that KDM5B preferentially binds to genes with broad H3K4me3 domains involved in maintaining the mature phenotype of CMs, such as *Tnni3*, *Mylk3* and *Myocd*. By contrast, no enrichment of KDM5B was found on housekeeping genes characterized by narrow peaks, such as *Dus1l*, *Snx19* and *Lsm6*. Notably, mRNA levels of these housekeeping genes were not altered in response to αKG, *Kdm5b* knockdown or combined treatment (Fig. 5j and Extended Data Fig. 10h). Taken together, our results indicate that accumulation of αKG caused by *Cpt1b* inactivation activates KDM5, which lowers broad H3K4me3 peaks in genes required for CM maturation, thereby reverting CMs to a more immature state and enabling CM cell cycle activity (Extended Data Fig. 10i).

## Discussion

Here we demonstrate that the cellular metabolism directly regulates H3K4me3 histone modifications at CM identity genes, thereby playing a pivotal role in controlling the maturation and proliferation of CMs. Metabolism-induced partial rewinding of the maturation program not only increases the proliferative capacity of CMs but also leads to various cardioprotective changes, including increased activation of adaptive HIF1 signalling[31] and αKG-dependent DNA damage repair[32]. Increased HIF1 signalling may also contribute to the relatively modest increase of CM surface area in *Cpt1b*^*cKO* mice[33], which was limited to larger CMs (> 300 μm²). *Cpt1b*-deficient CMs are characterized by reduced sarcomere density, a hallmark of neonatal CMs, which are able to proliferate but also support cardiac contractility. However, we do not know whether all CMs in *Cpt1b*-deficient hearts show the same propensity for cytokinesis, or whether only a distinct subset of CMs undergo cell division, which show a more severe disassembly of sarcomeres that cannot be easily identified on tissues sections. Furthermore, we observed improved coupling of glucose oxidation to glycolysis, which will reduce proton production and Ca²⁺ overload during ischaemia, preventing mitochondrial damage and cell death especially during reperfusion when the influx of O₂ stimulates FAO [34]. Thus, stimulation of both regeneration and cardioprotection seem to contribute to reduced scar formation in *Cpt1b*-deficient CMs after I–R injury. However, as inactivation of *Cpt1b* in CMs after I–R injury facilitates heart repair and restored cardiac functions—processes that cannot be explained by cardioprotection—we postulate that induction of cardiac regeneration dominates, a hypothesis that is also supported by the overall increase in CM numbers and in proliferating CMs in the infarct border zone.

The metabolic rewiring of CMs after abrogation of FAO causes complex transcriptional changes, which partially rewind the developmental program and endow CMs with a renewed proliferative potential. Metabolic rewiring does not only increase the flux from glucose, but also strongly increases the concentrations of catabolic intermediates generated during conversion of BCAAs to acetyl-CoA and/or succinyl-CoA, which fuel the Krebs cycle during anoxia and ischaemia[35]. Essentially, *Cpt1b*-deficient CMs adopt a metabolic profile that facilitates utilization of alternative energy substrates such as amino acids. We did not detect signs of lipotoxicity in *Cpt1b*^*cKO* and *Cpt1b*^*iKO* hearts, which has been reported for other *Cpt1b*-mutant mouse lines[36–38]. The absence of lipotoxicity might be explained by the catabolization of fatty acids in skeletal muscles, which is not affected in our cardiac-specific models, preventing accumulation of circulating free fatty acids and/or lipoproteins. Moreover, lowered expression levels of *Lpl* in hearts of *Cpt1b*^*cKO* mice will prevent efficient uptake of fatty acids into CMs owing to reduced hydrolysation of triacylglycerols in lipoproteins[39].

The marked, nearly twenty-fold accumulation of αKG is a hallmark of metabolic rewiring in *Cpt1b*-deficient CMs. In contrast to αKG, concentrations of other Krebs cycle metabolites were only marginally altered. αKG plays a crucial role in the Krebs cycle but also links metabolism to regulation of gene expression by serving as an essential co-substrate for αKG-dependent dioxygenases, including histone and DNA demethylases[40]. Our data indicate that the increase of αKG activates KDM5, leading to demethylation of H3K4me3 in cardiac identity genes that are required to maintain CMs in a mature state. H3K4me3 is a well-studied histone modification characteristic of actively transcribed genes. H3K4me3 shows a pronounced peak immediately downstream of the transcription start site, which can extend for more than 40 kilobases, probably contributing to increased PolII pausing and release, elongation and transcriptional strength[14,17]. Consistent with other studies[15,16], our findings show that genes with the broadest H3K4me3 peaks (top 25%) are transcriptionally more active than genes with narrow peaks and critical for cellular identity and maturation[14,27,28]. Recent studies questioned whether H3K4me3 is an instructive signal for gene activation, as reduction of H3K4me3 has a limited impact on global gene expression in yeast and mammalian cells[15]. However, numerous studies described an active role of H3K4me3 in gene expression (for example, by recruiting the NURF complex or by interacting with TAF3)[41–43]. In line with these reports, we identified 151 genes in *Cpt1b*-mutant CMs with reduced H3K4me3 peaks and diminished gene transcription. Of note, 92% of these genes contain broad or medium peaks and are mainly associated with cardiac development and contraction. We reason that cell-type-specific genes with broad H3K4me3 peaks are preferentially targeted by an αKG-dependent H3K4me3 demethylase. On the basis of inhibitor and overexpression studies, we reason that a member of the KDM5 family, most likely KDM5B, is the responsible H3K4me3 demethylase[30]. Future studies will unveil how precisely the αKG-dependent stimulation of a H3K4me3 demethylase achieves target selectivity and preferentially deactivates genes associated with cardiac development and contraction. We assume that genes containing broad H3K4me3 peaks provide more docking sites for KDM5 and therefore represent privileged targets. Genes with enhanced recruitment of KDM5 will automatically be more responsive to an increase of αKG and KDM5-mediated silencing. However, it is also possible that maturation-associated genes with broad H3K4me3 peaks possess other distinct features promoting enhanced recruitment of KDM5 (ref. 44).

Overall, the observation that a single metabolite, αKG, serves a pivotal role in connecting metabolic processes to transcriptional changes mediated by KDM5 offers numerous opportunities to manipulate the fate and function of CMs and stimulate cardiac regeneration.

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

# Methods

## Animals

*Cpt1b*<sup>fl/fl</sup> mice were generated in-house by using a targeting vector purchased from the European Conditional Mouse Mutagenesis Program, in which exons 10–11 of the *Cpt1b* gene are flanked by two *loxP* sites. Generation of *Ogdh*<sup>fl/fl</sup> animals, in which exons 3 and 4 were flanked by *loxP* sites, have been described before[26]. *αMHC-Cre*<sup>Pos/+</sup> and *αMHC-MCM*<sup>pos/+</sup> mice were obtained from The Jackson Laboratory. C57BL/6 mice were obtained from Charles River. Primers used for genotyping are listed in the Supplementary Information. All mice were maintained in individually ventilated cages, at $22.5\,°C \pm 1\,°C$ and a relative humidity of $50\% \pm 5\%$ with controlled illumination (12 h dark/light cycle). Mice were given ad libitum access to food and water. TAM (Sigma) was administered intraperitoneally at 75 mg per kilogram of BW daily for 10 days. In experiments, in which genes were inactivated by Cre recombinase-mediated recombination, corresponding Cre recombinase-expressing strains without the floxed target genes were always used as negative controls, unless indicated otherwise. *αMHC-MCM* control mice were subjected to the same TAM treatment as in the actual gene inactivation experiment. All mice were maintained on a C57BL/6 background, and littermates were used as controls in all experiments. Animals were assigned to different groups according to genotypes. The genotype of animals, from which individual samples were taken, was not known to the investigator, and experiments were performed in a blinded manner. After data collection, individual genotypes were revealed and the animals were assigned to separate groups for further data analysis. Observed results did not differ between male mice and female mice. No sex-specific experiments were performed. Sample sizes were determined on the basis of established practice and applicable standards. We opted for sample sizes that are commonly used sample sizes in the field. For in vivo studies, a minimum of three biological replicates were analyzed. All animal experiments were carried out in accordance with the Guide for the Care and Use of Laboratory Animals published by the US National Institute of Health (NIH Publication No. 85-23, revised 1996) and according to the regulations issued by the Committee for Animal Rights Protection of the State of Hessen (Regierungspraesidium Darmstadt, Wilhelminenstr. 1–3, 64283 Darmstadt, Germany) with the project numbers B2/1137, B2/1125 and B2/2034.

## Neonatal CM isolation and culture in vitro

Neonatal hearts were dissected from P0–1 C57bl/6 pups, washed with ice-cold PBS and dissociated using standard procedures. Neonatal CMs were seeded in culture plates precoated with fibronectin (0.8–1 million per 3.5-cm dish or 0.25 to 0.4 million per well of 2-well chamber slides) and cultured in primary neonatal CM culture medium (80% DMEM with 4.5 g l⁻¹ glucose, 20% Medium 199, 5% FCS and 100 U ml⁻¹ penicillin and streptomycin). After overnight culture, chemicals were added to the medium and cells were further cultured for 72–96 h before collection. Concentrations of chemical were as follows: etomoxir 100 μM (Cayman, 11969); octyl-αKG 500 μM (Cayman, 11970); CPI-HCl 25 μM (Selleckchem, S8287); R2HG 500 μM (Cayman, 16366). *Kdm5b* knockdown was achieved through transfection of pooled *Kdm5b* short interfering RNA (Dharmacon) with DharmaFECT 1 Transfection Reagent. Non-targeting pooled short interfering RNAs were used for the control group.

## Immunofluorescence and histological analysis

Hearts were immediately fixed in 4% PFA after dissection. For trichrome staining after I–R injury, hearts were embedded in paraffin and continuously sectioned from the apex to the ligation site. Every second section from hearts of each group was used for staining and quantification. In vitro-cultured neonatal CMs were fixed with 4% PFA for 10 min at room temperature and permeabilized (0.3% Triton X-100 and 5% BSA) for 1 h at room temperature. To determine the surface area of CMs, approximately 120 CMs randomly selected from 5–6 paraffin sections of each heart sample were measured using the ImageJ software tool. The sarcomere density of isolated adult CMs was analysed using the ImageJ software tool. For other quantification procedures, using either isolated CMs or tissue sections, hundreds of CMs were analysed for each individual heart. Values from each group of CMs (or sections) were averaged and are presented as one sample ($n = 1$). For experiments with neonatal CMs, values for each sample represent the results obtained from one isolation of CMs from pooled neonatal hearts. Antibodies for immunofluorescence staining are listed in the Supplementary Information. Microscopic images were acquired with a fluorescence stereomicroscope (Leica M205 FA). Regular immunofluorescence images were acquired with a fluorescence microscope (Zeiss Imager Z1) and processed with ZEN 2 imaging software. Confocal immunofluorescence images were acquired with a Leica SP8 confocal microscope and processed with LAS X software 3.5.7.23225. Acquisition of histological images was carried out with a light microscope (Zeiss Axioplan2).

## EdU incorporation assay

EdU and other reagents were prepared according to the manufacturer's instructions (ThermoFisher C10339). In vivo EdU incorporation assays were carried out according to previous publications[45]. To analyse EdU incorporation in cultured neonatal CMs, cells in 2-well chamber slides were labelled with 10 μM EdU for 12 h. After two washes with pre-warmed PBS, cells were fixed with 4% PFA for 10 min at room temperature and EdU incorporation was visualized using the Click-iT EdU kit (Invitrogen), following the manufacturer's protocol.

## Western blot assays

Freshly isolated or cultured cells were washed with ice-cold PBS and lysed in cell lysis buffer (20 mM Tris (pH 7.5), 400 mM NaCl, 1 mM EDTA, 1 mM EGTA, 1% Triton X-100, 1× Complete Protease Inhibitor Cocktail (Roche Diagnostics)) for 10 min on ice, followed by sonication using the Bioruptor (Dianagene) at 4 °C for 5 min. Proteins were separated by SDS–polyacrylamide gel electrophoresis and transferred to nitrocellulose membranes (Millipore). Proteins detected by antibodies were visualized using an enhanced chemiluminescence detection system (GE Healthcare) and quantified using the ChemiDoc gel documentation system (Bio-Rad). Antibodies and dilutions used in this study are listed in the Supplementary Information.

## Adult CM isolation and in vitro culture

Isolation of adult CMs was carried out as described previously[46]. In brief, dissected hearts were cannulated through the aorta and retrogradely perfused with calcium-free buffer. Cannulated hearts were enzymatically digested by perfusion with enzyme buffer solution and cut off from the cannula. Atria were separated, and ventricles were minced in enzyme buffer. After gentle pipetting, myocytes were centrifuged at 500 r.p.m. for 1 min and cell pellets containing CM fractions were resuspended in stop buffer. The calcium content of the cell suspension was then stepwise adjusted to 1 mM and CM-containing cell pellets were resuspended in M199 cell culture medium, supplemented with creatinine, L-carnitine, HEPES, penicillin–streptavidin, 5% FCS and insulin–transferrin–sodium selenite medium supplement. Cells were seeded in dishes precoated with laminin and maintained in a humidified incubator at 37 °C and 5% $CO_2$. To determine CM numbers in adult hearts, the dissected heart was washed with ice-cold PBS and fixed with 1% PFA overnight. After washing with ice-cold PBS, hearts were cut into 1–2-mm³ pieces and incubated with digestion buffer (PBS containing 0.5 U ml⁻¹ of collagenase B (Roche no. 11088807001) and 0.2% NaN₃) with constant shaking at 1,000 r.p.m. at 37 °C. Every 12–24 h, digested CMs were collected, and new digestion buffer was added until the heart was fully digested. CMs were pooled, plated into a Sedgewick rafter chamber and counted therein.

## Measurement of oxygen consumption rate with Agilent Seahorse XF

Adult CMs were isolated and seeded at a density of 6,000 cells per well in a 96-well plate for Seahorse measurements (Agilent Seahorse XFe96 Analyzer). Cells were washed with PBS and Seahorse base medium after attachment to the plate. The following substrates were added as energy substrates to the medium 1 h before measurements of the oxygen consumption rate: glucose 5 mM (Sigma), pyruvate 0.2 mM (Sigma), glutamine 4 mM (Sigma), palmitate–BSA 0.2 mM (Agilent), BSA control (Agilent), carnitine 0.2 mM (Sigma), valine 1 mM (Sigma), isoleucine 1 mM (Sigma), leucine 1 mM (Sigma), sodium propionate 0.05 mM (Sigma), sodium acetate 0.05 mM (Sigma), sodium octanoate 0.1 mM (Sigma), sodium decanoate 0.1 mM (Sigma). The oxygen consumption rate was measured using the Mito Stress Test kit (Agilent). The following inhibitors were injected: oligomycin (2 µM), FCCP (2 µM), rotenone and antimycin A (1 µM).

## Magnetic resonance imaging and data processing

Cardiac magnetic resonance imaging (MRI) measurements were carried out using a 7.0T Bruker Pharmascan (Bruker) equipped with a 760 mT m$^{-1}$ gradient system, using a cryogenically cooled four-channel phased array element $^1$H receiver coil (CryoProbe), a 72-mm room-temperature volume resonator for transmission, and the IntraGate self-gating tool[47]. Electrocardiogram parameters were adapted for one heart slice and transferred afterwards to the navigator signals of the remaining slices. Thus, in-phase reconstruction of all pictures was guaranteed. Measurements are based on the gradient echo method (repetition time = 6.2 ms; echo time = 1.3 ms; field of view = 2.20 × 2.20 cm; slice thickness = 1.0 mm; matrix = 128 × 128; oversampling = 100). The imaging plane was localized using scout images showing the two- and four-chamber view of the heart, followed by acquisition of images in short-axis view, orthogonal on the septum in both scouts. Multiple contiguous short-axis slices consisting of 7 to 10 slices were acquired for complete coverage of the left and right ventricle. Mice were measured under isoflurane (1.5–2.0% in oxygen and air with a flow rate of 1.0 l min$^{-1}$) anaesthesia. Body temperature was maintained at 37 °C by a thermostatically regulated water flow system during the entire imaging protocol. MRI data were analysed using Qmass digital imaging software (Medis Imaging Systems, Leiden, the Netherlands).

## FACS-based isolation of cardiac nuclei

Ventricles were washed with ice-cold PBS after dissection and snap frozen in liquid $N_2$. For isolation of cardiac nuclei, the frozen ventricle was thawed in 3 ml lysis buffer (5 mM $CaCl_2$, 3 mM MgAc, 2 mM EDTA, 0.5 mM EGTA and 10 mM Tris-HCl, pH 8) in M-tubes (Miltenyi Biotec) and homogenized using the gentleMACS Dissociator (Miltenyi Biotec), following the manufacturer's protocol (protein_01). The resultant homogenate was mixed with lysis buffer containing 0.4% Triton X-100, incubated on ice for 10 min, and subsequently filtered through 40-µm cell strainers (BD Bioscience). The flow-through was centrifuged at 1,000g for 5 min at 4 °C to collect nuclei. Nuclei were further purified by centrifugation at 1,000g for 5 min at 4 °C through a 1 M sucrose cushion (3 mM MgAc, 10 mM Tris-HCl, pH 8) and then stained with a PCM1 antibody in nuclei stain buffer (DPBS, 1% BSA, 0.2% Igepal CA-630, 1 mM EDTA). DNA was stained by DAPI before FACS. FACS was carried out using a FACSAria III (BD Biosciences). Quantification of PCM$^+$ cardiac nuclei and DNA content was carried out with the LSR Fortessa (BD Biosciences) analyser. Data acquisition and analysis were accomplished using the BD FACS Diva v8 software. The gating strategy is shown in the Supplementary Information.

## RNA-seq and data analysis

RNA was extracted from isolated adult CMs using the Direct-zol Total Kit (Zymo Research) combined with on-column DNase digestion (DNase-Free DNase Set, Qiagen) to avoid contamination by genomic DNA. RNA and library preparation integrity were verified using the LabChip Gx Touch 24 (Perkin Elmer). A 200 ng quantity of total RNA was used as input for the SMARTer Stranded Total RNA Sample Prep Kit - HI Mammalian (Clontech) following the manufacturer's instructions. Sequencing was carried out on a NextSeq500 instrument (Illumina) using v2 chemistry, resulting in an average of 22 million reads per library with a 1 × 75 bp single-end setup. Raw reads were assessed for quality, adapter content and duplication rates with FastQC 0.11.8 (http://www.bioinformatics.babraham.ac.uk/projects/fastqc). Trimmomatic version ≥ 0.36 was used to trim reads after a quality drop below a mean of Q15 in a window of five nucleotides[48]. Only reads of at least 15 nucleotides were cleared for subsequent analyses. Trimmed and filtered reads were aligned versus mouse genome version mm10 (GRCm38.p5) using STAR ≥ 2.5.4b with the parameters --outFilterMismatchNoverLmax 0.1 --alignIntronMax 200000 (ref. 49). The number of reads aligning to genes was counted with featureCounts ≥ 1.6.0 from the Subread package[50]. Only reads mapping at least partially inside exons were admitted and aggregated per gene. Reads overlapping multiple genes or aligning to multiple regions were excluded. Differentially expressed genes were identified using DESeq2 version ≥ 1.14.0 (ref. 51). Genes were classified as significantly differentially expressed with a $P$ value < 0.05. Annotations were enriched using UniProt data (release 24.03.2017) based on Ensembl gene identifiers (Activities at the Universal Protein Resource (UniProt)).

## ChIP, ChIP–seq and data analysis

Chromatin was prepared using the truChIP Chromatin Shearing Kit (COVARIS) and sheared to an average size of 200–500 bp by sonication (Diagenode Bioruptor). Protein–DNA complexes were immunoprecipitated with IgG or KDM5B antibodies, followed by incubation with Protein A/G magnetic beads (Dynabeads, Invitrogen). For ChIP–qPCR, beads were washed and protein–DNA complexes were eluted and purified using 10% Chelex-100 (w:v, Bio-Rad Laboratories) in Tris–EDTA. Immunoprecipitated chromatin was analysed by qPCR using SYBR Green quantitative real-time analysis with primers that are listed in the Supplementary Information. A detailed description of ChIP–seq analysis is provided in the Supplementary Information.

## I–R injury and measurement of AAR and infarct area out of AAR

Animals were anaesthetized using 4.5% isoflurane and endotracheally intubated with a 22-gauge intravenous catheter. Mice were placed on a 37 °C heating plate in the supine position and ventilated at a rate of 225 strokes min$^{-1}$ and a stroke volume of 250 µl with a mixture of oxygen and 1.5% isoflurane using a MiniVent rodent ventilator. Chest hair was removed, and skin was disinfected and opened with a small incision of several millimetres in length from the left armpit to the sternal border. Pectoralis major and minor muscles were separated, the chest was opened in the third intercostal space, and retractors were inserted. Next, the pericardium was opened to access the heart. The left coronary artery was ligated for 30 min and reopened for reperfusion in a proximal position using a prolene suture (7-0). The retractors were removed, and the chest wall was closed by bringing together the second and third rib using a vicryl suture (5-0). The muscles were placed into their original position, and the skin incision was closed with vicryl (5-0). Mice were ventilated with oxygen until awakening, followed by extubation, and placement into their cages. At 24 h after I–R surgery, the animals were euthanized for AAR and infarct area out of AAR measurement, for which hearts were removed and the aorta quickly cannulated for an injection of 500 µl 1% Evan's blue solution into the ventricle. Hearts were kept on ice-cold saline for further investigations. Afterwards, the heart was frozen and sliced at 0.5 mm. Heart sections were subsequently stained with TTC solution (1% in PBS) at 37 °C for 30 min and then fixed with formalin solution.

## Viability assay of adult CMs under hypoxic conditions

Freshly isolated adult CMs seeded in chamber slides were cultured either under normoxia or in a hypoxia chamber with 1% $O_2$, 5% $CO_2$ at 37 °C for 18 h. Cells were washed at room temperature with PBS and incubated with PBS containing ethidium homodimer 1 (4 μM) and calcineurin (2 μM) for 45 min at room temperature. Immunofluorescence images were acquired with a Zeiss Imager Z1 microscope.

## Metabolic flux assays and targeted metabolic analysis

Metabolic flux assays were carried out with isolated Langendorff-perfused hearts from $Ctrl^{Cre}$ and $Cpt1b^{cKO}$ mice. Hearts were quickly excised and cannulated through the aorta. The cannulated heart was connected to a perfusion column apparatus maintained at 37 °C using a temperature-controlled water bath. Hearts were perfused retrogradely for 60 min with Krebs−Henseleit buffer with the following substrates: glucose 8 mM; pyruvate 0.12 mM; palmitate−BSA 0.4 mM; isoleucine 0.176 mM. In each perfusion assay, only one metabolite was replaced by a $^{13}C$-labelled metabolite ([$^{13}C$]glucose, [$^{13}C$]isoleucine or [$^{13}C$]palmitate−BSA). Subsequently, hearts were snap frozen, pulverized in liquid nitrogen and subjected to metabolite extraction (acyl-CoAs or metabolites of the Krebs cycle) and quantification using liquid chromatography with triple-quadrupole mass spectrometry. Metabolic flux assays and α-ketoacids measurements were carried out using tissue from isolated hearts after perfusion with different substrates as indicated. Measurements of Krebs cycle metabolites and standardized targeted metabolic analysis were carried out with isolated adult CMs. A detailed description of the quantification of acyl-CoAs, Krebs cycle metabolites, α-ketoacids and targeted metabolome analysis is provided in the Supplementary Information.

## Lentiviral transduction of CMs

HEK293T cells were grown in DMEM (Sigma) supplemented with 10% FCS (Sigma), 2 mM L-glutamine, 100 U penicillin and 100 μg ml$^{-1}$ streptomycin at 37 °C, 5% $CO_2$. HEK293T cells ($2 \times 10^6$ per 10-cm dish) were transfected with 5 μg pLJM1-Kdm5b, pLJM1-Idh3b or pLJM1-Idh3g, 4.5 μg psPAX2 (Addgene, no. 12260) and 0.5 μg pMD2.G (Addgene, no. 12259) using the Turbofect transfection reagent and Opti-MEM for 6−8 h. The supernatants containing lentiviral particles were collected at 48 and 72 h after transfection and pooled. Lentiviruses were filtered through a 0.45 μM cell strainer to remove HEK293T cells and concentrated with a Lenti-X concentrator according to the manufacturer's instructions (TaKaRa, 631231). Primary neonatal CMs were infected in suspension with Polybrene (8 μg ml$^{-1}$) for 6 to 8 h.

## Analysis of gene expression using qPCR with reverse transcription and assessment of mitochondrial DNA copy numbers

Total RNA was extracted using the TRIzol reagent (Invitrogen) according to the manufacturer's instructions. RNA was reverse transcribed with Superscript II (Invitrogen) following standard procedures. Real-time PCR was carried out with two technical replicates using the StepOne real-time PCR system and KAPA SYBR FAST qPCR Master Mix (KAPA Biosystems). Relative quantification of gene expression was carried out using the ΔΔCT method. The Ct values of the target genes were normalized to expression of the *36b4* gene using the equation $\Delta Ct = Ct_{reference} - Ct_{target}$ and expressed as ΔCt. Relative mRNA expression values were shown with the average from control samples set as 1. Mitochondrial DNA copy numbers were determined using DNA extracted from isolated adult CMs. The data were normalized to internal controls (*H19* or *Mx1*) and cell numbers. Primers and PCR conditions are listed in the Supplementary Information.

## Electron microscopy

Hearts were isolated and fixed in 1.5% glutaraldehyde (v/v), 1.5% PFA (v/w) in 0.15 M HEPES (v/w), pH 8.0 at 4 °C for at least 24 h, and subsequently incubated with 1% osmium tetroxide for 2 h. Samples were stained en bloc with 50%-saturated watery uranyl acetate, followed by sequential ethanol dehydration (30%, 50%, 75%, 95%), and embedded in Agar 100. Ultrathin sections were cut using an ultramicrotome and image acquisition was carried out with a Philips CM10 electron microscope. All images were captured with a slow-scan 2k CCD (charge-coupled device) camera.

## Statistical analysis

For all quantitative analyses, a minimum of three biological replicates were analysed. Statistical tests were selected on the basis of the assumption that sample data are from a population following a probability distribution based on a fixed set of parameters. Student's $t$-tests were used to determine the statistical significance of differences between two groups. One-way AVOVA was used for multiple comparison tests. The following values were considered to be statistically significant: *$P < 0.05$, **$P < 0.01$, ***$P < 0.001$, ****$P < 0.0001$. Calculations were carried out using the GraphPad Prism 9 software package. Data are always represented as mean ± the standard error of the mean. No statistical method was used to predetermine sample size.

## Reporting summary

Further information on research design is available in the Nature Portfolio Reporting Summary linked to this article.

## Data availability

Data have been deposited in public databases. Sequencing data are available under the accession number GSE172415 and include results from ChIP−seq and RNA-seq experiments. RNA-seq data for neonatal CMs treated with DMSO and aKG are available under the accession number GSE217188. Source data are provided with this paper.

## Code availability

Codes used in the study are available at https://github.com/loosolab/Li_et_al_2023_heart_regeneration (https://doi.org/10.5281/zenodo.7828994). The codes have already been used and published in a different project[52].

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

**Acknowledgements** We thank S. Kreutzer, K. Mattonet, S. Krüger, K. Khrievono and K. Richter for technical support; and U. Gärtner for carrying out electron microscopy analysis. This work was supported by the Excellence Cluster Cardio Pulmonary System (CPI), the DFG Transregional Collaborative Research Centres 81 (TP A02) and 267 (TP A05), the DFG Collaborative Research Centres 1213 (TP A02 and B02) and 1531 (TP B08) and the German Centre for Cardiovascular Research. The work of M.P. was supported by the European Research Council Consolidator Grant EMERGE (no. 773047). M.P., X.Y. and T.B. are members of the German Center for Cardiovascular Research (DZHK).

**Author contributions** X.Y. and T.B. conceived and designed experiments. X.L. carried out most of the experiments, analysed the data and prepared figures. S.K., S.Z., I.F. and G.P. carried out the metabolic analysis. S.G. carried out next-generation deep sequencing. M.L., S.G. and C.K.

carried out bio-informatics analysis. T.Z. contributed to analysis of ChIP–seq experiments. M.W. conducted cardiac surgery. A.W. carried out the MRI measurements. M.P. provided transgenic mouse lines and critical comments. F.W. and A.A. contributed to experimental design, data analysis, discussions and advice. T.B., X.Y. and X.L. wrote the manuscript.

**Funding** Open access funding provided by Max Planck Society.

**Competing interests** The authors declare no competing interests.

**Additional information**
**Correspondence and requests for materials** should be addressed to Xuejun Yuan or Thomas Braun.

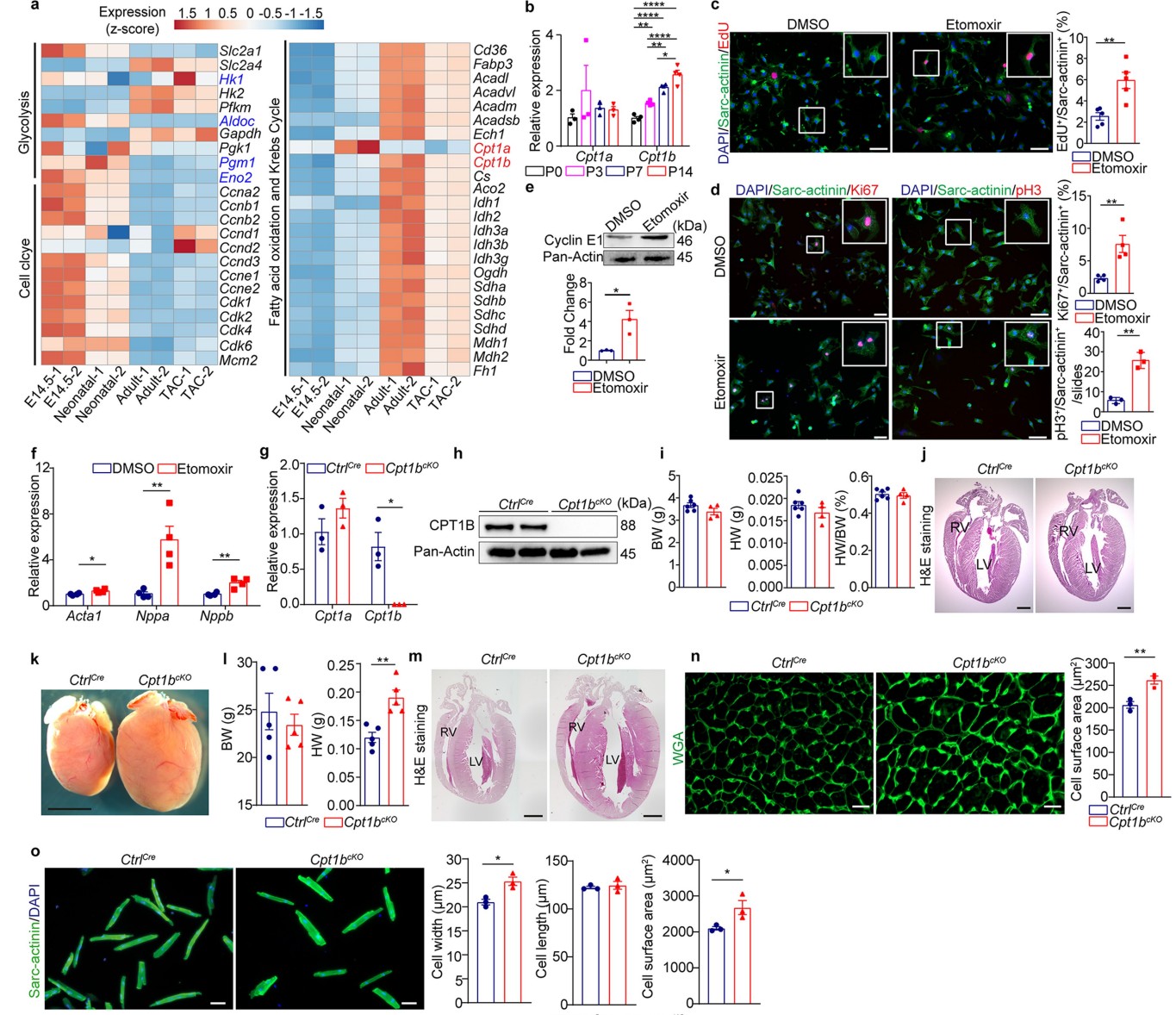

**Extended Data Fig. 1 | Inhibition of CPT1 stimulates cell cycle activity and hypertrophic growth of neonatal CMs. a**, Heat map representing expression of genes involved in glycolysis, cell cycle, fatty acid oxidation, and the Krebs Cycle. Publicly available RNA-seq data of CMs isolated from mice at E14.5, P1-2, P60, and adult mice one week after transaortic constriction were used (GSE79883). **b**, RT-qPCR analysis of *Cpt1a* (n = 3) and *Cpt1b* (n = 4) expression in mouse hearts at different developmental stages (P0, P3, P7, P14). *36b4* expression served as reference. **c**, EdU incorporation in neonatal CMs, 72 h after DMSO or etomoxir treatment (n = 5, each). Areas of enlarged images are labelled by white frames. Quantification of EdU⁺sarc-actinin⁺ CMs is in the right panel. Scale bar: 50 µm. **d**, Immunofluorescence staining of neonatal CMs for sarc-actinin and Ki67 (n = 4) or pH3 (Ser10) (n = 3) with or without etomoxir treatment. Quantification of Ki67⁺sarc-actinin⁺ and pH3⁺sarc-actinin⁺ CMs is in the right panels. Scale bar: 50 µm. **e**, Western blot analysis of Cyclin E1 in P0-1 CMs after 3-days-culture with DMSO or etomoxir (n = 3, each). Pan-actin served as loading control. **f**, RT-qPCR analysis of hypertrophy-associated genes in P0-1 CMs, 72 h after DMSO or etomoxir treatment (n = 4, each). *36b4* was used as reference gene. **g**, RT-qPCR analysis of *Cpt1a* and *Cpt1b* expression in CMs from 10-weeks-old *Ctrl^Cre* and

*Cpt1b^cKO* mice (n = 3, each). *36b4* served as reference gene. **h**, Western blot analysis of CPT1B in CMs from 10-weeks-old *Ctrl^Cre* and *Cpt1b^cKO* mice (n = 2, each). Pan-actin served as loading control. **i**, BW, HW, and HW/BW ratios of P7 *Ctrl^Cre* (n = 6) and *Cpt1b^cKO* (n = 4) mice. **j**, H&E-stained heart sections from P7 *Ctrl^Cre* and *Cpt1b^cKO* mice (n = 3, each). Scale bar: 500 µm. **k**, Images of hearts from 10-weeks-old *Ctrl^Cre* and *Cpt1b^cKO* mice (n = 3, each). Scale bar: 2 mm. **l**, BW and HW of 10-weeks-old *Ctrl^Cre* and *Cpt1b^cKO* mice (n = 5, each). **m**, H&E staining of heart sections from 10-weeks-old *Ctrl^Cre* and *Cpt1b^cKO* mice (n = 3, each). Scale bar: 600 µm. **n**, Immunofluorescence staining for α-WGA and quantification of cell surface areas on heart sections from 10-weeks-old *Ctrl^Cre* and *Cpt1b^cKO* mice (n = 3, each). Scale bar: 20 µm. **o**, Immunofluorescence images of sarc-actinin⁺ CMs from 10-weeks-old *Ctrl^Cre* and *Cpt1b^cKO* mice (n = 3, each). Quantifications of cell length, width, and surface area are in the right panels. Scale bar: 50 µm. Error bar represents mean ± s.e.m. N-numbers refer to individual mice in b, e-o. N-numbers refer to independent experiments in c, d. Two-tailed, unpaired student t-tests for statistical analysis in c-g, i, l, n-o. One-way ANOVA with Tukey tests for correction of multiple comparisons in b. *P < 0.05, **P < 0.01, ****P < 0.0001. For gel source data in e, h, see Supplementary Information.

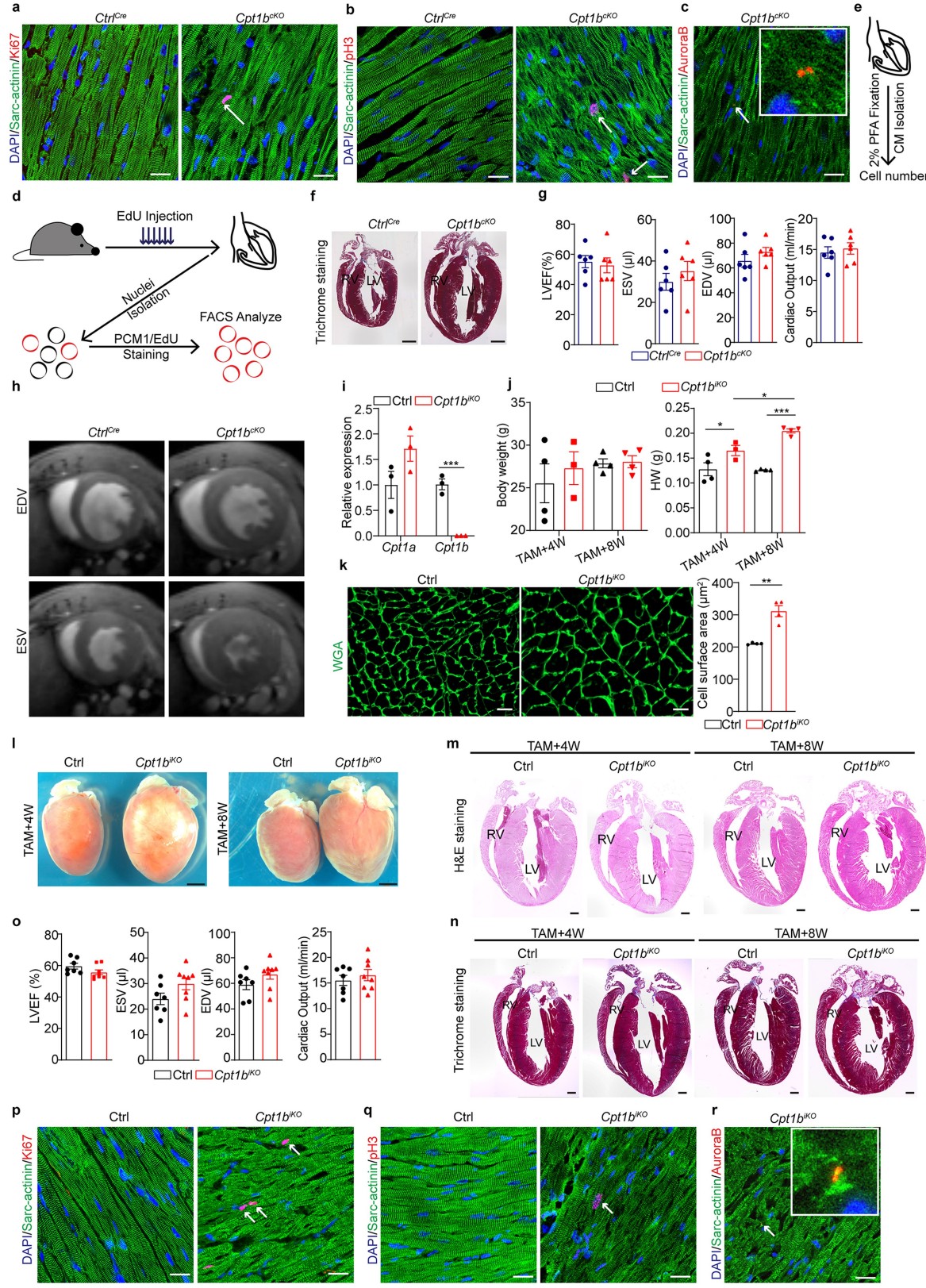

**Extended Data Fig. 2** | See next page for caption.

**Extended Data Fig. 2 | Deletion of *Cpt1b* in adult CMs reverses cell cycle arrest. a**–**c**, Immunofluorescence staining for Ki67 (a), pH3 (b), Aurora B (c), and sarc-actinin on heart sections from 10-weeks-old *Ctrl^Cre^* and *Cpt1b^cKO^* mice (n = 3, each). Scale bar: 50 μm. **d**, Outline of the EdU incorporation assays using isolated cardiac nuclei. **e**, Strategy for quantification of CMs numbers in adult mouse hearts. **f**, Trichrome staining of heart sections from 10-weeks-old *Ctrl^Cre^* and *Cpt1b^cKO^* mice (n = 3, each). Scale bar: 600 μm. **g**-**h**, Cardiac MRI analysis of 10-weeks-old *Ctrl^Cre^* and *Cpt1b^cKO^* mice (n = 6, each). **i**, RT-qPCR analysis of *Cpt1a* and *Cpt1b* expression in CMs from adult hearts of Ctrl and *Cpt1b^iKO^* mice, 4 weeks after termination of tamoxifen treatment for *Cpt1b* deletion (n = 3, each). *36b4* served as reference gene. **j**, Analysis of BW (left panel) and HW (right panel) of Ctrl and *Cpt1b^iKO^* mice 4 and 8 weeks after TAM injection (Ctrl 4 and 8 weeks after TAM, *Cpt1b^iKO^* 8 weeks after TAM, n = 4; *Cpt1b^iKO^* 4 weeks after TAM, n = 3). **k**, Immunofluorescence staining for α-WGA and quantification of cell surface areas on heart sections of Ctrl and *Cpt1b^iKO^* mice, 8 weeks after termination of tamoxifen treatment for *Cpt1b* deletion (n = 4, each). Scale bar: 20 μm. **l**, Macroscopic images of Ctrl and *Cpt1b^iKO^* hearts, 4 and 8 weeks after termination of tamoxifen treatment for *Cpt1b* deletion. Scale bar: 2 mm. **m**-**n**, H&E and Trichrome staining of heart sections from Ctrl and *Cpt1b^iKO^* mice, 4 and 8 weeks after termination of tamoxifen treatment for *Cpt1b* deletion (n = 3, each). Scale bar: 600 μm. **o**, MRI analysis of left ventricle ejection fraction (LVEF), ESV (End-Systolic Volume), EDV (End-Diastolic Volume) and cardiac output of Ctrl (n = 7) and *Cpt1b^iKO^* (n = 8) mice, 8 weeks after termination of tamoxifen treatment for *Cpt1b* deletion. **p**-**r**, Immunofluorescence staining for Ki67 (p), pH3 (q), Aurora B (r), and sarc-actinin on heart sections from Ctrl and *Cpt1b^iKO^* mice. Scale bar: 50 μm, p, n = 4; q and r, n = 3. Error bar represents mean ± s.e.m. N-numbers refer to individual mice. Two-tailed, unpaired student t-tests for statistical analysis in g,i,k,o. One-way ANOVA with Tukey tests for correction of multiple comparisons in j. *$P < 0.05$, ***$P < 0.001$, ****$P < 0.0001$.

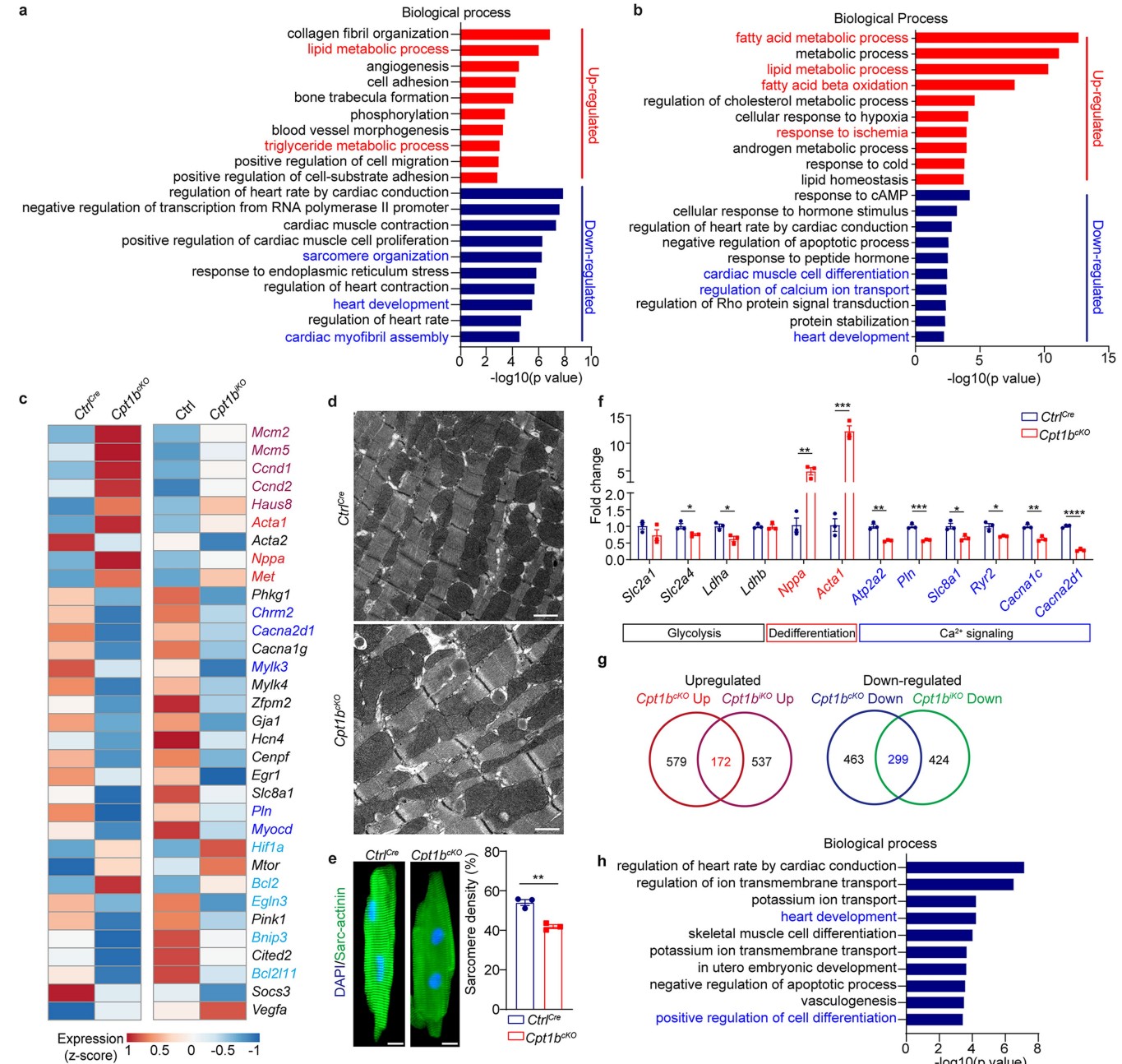

**Extended Data Fig. 3 | Abrogation of FAO reverts maturation of CMs.**
**a-b**, GO-term enrichment analysis of differentially expressed genes (DEGs) in *Cpt1b^cKO* and *Cpt1b^iKO* CMs. Enriched biological processes for up-regulated genes are in red and for down-regulated genes in blue. **c**, Heat map of selected DEGs involved in cell cycle (Magenta), maturation (red), contraction (blue), and HIF1A signaling (light blue) based on z-score transformed normalized DESeq counts. **d**, EM images showing the cytoarchitecture of CMs isolated from 10-weeks-old *Ctrl^Cre* and *Cpt1b^cKO* mice (n = 3, each). Scar bar: 500 nm. **e**, left: IF staining of isolated single CMs from *Ctrl^Cre* and *Cpt1b^cKO* adult hearts for sarcomeric actinin (identical exposure time for both CMs); right: quantification of sarcomere density based on IF staining for sarcomeric actinin, comparing *Ctrl^Cre* and *Cpt1b^cKO* adult CMs (n = 3, each). Scale bar: 10 μm. **f**, RT-qPCR analysis of selected genes in CMs from 10-weeks-old *Ctrl^Cre* and *Cpt1b^cKO* mice. *36b4* served as reference gene (n = 3, each). **g**, Overlap of up-regulated (left panel) and down-regulated (right panel) genes in *Cpt1b^cKO* and *Cpt1b^iKO* CMs compared to control CMs. **h**, GO term enrichment analysis of genes from the overlap shown in (g). Error bar represents mean ± s.e.m. N-numbers refer to individual mice. Two-tailed, unpaired student t-tests for statistical analysis in (e,f). *$P < 0.05$, **$P < 0.01$, ***$P < 0.001$, ****$P < 0.0001$.

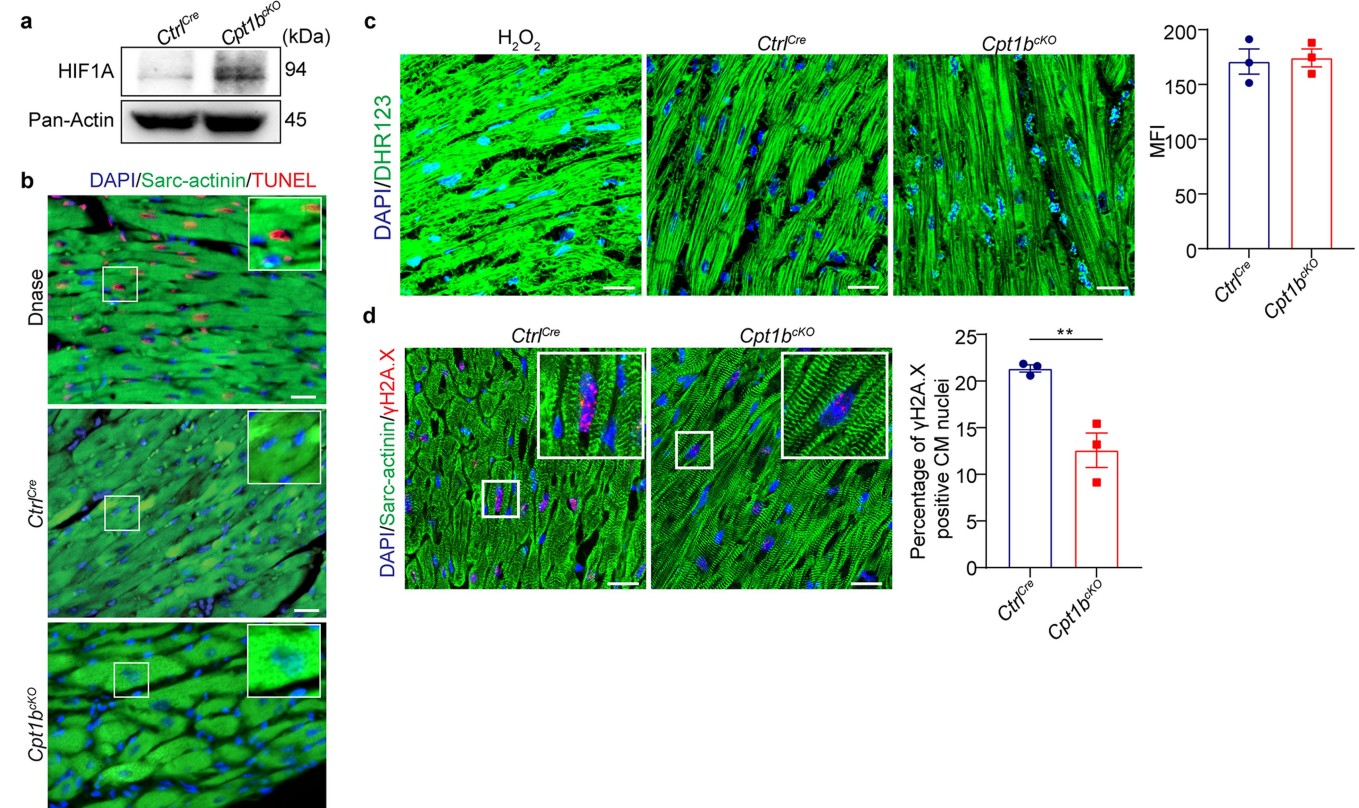

**Extended Data Fig. 4 | Inhibition of FAO in CMs increases HIF1A levels and reduces DNA damage. a**, Western blot analysis of HIF1A in CMs from 10-weeks-old *Ctrl^Cre* and *Cpt1b^cKO* mice (n = 3, each). Pan-actin was used as loading control. For gel source data, see Supplementary Information. **b**, TUNEL assays on heart sections from 20–25-weeks-old *Ctrl^Cre* and *Cpt1b^cKO* mice (n = 3, each). DNase-treated samples were used as positive control. Scale bar: 20 μm. **c**, Immunofluorescence staining of 10-weeks-old *Ctrl^Cre* and *Cpt1b^cKO* heart sections using DAPI and DHR123. $H_2O_2$ treated heart sections were used as positive controls. Quantification of mean fluorescence intensity (MFI) per image is shown in the right panel (n = 3, each). **d**, Immunofluorescence staining of 10-weeks-old *Ctrl^Cre* and *Cpt1b^cKO* heart sections using γH2A.X and sarc-actinin antibodies. Quantification of the percentage of γH2A.X⁺sarc-actinin⁺ CMs per heart section is shown in the right panel (n = 3, each). Error bar represents mean ± s.e.m. N-numbers refer to individual mice. Two-tailed, unpaired student t-tests for statistical analysis in c and d. **P < 0.01.

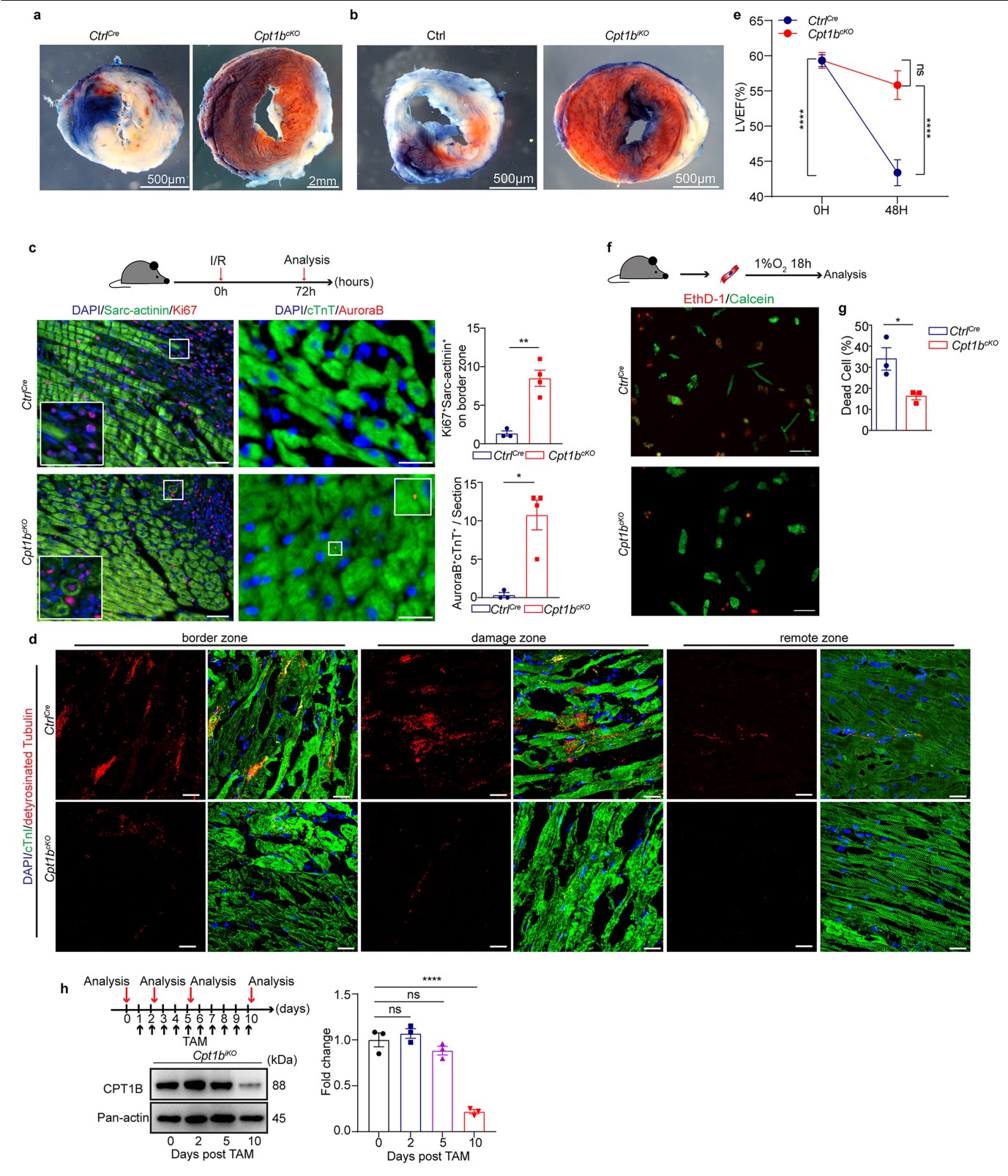

**Extended Data Fig. 5** | See next page for caption.

**Extended Data Fig. 5 | Abrogation of FAO by deletion of *Cpt1b* protects from I/R damage and enables heart regeneration. a**, **b**, Images of *Ctrl^Cre^* and *Cpt1b^cKO^* (a) and Ctrl and *Cpt1b^iKO^* (b) heart sections after Evans blue injection and TTC staining, 24 h after I/R injury. I/R surgery was done using 7-weeks-old mice (a) or 14-weeks-old mice, 3 weeks after completion of TAM treatment (b). **c**, Staining for Ki67 and Sarc-actinin (left panel), and Aurora B and cTnT (right panel) on sections from hearts of adult *Ctrl^Cre^* and *Cpt1b^cKO^* mice, 72 h after I/R injury (*Ctrl^cre^* n = 3, *Cpt1b^cKO^* n = 4). Scale bar: 50 μm (left panel), 20 μm (right panel). **d**, Immunofluorescence staining for detyrosinated tubulin and cTnI using sections from *Ctrl^Cre^* and *Cpt1b^cKO^* hearts, 24 h after IR surgery (n = 3, each). **e**, MRI-based assessment of heart functions in *Ctrl^Cre^* and *Cpt1b^cKO^* mice before and 48 h after I/R surgery (0 h, *Ctrl^Cre^* n = 8, *Cpt1b^cKO^* n = 7; 48 h, n = 7, each).

**f**, **g**, Cell viability assay of CMs isolated from adult *Ctrl^Cre^* and *Cpt1b^cKO^* hearts after exposure to 1% $O_2$ for 18 hrs. The experimental approach is outlined in the upper panel. Quantification of dead cells (EthD-1$^+$) is shown in g (n = 3, each). **h**, Western blot analysis and quantification of CPT1B levels in *Cpt1b^iKO^* mice, 0, 2, 5, and 10 days after initiation of TAM treatment (n = 3, each). Pan-actin served as loading control. For gel source data, see Supplementary Information. Error bars represent mean ± s.e.m. N-numbers refer to independent experiments in g, others refer to individual mice. Two-tailed, unpaired student t-tests for statistical analysis in c, g. Two-way ANOVA with Tukey tests for correction of multiple comparisons in e and one-way ANOVA with Tukey tests for correction of multiple comparisons in h. ns: not significant, *$P < 0.05$, **$P < 0.01$, ****$P < 0.0001$.

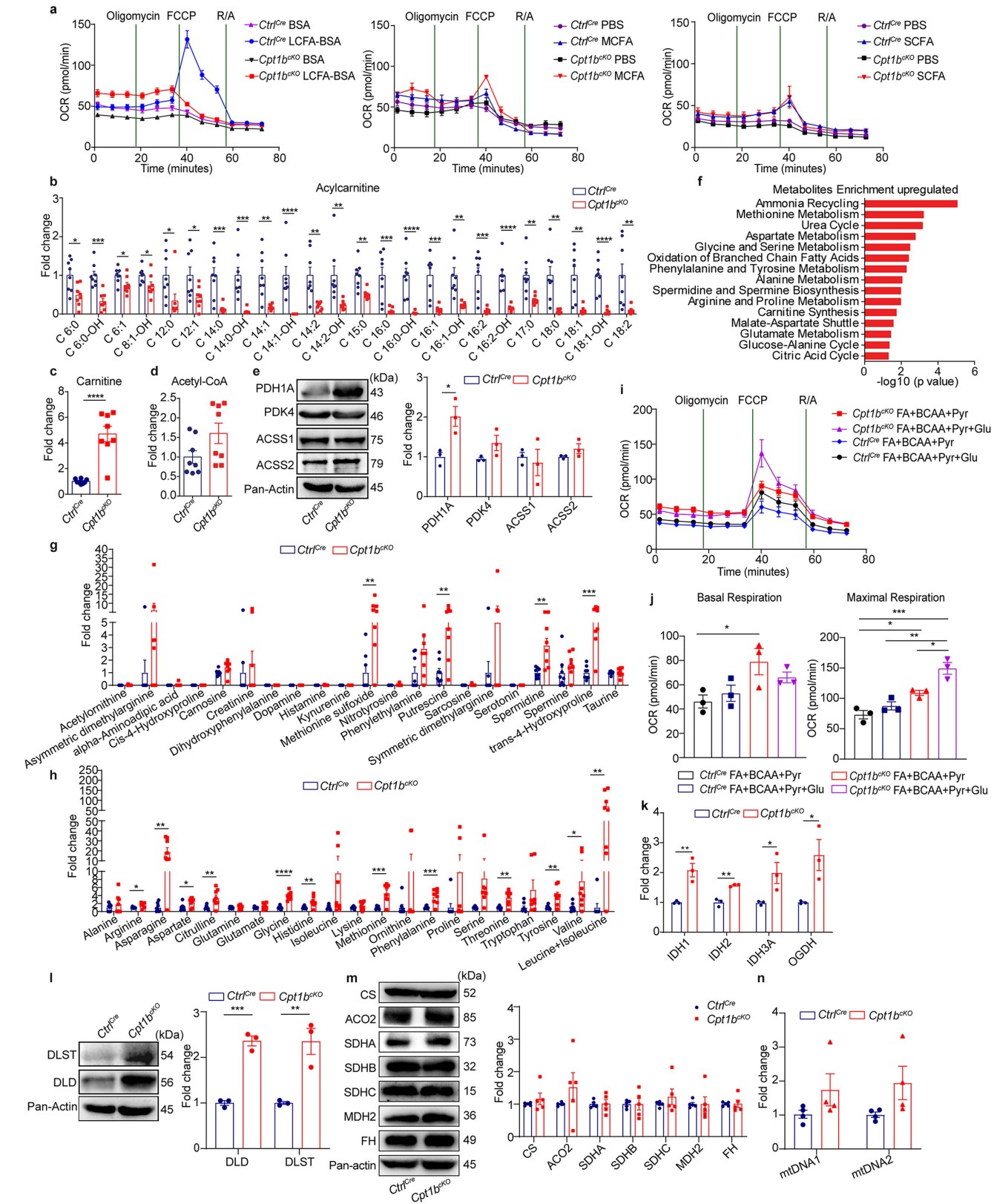

**Extended Data Fig. 6** | See next page for caption.

**Extended Data Fig. 6 | Characterization of metabolic changes in *Ctp1b*-deficient CMs. a**, Oxygen consumption rate (OCR) measurements of adult CMs isolated from *Ctrl^Cre^* and *Cpt1b^cKO^* hearts after addition of Palmitate-BSA (long chain fatty acids (LCFA), left panel), medium chain fatty acids (MCFA, middle panel) and short chain fatty acids (SCFA, right panel) using the Mito Stress kit for Seahorse instruments. **b**–**d**, Quantification of acylcarnitine (b), free carnitine (c), and acetyl-CoA (d) in CMs isolated from adult *Ctrl^Cre^* and *Cpt1b^cKO^* mice (n = 8, each). **e**, Western blot analysis and quantification of enzymes involved in acetyl-CoA production (n = 3, each). Pan-actin was utilized as loading control. **f**, Activated metabolic pathways in adult *Cpt1b^cKO^* CMs based on targeted metabolome analysis. **g**, **h**, Fold-changes of amino acids (g) and biogenic amines (h) in *Cpt1b^cKO^* compared to *Ctrl^Cre^* CMs (n = 8, each). **i**, **j**, OCR analysis of adult CMs from *Ctrl^Cre^* and *Cpt1b*^cKO^ mice using the Mito Stress kit for Seahorse instruments. Palmitate, glucose, BCAA and pyruvate were added as substrates for energy production. Basal and maximal respiration rates are shown in j (n = 3, each). **k**, Quantification of western blot analysis for enzymes involved in αKG production (n = 3, each). **l**, Western blot analysis of enzymes of the αKG dehydrogenase complex in adult *Ctrl^Cre^* and *Cpt1b^cKO^* CMs (n = 3, each). **m**, Western blot analysis of Krebs Cycle enzymes in adult *Ctrl^Cre^* and *Cpt1b^cKO^* CMs (n = 5, each). **n**, Quantification of mitochondrial DNA copy number using DNA from adult CMs of *Ctrl^Cre^* and *Cpt1b^cKO^* mice (n = 4, each). *H19* and *Mx1* were used as controls. Error bar represents mean ± s.e.m. N-numbers refer to individual mice. Two-tailed, unpaired student t-tests for statistical analysis in b-e, g-h, k-n. One-way ANOVA with Tukey tests for correction of multiple comparisons in j. *$P < 0.05$, **$P < 0.01$, ***$P < 0.001$, ****$P < 0.0001$. For gel source data in e,l,m, see Supplementary Information.

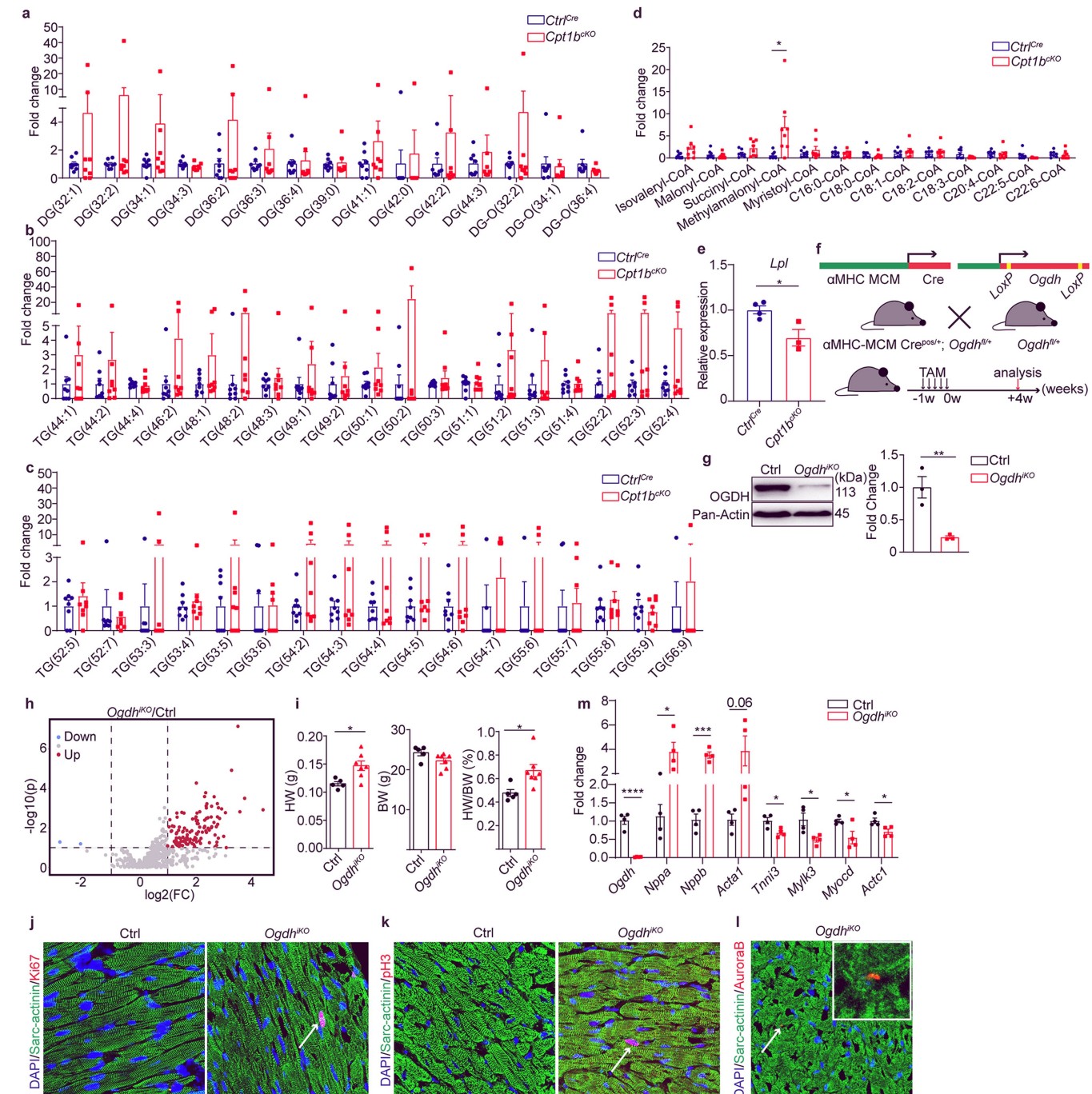

**Extended Data Fig. 7 | CM-specific inactivation of *Ogdh* in adult mice induces hyperplastic and hypertrophic cardiac growth, similar to *Cpt1b*-deficient mice. a–d,** Fold-changes of diacylglycerols (a), triacylglycerols (b,c) and CoAs (d) in *Cpt1b^cKO^* compared to *Ctrl^Cre^* CMs (n = 8, each). **e,** RT-qPCR expression analysis of *Lpl* in CMs isolated from adult *Ctrl^Cre^* and *Cpt1b^cKO^* mice (*Ctrl^Cre^* n = 4, *Cpt1b^cKO^* n = 3). *36b4* served as reference gene. **f,** Generation of *Ogdh* conditional KO mice. **g,** Western blot analysis of OGDH in CMs from Ctrl and *Ogdh^iKO^* mice, 3-4 weeks after termination of tamoxifen treatment for *Ogdh* deletion (n = 3, each). Pan-actin was used as sample processing control. For gel source data, see Supplementary Information. **h,** Volcano plot showing differentially present metabolites in CMs of Ctrl and *Ogdh^iKO^* mice. Red dots indicate increased and blue dots reduced concentrations of metabolites in *Ogdh^iKO^* compared to Ctrl CMs. **i,** HW, BW and HW/BW ratios of Ctrl and *Ogdh^iKO^* mice, 3-4 weeks after termination of tamoxifen treatment for *Ogdh* deletion (Ctrl n = 5, *Ogdh^iKO^* n = 7). **j–l,** Immunofluorescence analysis of heart sections from Ctrl and *Ogdh^iKO^* mice for Ki67 (j), pH3 (k) and Aurora B (l), counterstained for sarc-actinin to identify CMs. Scale bar: 50 μm, (j, Ctrl n = 5, *Ogdh^iKO^* n = 4; k and l, n = 4, each). **m,** RT-qPCR analysis of markers related to CMs maturation in Ctrl and *Ogdh^iKO^* mice, 3-4 weeks after termination of tamoxifen treatment for *Ogdh* deletion (n = 4, each). *36b4* served as reference gene. Error bar represents mean ± s.e.m. N-numbers refer to individual mice. Two-tailed, unpaired student t-tests for statistical analysis in a-e, g,i,m. *$P < 0.05$, **$P < 0.01$, ***$P < 0.001$, ****$P < 0.0001$.

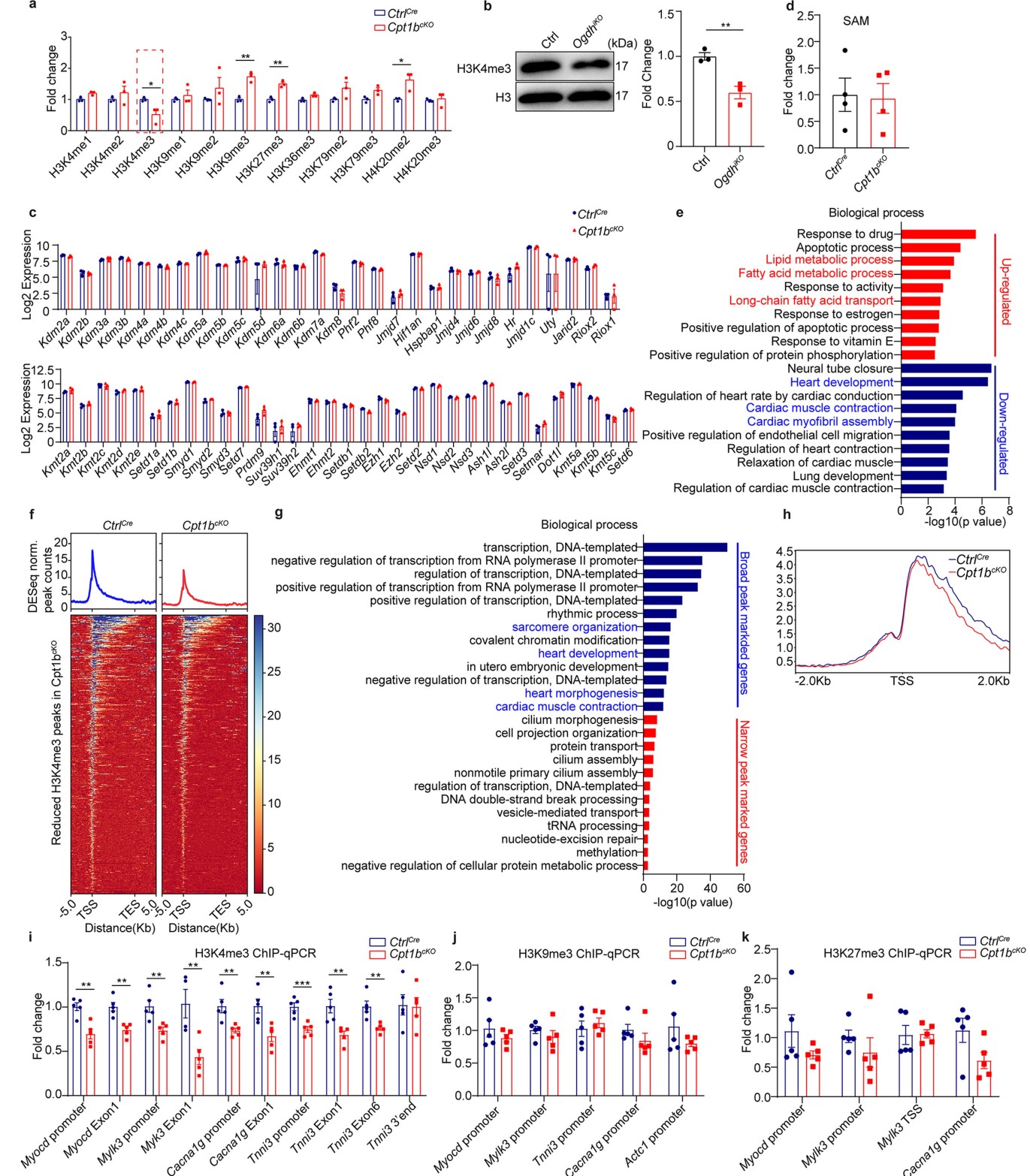

**Extended Data Fig. 8** | See next page for caption.

**Extended Data Fig. 8 | Analysis of epigenetic and expression changes in CMs after inactivation of *Cpt1b* or *Ogdh* in adult mice. a**, Quantification of histone methylation modifications using FACS-isolated CM nuclei from 10-weeks-old *Ctrl^Cre* and *Cpt1b^cKO* mice (n = 3, each). **b**, Western blot analysis of H3K4me3 in CMs from Ctrl and *Ogdh^iKO* mice, 3-4 weeks after termination of tamoxifen treatment for *Ogdh* deletion (n = 3, each). H3 was used as loading control. For gel source data, see Supplementary Information. **c**, Expression levels of selected *Jmjc* domain-containing histone demethylase (upper panel) and histone methyltransferase (lower panel) genes based on log2-transformation of DESeq-normalized counts (n = 3, each). **d**, Quantification of SAM concentrations in adult CMs from 10-weeks-old *Ctrl^Cre* and *Cpt1b^cKO* mice (n = 4, each). **e**, GO-terms of genes with differentially present H3K4me3 peaks in *Cpt1b^cKO* CMs. Enriched biological processes for genes with increased peaks are represented in red and for genes with reduced peaks in blue. **f**, Heat map of genes with reduced H3K4me3 ChIP-seq signals on promoters normalized for reads mapped in peaks. Corresponding coverage plots are shown in the top panels. **g**, Biological processes enriched for genes containing broad and narrow H3K4me3 peaks. Enriched GO-terms for genes with broad H3K4me3 peaks are shown in blue and for genes with narrow H3K4me3 peaks in red. **h**, Coverage plot of H3K4me3 peaks of genes with an overlap as shown in (Fig. 4d) between reduced expression and reduced H3K4me3 deposition in *Cpt1b^cKO* compared to *Ctrl^Cre* CMs. **i-k**, ChIP-qPCR analysis of H3K4me3, H3K9me3 and H3K27me3 enrichment in genes related to cardiac maturation using FACS-isolated adult CMs nuclei from 10-weeks-old *Ctrl^Cre* and *Cpt1b^cKO* mice (i, *Mylk3 Exon1* Ctrl n = 4, others n = 5; j and k, n = 5, each). Error bar represents mean ± s.e.m. N-numbers refer to individual mice. Two-tailed, unpaired student t-tests for statistical analysis in a-b, d, i-k. Wald-test with Benjamini–Hochberg correction in c. *$P < 0.05$, **$P < 0.01$, ***$P < 0.001$.

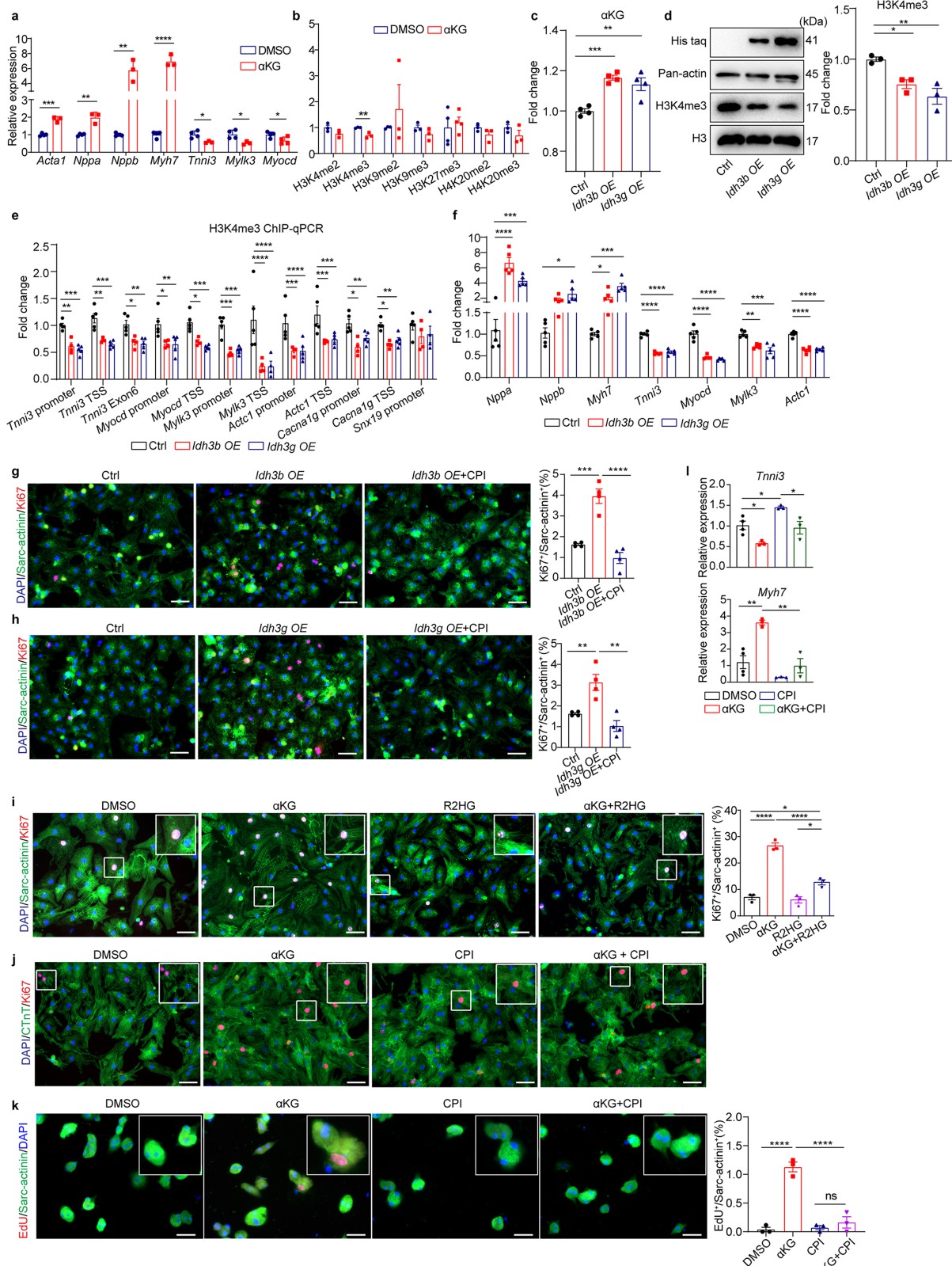

**Extended Data Fig. 9** | See next page for caption.

**Extended Data Fig. 9 | αKG stimulates proliferation of CMs. a**, RT-qPCR analysis of P0-1 neonatal CMs treated with DMSO or αKG (*Acta1, Nppa, Nppb, Myh7, Tnni3, Mylk3*, n = 4 in DMSO and n = 3 in αKG; *Myocd*, n = 5 in DMSO and n = 4 in αKG). *36b4* served as reference gene. **b**, Quantification of histone methylation levels in P0-1 neonatal CMs after 3-days in presence of DMSO or αKG (H3K27me3, n = 4; others n = 3, each). **c**, Quantification of αKG levels in P0-1 CMs transduced with *Idh3b* and *Idh3g* lentiviruses (n = 4, each). **d**, Western blot analysis of His-tagged IDH3B, IDH3G and H3K4me3 in neonatal CMs after transduction with *Idh3b* and *Idh3g* lentiviruses. Quantification of H3K4me3 levels is shown in the right panels (n = 3, each). H3 and pan-actin were used as internal controls. For gel source data, see Supplementary Information. **e**, ChIP-qPCR showing reduced enrichment of H3K4me3 in key cardiac maturation genes with broad but not with narrow H3K4me3 peaks (*Snx19*) in P0-1 CMs after transduction of *Idh3b* and *Idh3g* (Ctrl n = 5, *Idh3b*, n = 4; *Idh3*g, *Mylk3 TSS* and *Snx19 Promoter* n = 4, others n = 5). **f**, RT-qPCR analysis of cardiac maturation markers in P0-1 CMs after transduction with *Idh3b* and *Idh3g* lentiviruses (n = 5, each). *36b4* served as reference gene. **g**, **h**, Immunofluorescence analysis of Ki67 in sarc-actinin⁺ P0-1 CMs, 4 days after transduction of *Idh3b* (g) or *Idh3g* (h) lentiviruses with and without CPI treatment (n = 4, each). Scale bar: 50 μm. **i**, Immunofluorescence analysis of Ki67 in sarc-actinin⁺ P0-1 CMs after 3-days-culture in the presence of DMSO, αKG, R2HG, and a combination of αKG and R2HG (n = 3, each). Quantification of Ki67⁺ sarc-actinin⁺ CMs is shown in the right panel. Scale bar: 50 μm. **j**, Immunofluorescence analysis of Ki67 in cTnT⁺ P0-1 CMs after 3 days in presence of DMSO, αKG, CPI, and a combination of αKG and CPI. Scale bar: 50 μm. **k**, EdU incorporation in adult CM treated for 96 h with DMSO, αKG, and CPI (n = 3). Enlarged images are labelled by white frames. Quantification of EdU⁺ sarc-actinin⁺ CMs is shown in the right panel. Scale bar: 50 μm. **l**, RT-qPCR analysis of genes in P0-1 CMs after 4 days in presence of αKG and CPI (DMSO, n = 4; αKG, CPI, αKG + CPI, n = 3). *36b4* served as reference gene. Error bars represent mean ± s.e.m. N-numbers refer to independent experiments. Two-tailed, unpaired student t-tests for statistical analysis in a-b. One-way ANOVA with Tukey tests for correction of multiple comparisons in c-i, k-l. *$P < 0.05$, **$P < 0.01$, ***$P < 0.001$, ****$P < 0.0001$.

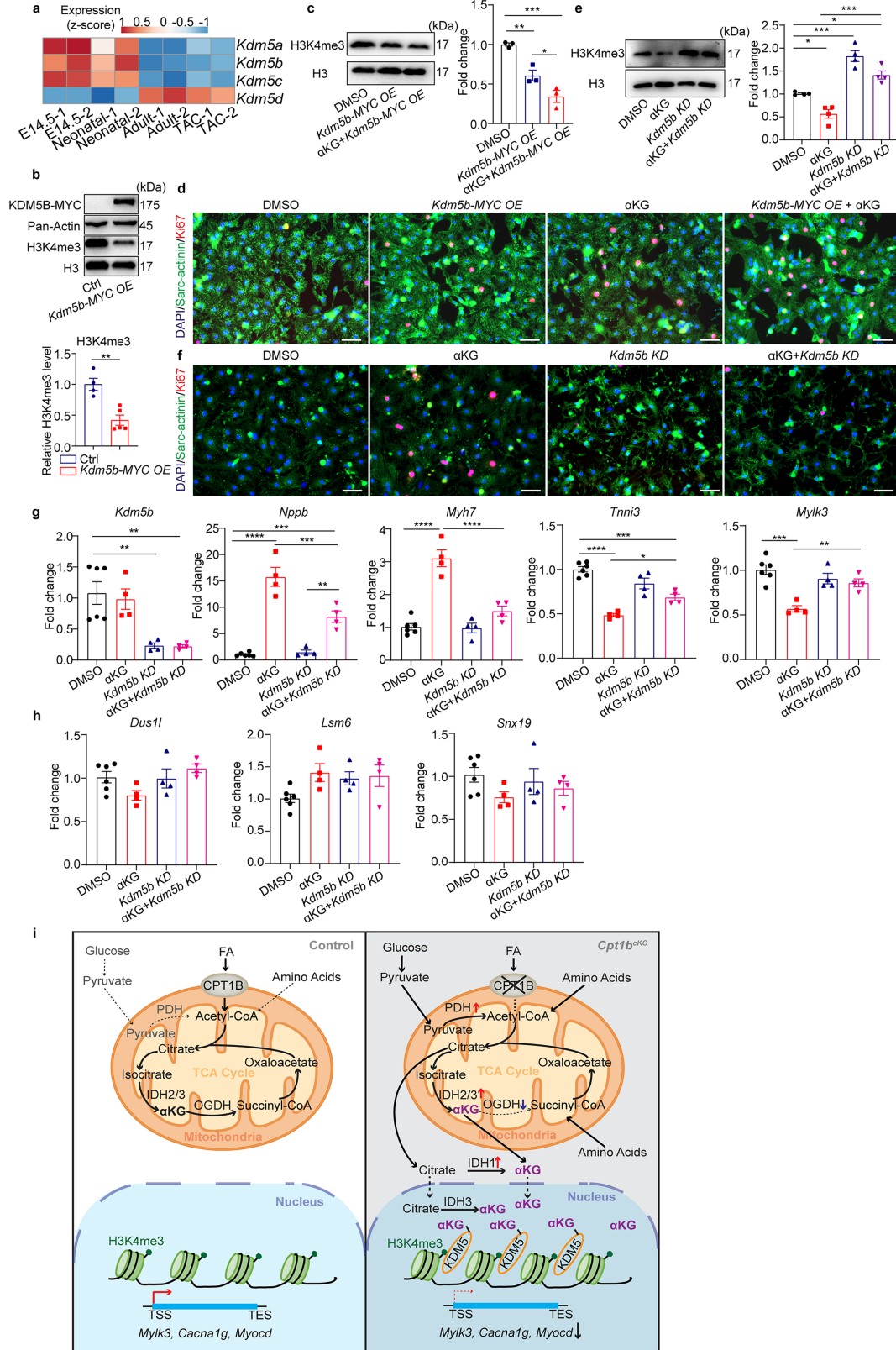

**Extended Data Fig. 10** | See next page for caption.

**Extended Data Fig. 10 | Effects of αKG on gene expression, H3K4me3 deposition, and proliferation of CMs are mediated via KDM5. a**, Heat map of gene expression of *Kdm5* family members at different stages of heart development and after transaortic constriction (TAC) based on z-scores of DESeq-normalized counts (RNAseq data are from GSE79883). **b**, Western blot analysis of myc-tagged KDM5B and H3K4me3 in neonatal CMs transduced with a *Kdm5b* lentivirus. Quantification of H3K4me3 levels is shown in the lower panel (Ctrl n = 4, *Kdm5b-myc* n = 5). H3 was used as loading control and pan-actin as sample processing control. **c**, Western blot analysis of H3K4me3 in P0-1 CMs transduced with a *Kdm5b* lentivirus with and without αKG treatment (n = 3, each). H3 was used as sample processing control. Quantification of H3K4me3 levels is shown in the right panel. **d**, Immunofluorescence analysis of Ki67 in sarc-actinin⁺ neonatal CMs (P0-1), 4-days after lentiviral transduction of *Kdm5b* with and without αKG treatment. Scale bar: 50 μm. **e**, Western blot analysis of H3K4me3 in P0-1 CMs after 4 days in presence of *Kdm5b* siRNA, αKG, and a combination of αKG and *Kdm5b* siRNA (n = 4, each). H3 was used as loading control.

Quantification of western blots is in the right panel. **f**, Immunofluorescence analysis of Ki67 in sarc-actinin⁺ P0-1 CMs, 4 days after treatment with *Kdm5b* siRNA with and without αKG. Scale bar: 50 μm. **g-h**, RT-qPCR analysis of genes related to cardiac maturation (g) and selected house-keeping genes (h) in P0-1 CMs after treatment with *Kdm5b* siRNA with and without αKG (DMSO n = 6; αKG, *Kdm5b* KD and αKG+*Kdm5b* KD, n = 4). *36b4* expression served as reference gene. **i**, Proposed model: inactivation of *Cpt1b* in CMs increases αKG levels due to increased expression of IDH and reduced enzymatic activity of OGDH, which stimulates KDM5 activity, leading to reduction of H3K4me3 deposition in cardiac maturation genes, enabling proliferation of CMs. The mechanisms underlying the transport of αKG and citrate (dashed lines) into the nucleus are not fully understood. Error bar represents mean ± s.e.m. N-numbers refer to independent experiments. Two-tailed, unpaired student t-tests for statistical analysis in b. One-way ANOVA with Tukey tests for correction of multiple comparisons in c,e, g-h. *$P < 0.05$, **$P < 0.01$, ***$P < 0.001$, ****$P < 0.0001$. For gel source data in b,c,e, see Supplementary Information.

| | |
|---|---|

# Reporting Summary

## Statistics

For all statistical analyses, confirm that the following items are present in the figure legend, table legend, main text, or Methods section.

| n/a | Confirmed | |
|---|---|---|
| ☐ | ☒ | The exact sample size (*n*) for each experimental group/condition, given as a discrete number and unit of measurement |
| ☐ | ☒ | A statement on whether measurements were taken from distinct samples or whether the same sample was measured repeatedly |
| ☐ | ☒ | The statistical test(s) used AND whether they are one- or two-sided<br>*Only common tests should be described solely by name; describe more complex techniques in the Methods section.* |
| ☒ | ☐ | A description of all covariates tested |
| ☐ | ☒ | A description of any assumptions or corrections, such as tests of normality and adjustment for multiple comparisons |
| ☐ | ☒ | A full description of the statistical parameters including central tendency (e.g. means) or other basic estimates (e.g. regression coefficient) AND variation (e.g. standard deviation) or associated estimates of uncertainty (e.g. confidence intervals) |
| ☐ | ☒ | For null hypothesis testing, the test statistic (e.g. *F*, *t*, *r*) with confidence intervals, effect sizes, degrees of freedom and *P* value noted<br>*Give P values as exact values whenever suitable.* |
| ☒ | ☐ | For Bayesian analysis, information on the choice of priors and Markov chain Monte Carlo settings |
| ☒ | ☐ | For hierarchical and complex designs, identification of the appropriate level for tests and full reporting of outcomes |
| ☒ | ☐ | Estimates of effect sizes (e.g. Cohen's *d*, Pearson's *r*), indicating how they were calculated |

*Our web collection on statistics for biologists contains articles on many of the points above.*

## Software and code

Policy information about availability of computer code

| Data collection | ZEN 2 Imaging Software (ZEISS Software), ChemiDoc Imaging Systems (Biorad), Medis Imaging Systems, Leica SP8 - software LAS X 3.5.7.23225, FACS Aria TM III (BD Biosciences) - BD FACS Diva v8 Software, All the sequencing data were collected using Illumina Nextseq500 platform, Agilent Seahorse XFe96 Analyzer |
|---|---|
| Data analysis | Prism 9 software package, FastQC 0.11.8 (http://www.bioinformatics.babraham.ac.uk/projects/fastqc), Trimmomatic version >=0.36 (http://www.usadellab.org/cms/?page=trimmomatic), STAR >=2.5.4b, DESeq2 version >=1.14.0, Picard 2.21.7 (https://github.com/broadinstitute/picard/releases/tag/2.21.7), MUSIC peakcaller (version of Dec. 2015), bigWigAverageOverBed (UCSC Toolkit), IGV 2.3.52<br>Codes used in the study been deposited at github and are available under 'https://github.com/loosolab/Li_et_al_2023_heart_regeneration' (DOI: 10.5281/zenodo.7828994) |

For manuscripts utilizing custom algorithms or software that are central to the research but not yet described in published literature, software must be made available to editors and reviewers. We strongly encourage code deposition in a community repository (e.g. GitHub). See the Nature Portfolio guidelines for submitting code & software for further information.

## Data

Policy information about availability of data

All manuscripts must include a data availability statement. This statement should provide the following information, where applicable:
- Accession codes, unique identifiers, or web links for publicly available datasets
- A description of any restrictions on data availability
- For clinical datasets or third party data, please ensure that the statement adheres to our policy

Data have been deposited in public data bases. The sequencing reads were aligned versus mouse genome version mm10 (GRCm38.p5). Sequencing data are available at https://www.ncbi.nlm.nih.gov/geo/ under the accession number GSE172415 (https://www.ncbi.nlm.nih.gov/geo/query/acc.cgi?acc=GSE172415) and include results from ChIP-seq and RNA-seq experiments. RNA-seq data of neonatal CMs treated with DMSO and aKG have the accession number: GSE217188 (https://www.ncbi.nlm.nih.gov/geo/query/acc.cgi?acc=GSE217188).

## Research involving human participants, their data, or biological material

Policy information about studies with human participants or human data. See also policy information about sex, gender (identity/presentation), and sexual orientation and race, ethnicity and racism.

| | |
|---|---|
| Reporting on sex and gender | Not applicable |
| Reporting on race, ethnicity, or other socially relevant groupings | Not applicable |
| Population characteristics | Not applicable |
| Recruitment | Not applicable |
| Ethics oversight | Not applicable |

Note that full information on the approval of the study protocol must also be provided in the manuscript.

# Field-specific reporting

Please select the one below that is the best fit for your research. If you are not sure, read the appropriate sections before making your selection.

☒ Life sciences          ☐ Behavioural & social sciences          ☐ Ecological, evolutionary & environmental sciences

For a reference copy of the document with all sections, see nature.com/documents/nr-reporting-summary-flat.pdf

# Life sciences study design

All studies must disclose on these points even when the disclosure is negative.

| | |
|---|---|
| Sample size | Sample size were determined based on established practice and applicable standards. We opted for sample sizes which are commonly used sample size in the field. For in vivo studies, a minimum of three biological replicates were analyzed. For In vitro studies, experiments in which data were not quantified were performed with at least two replicates. Each experiment in which data were quantified was performed with at least 3 replicates. |
| Data exclusions | For both the in vivo and in vitro experiments, all the attempts were successfully and all the data were included during data processing. |
| Replication | All in vivo studies were performed with indicated number of animals. The in vitro study were performed at least 3 times independently and each replicate was successful. Sample sizes, statistical analyses and significance levels are all indicated in the figure legends or the method part. |
| Randomization | Animals were assigned to different groups according to genotypes. The genotype of animals from which individual samples was not known and experiments were performed in a blinded pattern. After data collection, individual genotypes were revealed and the animals were assigned to separate groups for further statistical analysis. For in vitro experiments, the samples were allocated into groups randomly. |
| Blinding | For in vivo study, all animals were numbered and experiments were performed in a blinded pattern. After data collection, genotypes were revealed and animals assigned to groups based on genotype for data analysis. In vitro experiments were not blinded during data collection or analysis since we know the treatment of each group before data collection. Positive controls, negative controls and samples were analyzed in exactly the same manner. |

# Reporting for specific materials, systems and methods

We require information from authors about some types of materials, experimental systems and methods used in many studies. Here, indicate whether each material, system or method listed is relevant to your study. If you are not sure if a list item applies to your research, read the appropriate section before selecting a response.

## Materials & experimental systems

| n/a | Involved in the study |
|-----|----------------------|
| ☐ | ☒ Antibodies |
| ☐ | ☒ Eukaryotic cell lines |
| ☒ | ☐ Palaeontology and archaeology |
| ☐ | ☒ Animals and other organisms |
| ☒ | ☐ Clinical data |
| ☒ | ☐ Dual use research of concern |
| ☒ | ☐ Plants |

## Methods

| n/a | Involved in the study |
|-----|----------------------|
| ☐ | ☒ ChIP-seq |
| ☐ | ☒ Flow cytometry |
| ☒ | ☐ MRI-based neuroimaging |

## Antibodies

**Antibodies used**

α-Actinin (Sarcomere) 1:500 (Sigma-Aldrich A7811); Ki67 1:500 (Abcam ab15580); cTnT-FITC 1:1000 (Abcam ab105439); pH3 1:500 (Millipore 06-570); Ccne1 1:1000 (Abcam ab7959); Aurora B 1:200(Abcam Ab2254); Cpt1b 1:2000(Proteintech 22170-1-AP); pan-actin 1:5000(Cell Signaling 4968); WGA Alexa FluorTM 488 1:500 (ThermoFisher Scienticfic W11261); PCM1 1:500 (Sigma-Aldrich HPA023374-100UL); PDH1a 1:2000(Proteintech 18068-1-AP); PDK4 1:1000(Proteintech 12949-1-AP); ACSS1 1:1000(Proteintech 17138-1-AP); ACSS2 1:1000 (GeneTex GTX30020); ACL 1:1000 (Proteintech,18068-1-AP); IDH1 1:2000 (Biorbyt orb135710); IDH2 1:2000 (ThermoFisher Scienticfic MA5-17271); IDH3a 1:2000(Abcam ab58641); OGDH 1:2000(Sigma-Aldrich HPA020347-100UL); DLST 1:1000(Cell Signaling 5556); DLD 1:1000(ThermoFisher Scienticfic PA5-27367); CS 1:1000(Proteintech, 16131-1-AP); MDH2 1:1000 (Proteintech, 15462-1-AP); SDHC 1:1000 (Proteintech, 14575-1-AP); FH 1:1000 (Proteintech,11375-1-AP); SDHB 1:1000 (Proteintech, 10620-1-AP); SDHA 1:1000 (Cell Signaling, 6922); ACO2 1:1000 (Cell Signaling, 5839); Phospho-Histone H2A.X (Ser139) 1:500 (Cell Signaling, 2577); Detyrosinated alpha Tubulin 1:500 (Abcam, ab48389); H3K4me1 1:1000 (Abcam ab8895); H3K4me2 1:1000 (Active Motif 39141); H3K4me3 1:1000 (Millipore 07-473); H3K9me1 1:1000 (Abcam ab8896); H3K9me2 1:1000 (Abcam ab12200); H3K9me3 1:1000 (Abcam ab8898); H3K27me3 1:1000 (Millipore 07-449); H3K36me3 1:1000 (Abcam ab9050); H3K79me2 1:1000 (Millipore 04-835); H3K79me3 1:1000 (Abcam ab2621); H4K20me1 1:1000 (Abcam ab9051); H4K20me2 1:1000 (Abcam ab9052); H4K20me3 1:1000 (Abcam ab9053); H3 1:5000 (Abcam ab1791); H3K4me3 1:100 (Diagenode C15410003-50); KDM5B 1:50 (Cell Signaling 15327S); Goat anti-Mouse IgG (H+L) Alexa Fluor 488 1:1000 (ThermoFisher Scienticfic A28175); Goat anti-Rabbit IgG (H+L) Alexa Fluor 594 1:1000 (ThermoFisher Scienticfic A-11037).

**Validation**

α-Actinin (Sarcomere) has been validated with mouse sample by Luo et al, Circ Res, 2017. Ki67, pH3, AuroraB, PCM1 was validated by Chen et al, Science , 2021. Ccne1 has been validated with mouse sample in WB by Cang Y, et al, Cell, 2006. Cpt1b was validated with mouse sample in WB by Simcox, et al, Cell Metab, 2017. Pan-actin was validated with mouse sample in WB by Hillege, et al, Elife, 2022. WGA was validated in IF with mouse heart tissue by Chatterjee et al, Cell, 2016. OGDH, DLD, DLST were validated with mouse sample in WB by Andrade J, et al, Nat Cell Biol, 2021. PDH1a, PDK4, IDH1 were validated with mouse kidney tissue lysate via information on manufacturer's website. ACSS1 was validated in WB with mouse sample by Odera JO, et al, Biochem J, 2020. ACSS2 was validated with mouse liver lysate with information from manufacturer's website. IDH2, IDH3a was validated with mouse brain extracts with information from manufacturer's website. CS was validated with mouse sample in WB by Xu Yj, et al, EMBO J, 2021. MDH2 antibody was validated with mouse sample by Sinha T, et al, iScience, 2020. SDHC antibody was validated with mouse sample in WB by Kauppila JH, et al, Nucleic Acids Res, 2018. FH antibody was validated with mouse sample in WB by Yuan MH, et al, Aging Dis, 2022. SDHB antibody was validated with mouse sample in Ding M, et al, Diabetes, 2022. ACO2 antibody was validated with mouse sample via Silvia V, et al, Cell Rep, 2021. Phospho-Histone H2A.X (Ser139) was validated in IF by Ratnaparkhe M, et al, Nat Commun, 2018. Detyrosinated alpha Tubulin was validated with mouse heart sample by Schuldt M, et al, Front Cardiovasc Med, 2021. All histone antibody, includes H3K4me1, H3K4me2, H3K4me3, H3K27me3, H3K79me2 and H4K20me1, H3K9me1, H3K9me2, H3K9me3, H3K36me3 and H3K79me3, H4K20me2 and H4K20me3, H3 was validated Interactive Database for the Assessment of Histone Antibody Specificity. KDM5B was validated with extracts from K-562cell lines in ChIP experiment.

## Eukaryotic cell lines

Policy information about cell lines and Sex and Gender in Research

**Cell line source(s)**

HEK293T from ATCC

**Authentication**

Authenticated human cell lines were obtained from ATCC and maintained as instructed. The cell was checked for morphology for cell authentication. .

**Mycoplasma contamination**

mycoplasma free

**Commonly misidentified lines**
(See ICLAC register)

No commonly misidentified cell lines were used

# Animals and other research organisms

Policy information about studies involving animals; ARRIVE guidelines recommended for reporting animal research, and Sex and Gender in Research

| | |
|---|---|
| Laboratory animals | All mice were maintained on a C57BL/6 background, and littermates were used as controls in all  experiments. All mice were maintained in individually ventilated cages, at 22.5 °C ± 1 °C and a relative humidity of 50% ± 5% with controlled illumination (12 h dark/light cycle). Mice were given ad libitum access to food and water. All experiments were performed on balanced cohorts of male and female mice. In experiments, in which genes were inactivated by Cre recombinase-mediated recombination, corresponding Cre recombinase-expressing strains without the floxed target genes were always used as negative controls. αMHC-MCM control mice were subjected to the same tamoxifen treatment as in the actual gene inactivation experiment |
| Wild animals | Studies did not involve wild animals. |
| Reporting on sex | Observed results did not differ between male and female mice. No gender-specific experiments were performed. |
| Field-collected samples | Studies did not involve samples collected in the field. |
| Ethics oversight | All animal experiments were done in accordance with the Guide for the Care and Use of Laboratory Animals published by the US National Institutes of Health (NIH Publication No. 85-23, revised 1996) and were approved by the responsible Committee for Animal Rights Protection of the State of Hessen (Regierungspraesidium Darmstadt, Wilhelminenstr. 1-3, 64283 Darmstadt, Germany) with the project number B2/1125, B2/1137, B2/1056 and B2/2034. |

Note that full information on the approval of the study protocol must also be provided in the manuscript.

# Plants

| | |
|---|---|
| Seed stocks | Not applicable |
| Novel plant genotypes | Not applicable |
| Authentication | Not applicable |

# ChIP-seq

## Data deposition

☒ Confirm that both raw and final processed data have been deposited in a public database such as GEO.

☒ Confirm that you have deposited or provided access to graph files (e.g. BED files) for the called peaks.

| | |
|---|---|
| Data access links *May remain private before publication.* | https://www.ncbi.nlm.nih.gov/geo/query/acc.cgi?acc=GSE172415 |
| Files in database submission | GSM5255555  Input-CtrlCre_1<br>GSM5255556  Input-CtrlCre_2<br>GSM5255557  Input-CtrlCre_3<br>GSM5255558  Input-Cpt1bcKO_1<br>GSM5255559  Input-Cpt1bcKO_2<br>GSM5255560  Input-Cpt1bcKO_3<br>GSM5255561  ChIP-CtrlCre_1<br>GSM5255562  ChIP-CtrlCre_2<br>GSM5255563  ChIP-CtrlCre_3<br>GSM5255564  ChIP-Cpt1bcKO_1<br>GSM5255565  ChIP-Cpt1bcKO_2<br>GSM5255566  ChIP-Cpt1bcKO_3 |
| Genome browser session (e.g. UCSC) | N/A |

## Methodology

| | |
|---|---|
| Replicates | Three biological replicates for H3K4me3 ChIP-seq. |
| Sequencing depth | Input-CtrlCre_1 27M 80% aligned<br>Input-CtrlCre_2 20M 79% aligned<br>Input-CtrlCre_3 32M 79% aligned<br>Input-Cpt1bcKO_1 29M 79% aligned<br>Input-Cpt1bcKO_2 35M 80% aligned<br>Input-Cpt1bcKO_3 25M 78% aligned<br>ChIP-CtrlCre_1 26M 79% aligned |

ChIP-CtrlCre_2 25M 79% aligned
ChIP-CtrlCre_3 26M 78% aligned
ChIP-Cpt1bcKO_1 31M 80% aligned
ChIP-Cpt1bcKO_2 30M 81% aligned
ChIP-Cpt1bcKO_3 31M 79% aligned
Single end.

| Antibodies | H3K4me3 (Diagenode C15410003-50) |
|---|---|

**Peak calling parameters**

The MUSIC peakcaller (version of Dec. 2015) was employed in punctate mode to identify enriched regions when comparing the respective ChIP to input samples. The MUSIC FDR was set to 0.2. Peaks overlapping ENCODE blacklisted regions (known misassemblies, satellite repeats) were excluded. In order to compare peaks in different samples for assessment of reproducibility, the resulting lists of significant peaks were overlapped and unified to represent identical regions. Sample counts for union peaks were produced using bigWigAverageOverBed (UCSC Toolkit) and normalized with DESeq2 1.26.0 to compensate for differences in sequencing depth, library composition, and efficiency. Peaks were annotated with the promoter of the nearest gene in range (TSS +- 5000 nt) using reference data of GENCODE vM15. Peaks were classified to be significantly differentially expressed with P-Value < 0.05 as produced by DESeq2. Peaks were divided into 3 groups based on peak length (Broad: >75th %, Medium: 75th – 25th %, Narrow: <25th %).

**Data quality**

id #          peaks raw
ChIP-Cre_1   16043
ChIP-Cre_2   10913
ChIP-Cre_3   16138
ChIP-KO_1    13312
ChIP-KO_2    16303
ChIP-KO_3    15843

**Software**

FastQC 0.11.8 (http://www.bioinformatics.babraham.ac.uk/projects/fastqc), Trimmomatic version >=0.36 (http://www.usadellab.org/cms/?page=trimmomatic), STAR >=2.5.4b, DESeq2 version >=1.14.0, Picard 2.21.7 (https://github.com/broadinstitute/picard/releases/tag/2.21.7), MUSIC peakcaller (version of Dec. 2015), bigWigAverageOverBed (UCSC Toolkit), IGV 2.3.52

# Flow Cytometry

## Plots

Confirm that:

☒ The axis labels state the marker and fluorochrome used (e.g. CD4-FITC).

☒ The axis scales are clearly visible. Include numbers along axes only for bottom left plot of group (a 'group' is an analysis of identical markers).

☒ All plots are contour plots with outliers or pseudocolor plots.

☒ A numerical value for number of cells or percentage (with statistics) is provided.

## Methodology

**Sample preparation**

Ventricle were washed with ice-cold PBS after dissection and snap frozen in liquid N2. For cardiac nuclei isolation, the frozen ventricle was thawed in 3ml lysis buffer (5 mM CaCl2, 3 mM MgAc, 2 mM EDTA, 0.5 mM EGTA, and 10 mM Tris-HCl, pH 8) in M tube (Miltenyi Biotec) and homogenized by the gentleMACS Dissociator (Miltenyi Biotec) following the manufacturer's protocol (protein_01). The resultant homogenate was mixed with lysis buffer containing 0.4% Triton X-100, incubated on ice for 10 min, and subsequently filtered through 40um cell strainer (BD Bioscience). The flow-through was subjected to centrifugation at 1000 g for 5min at 4°C to harvest nuclei. Nuclei were further purified by centrifugation at 1000g for 5 min at 4°C through a 1M sucrose cushion (3 mM MgAc, 10 mM Tris-HCl, pH8) and then stained with a PCM1 antibody in nuclei stain buffer (DPBS, 1% BSA, 0.2% Igepal CA-630, 1 mM EDTA). DNA was stained by DAPI before FACS sorting.

| Instrument | FACS sorting was done using a FACSAriaTM III (BD Biosciences). |
|---|---|
| Software | BD FACS Diva v8 software |

**Cell population abundance**

Sorted cells were reanalyzed to assess purity. A 85% purity was achieved. The sorted PCM1 positive cardiac nuclei were further checked under immunofluorescence microscope to confirm the purity.

**Gating strategy**

The detailed gating strategy was shown in Supplementary Figure 7. In brief, the samples were firstly cleaned utilizing the SSC-A and DAPI-A to remove the DAPI-negative cell debris, the selected population was labeled as P1. The P1 population was further cleaned with cutoffs from DAPI-A and FSC-W, and the positive population was labeled as P2. The P2 population was then selected based on SSC-A and DAPI-W and the positive population was labeled as P3. Within the P3 population, the final population was selected based on the PCM1 signal intensity, the positive population was labeled and shown in Supplementary Figure 7. The PCM1 positive nuclei were further characterized into sub-clusters based on DNA content reflected by DAPI signal intensity.

☒ Tick this box to confirm that a figure exemplifying the gating strategy is provided in the Supplementary Information.

