## [Peer Review File · Nature]

Manuscript Title: Inhibition of fatty acid oxidation enables heart regeneration in adult mice

Editorial Notes:

Redactions – unpublished data

Reviewer Comments & Author Rebuttals

Reviewer Reports on the Initial Version:

Referees' comments:

Referee #1 (Remarks to the Author):

This study reports that inhibition of fatty acid oxidation (FAO) in the mouse heart by deleting CTP1b (KO) results in cardiomyocyte proliferation and cardiac hypertrophy but no change in cardiac function. Moreover, adult KO hearts are protected from ischemia/reperfusion injury and demonstrate robust regeneration capacity. Mechanistically, the study finds that alpha-KG level is markedly increased in KO, and subsequently shows that high alpha-KG leads to reduced H3K4me3. The study further proposes that increased alpha-KG activated KDM5B and through reduced H3K4me3 to promote expression of immature cardiac genes and proliferation.

The study results are consistent with prior findings that decreased FAO and increased glucose metabolism promote cardiomyocyte proliferation. Different from the previous studies, results here identify activation of KDM5 by alpha-KG and subsequent reduction of H3K4me3 as novel mechanisms for cardiomyocytes proliferation. This part of the study is interesting as it moves forward from the generally held ROS-DDR hypothesis and proposes a specific link between metabolism and gene regulation. There are, however, several significant deficiencies of the study.

1. Metabolic assessment is inadequate; some speculations are based on incorrect assumptions or interpretations.

Mechanism(s) leading to alpha-KG accumulation was speculative. The authors proposed that alpha-KG production exceeded clearance in KO hearts resulting in accumulation. Protein levels of ICDH and OGDH, enzymes responsible for alpha-KG production and consumption respectively, both increased in KO. However, enzymatic activity of OGDH was lower in KO. Thus, it was speculated that increased production and decreased clearance caused alpha-KG accumulation. There are several problems in this interpretation. 1) metabolites downstream of OGDH, succinyl-CoA, was also higher while other TCA cycle metabolites did not change in KO, not supporting OGDH was the bottleneck; 2) No explanation why OGDH activity was lower despite higher protein amount. It was speculated that OGDH activity was inhibited by higher BCAA catabolites in KO. This does not make sense since OGDH activity assay was performed in vitro which would not reflect the influence by metabolites in vivo unless there was co-valent modification. Further study of OGDH is required. 3) It was not shown whether only protein levels of ICDH and OGDH changed or other enzymes in the TCA cycle also changed. If all enzymes in TCA cycle are upregulated in KO, why only alpha-KG accumulates?

Fatty acids are the primary fuel of the heart. Deletion of CPT1b deprives mitochondria of fatty acid supplies but the KO heart presents normal contractile function. This suggested that inhibition of FAO minimally affected energy supply in KO. Metabolic adaptation to sustain energy homeostasis may be

key to the reprogramming observed here, including the intriguing accumulation of alpha-KG. However, metabolic characterization was done poorly. The authors mentioned a variety of mechanisms, e.g. glycolysis, amino acids metabolism, pyruvate oxidation, but seemed uncertain which ones are contributing. Indeed, there was no evidence of increased glycolysis although the authors stressed it in the discussion. Also unclear how changes in amino acid metabolism described in the results could contribute to energy homeostasis. Ultimately, measurement of metabolic flux is required to sort out the reprogramming of fuel metabolism.

2. The I/R experiment was problematic.

Critical information, such as area at risk (AAR) and % infarct area out of AAR were missing. They are required to provide assurance that I/R caused comparable insult in all groups. Histology showed very small infarct in control I/R hearts. Have the authors aimed to produce significant I/R injury as described in the literature?

Many studies have shown that upregulation of HIF1a and/or switch from FAO to glucose metabolism prevent ischemia/reperfusion injury. This part of the finding is confirmatory but not novel. However, it would be interesting to ask whether a proliferating heart that at the same time unable to utilize FAO for energy supply can maintain contractile function after I/R insult. In an I/R heart with sufficient myocardial injury, the remaining myocardium is subjected to higher workload. Previous studies showed that CPT1b^{+/-} hearts could not tolerate increased workload. How does cardiac function and survival of CPT1b KO look immediately after I/R? It has also been suggested that promoting cardiomyocytes re-entry into cell cycle would compromise contractile function in adult hearts. It will be very significant to determine whether metabolic reprogramming would remedy or exacerbate the situation.

3. Missing links connecting alpha-KG accumulation, epigenetics change and proliferating cardiomyocytes.

The authors made compelling observations that supplementation of cell permeable alpha-KG or manipulating KDM5 activity could alter H3K4me3 and related gene expression. These actions occur outside mitochondria. How does inhibition of mitochondrial FAO lead to increased alpha-KG in the cytosol/nucleus? Metabolomics or enzyme activity assays did not distinguish the two subcellular compartments.

The authors showed that iKO in adult mice could trigger cardiomyocyte proliferation. Experiments manipulating alpha-KG or KDM5 were performed in neonatal cardiomyocytes. Will similar changes be observed in adult cardiomyocytes? This is needed to address the question whether metabolic reprogramming prevents or reverse cell cycle arrest.

The study showed that reduction of H3K4me3 promoted the expression of immature cardiac gene. What mechanism(s) suppress the mature cardiac genes in order for adult cardiomyocytes become proliferating? Did those mechanisms also reverse the metabolic maturation, e.g. mitochondrial biogenesis or oxidative metabolism?

4. Other points:

Some misconceptions of metabolism in the text should be corrected. a) the heart switches from

glycolysis to FAO in perinatal period not one week after birth (first paragraph in Discussion). b) Oxidative metabolism of any substrates, not just FAO, generates ROS (Introduction). The fact that CPT1b KO cardiomyocytes continue oxidative metabolism and at the same time proliferate questions the role of ROS induced DDR as a major mechanism of cell cycle arrest. c) Inhibition of FAO does not increase glycolysis only, it has to increase oxidation of other substrates to maintain ATP supply of the heart (first paragraph of Discussion).

Concerns of statistical analysis:

Many experiments have n=3. How was statistics analysis performed?

For measurements taken from individual cardiomyocyte, results from all cardiomyocytes isolated from one heart should be considered n=1. Counting each cardiomyocyte from the same heart as an independent n=1 artificially inflates statistical power.

Referee #2 (Remarks to the Author):

In this manuscript from the Braun group, the authors set out to examine the role of myocardial fatty acid oxidation in regulation of heart regeneration through epigenetic mechanisms mediated by H3K3me3.

The authors focused on Cpt1b, one of the rate limiting enzymes in fatty acid oxidation. They show that loss of this enzymes enhances myocyte survival and proliferation in neonatal and adult mice. Interestingly, they show that α -ketoglutarate levels are increased, in the KO hearts, and thus they examine the α -ketoglutarate dependent lysine demethylase KDM5. They show that activation of KDM5 demethylates numerous H3K3 me3 domains in cardiomyocyte maturation genes, thereby maintaining cardiomyocytes in a less differentiated and proliferative state.

Overall this is an interesting and well written manuscript. The data quality is high, and the authors use state of the art techniques to examine their hypothesis. The central premise of this manuscript is that although the role of myocyte maturation and the role of fatty acid oxidation in regulation of heart regeneration is known, these two pathways have not been linked, and this is an important advance in the field. I do, however, have several comments:

Major:

1) It is interesting that KDM5 alone can have these effects. How does it function though in the absence of increase aKG? The authors conclude from the first set of studies that Cpt1b mediated increase in aKG the reason for the KDM5 activity. Do aKG levels increase with KDM5 gain of function? And if so, how? This is a central point in the manuscript that needs to be clarified experimentally.

2) It is interesting that in Figure S3, FA metabolism is upregulated at the transcript level. This raises an important concern as to whether there is a true shift in substrate metabolism. As such, flux analysis studies are crucial to confirm the metabolic consequences of Cpt1b deletion.

3) Several previous reports indicate that inactivation of FA oxidation can result in cardiomyocyte

proliferation citing various mechanisms including DNA damage, and as the authors state in their text, other redox modifying interventions have similar effects. It would be important for the authors to delineate whether there is Cpt1b-specific effects, or are these effects shared by all FA and ROS modulating interventions? For example, are ROS changed in the Cpt1b hearts? Is DNA damage changed? And if so, would artificial increase in ROS reverse the observed phenotype?

4) Along the lines of point #1, recent reports by the Mahmoud group indicate that manipulation of cardiac metabolites by exogenously administering these metabolites can induce heart regeneration. If the author's claim that the effect of Cpt1b is dependent on the increase in aKG, does administration of aKG affect KDM5 levels? Or cardiomyocyte maturation and proliferation? Certainly, this would be a critical therapeutic angle.

5) The dichotomy of increased cell size and proliferation in the Cpt1b ko hearts is very interesting. It is typically viewed that immature cardiomyocytes are smaller, as indicated by numerous reports. How do the authors reconcile that difference? Is it possible to be immature, proliferative, and hypertrophied at the same time? Can the authors provide some examples?

6) It is very interesting that the drug etomoxir can induce cardiomyocyte proliferation in vitro. From a therapeutic standpoint, the authors should test this in vivo. Does etomoxir induce cardiomyocyte proliferation in vivo? Can it induce regeneration? This needs to be tested.

7) The authors use Ki67 for proliferation studies in vivo. However, Ki67 is not an accurate indicator of mitosis. The standard in the field now includes pH3 and aurora B kinase +/- stereology. Some of these should be shown especially for the in vivo studies

8) In Figure 4 the KO heart size in panel A is massive, even compared to the KO heart size in panel B despite being a younger mouse. Is this to scale? Regardless the use of the Cre model rather than the MCM model here is not appropriate. If in fact this constitutes a regenerative effect, deletion should occur after MI, not before. This will allow for assessment of a pro regenerative effect irrespective of a potential protective effect.

Minor:

1) In figure 1D, Y axis probably should be per HPF? Or plate? These are not sections

2) The cardiomyocyte proliferation images in vivo are too small and not clear. Please provide high resolution images of mitotic cardiomyocytes where applicable.

3) I am not convinced that there is almost doubling of cardiomyocyte number as shown in figure 2L. The heart size does not reflect that. There is a moderate increase in heart size accompanied by hypertrophy (which could certainly explain the increase in heart size). An additional doubling of cardiomyocyte number would likely result in a far greater increase in heart size. Certainly, my comments are arbitrary, and I do realize that getting an accurate numerical correlation between heart size and cell number is not possible. However, perhaps the authors can show an additional parameter like stereology which would greatly enhance confidence in the data.

4) In figure 3, it is interesting that Hif1 and mTOR mRNA are upregulated. This needs to be discussed in the discussion section as a possible explanation for the observed hypertrophy and proliferation phenotype.

Referee #3 (Remarks to the Author):

In the study title “Enhanced H3K4me3 methylation by inhibition of fatty acid oxidation enables heart regeneration in adult mice”, the authors found that that inhibition of FAO metabolic pathway by the deletion of Cpt1b in cardiomyocytes promoted growth and promotes heart regeneration after ischemia/reperfusion damage. Interestingly, the authors found that Cpt1b deletion led to increased cellular concentration of alpha-ketoglutarate which promotes H3K4me3 demethylation by the KDM5 family of enzymes. Cpt1b deletion led to a decrease specifically in broad H3K4me3 on a cohort of genes involved in multiple cardiac processes resulting in decreased gene expression. Overall, this study provides a unique crosstalk between metabolism and chromatin during cardiac maturation.

This review focusses on the novelty and technical quality of the gene expression and chromatin aspects of this study, and connections to metabolism. The novelty of this study lies in the finding that increased α KG leads to specific increase in activity of the KDM5 family of demethylases and decreased H3K4me3 over cardiac cell identity genes. These are interesting mechanistic findings with important physiological impact. However, there are important issues and questions in the findings that need to be resolved, and functional experiments to solidify the mechanism of FAO metabolism on regulation of cardiac maturation genes, before further consideration of this manuscript for Nature.

Major Points:

1. One central argument is that α KG accumulation is due to elevated levels of IDH1, IDH2, and IDH3A proteins resulting in increased α KG production (Figure 5E, 5I). First, is this elevation of IDH proteins at the RNA level or only protein level? Second, importantly, to confirm this mechanism, authors should demonstrate that inhibition/depletion of IDH proteins prevents α KG accumulation in Cpt1bKO, and prevents the loss of H3K4me3 levels by Western and ChIPseq.
2. There is a major problem in the logic in focusing on the decrease in H3K4me3 in the Cpt1bKO. That is because equally or even more striking is the increase in repressive histone modifications H3K9me3, H3K27me3 (Fig 6A). Since both decreased H3K4me3 and increased H3K9me3/H3K27me3 cause lowered transcription, there is no strong justification to focus exclusively on H3K4me3 decrease—beyond the α KG increase and IDH increases. Hence, first, are SAM levels changed? SAM is a major metabolite and if it is increased, then this would provide increased cofactor for KMT enzymes for H3K9me3 and H3K27me3. Second, are the KMT enzymes for K9 and K27 increased? Third, does H3K27me3 increase by ChIPseq over the cardiac cell identity genes (which show decreased H3K4me3)? What about H3K9me3?
3. A third major finding is that the effects of Cpt1b mutations occurred more strongly on broad H3K4me3 peaks as compared to narrow. This is likely dependent on the localization of KDM proteins at broad vs narrow sites (Fig 7N) using ChIP-qPCR. To better demonstrate this key point, authors should perform genome-wide analysis of KDM5B to show this preferential pattern.
4. The authors indicate that the accumulation of α KG leads to increased expression of the cardiac proliferating genes through increased activity of KDM5. The authors introduce α KG into neonatal cardiomyocytes (CM) and show increase cardiomyocytes and relevant genes by RT-qPCR; to show specificity, the authors should carry out RNA-seq. In addition, the authors examine histone methylation by Western and see decrease in H3K4me3 but should also examine H3K27me3 for an increase (see above). The authors performed overexpression of KDM5B to promote cardiac

maturation and proliferation (Fig 7M). Although the authors used chemical inhibition of KDM5B (Fig 7H – 7K) the authors should also demonstrate that depletion/deletion of KDM5B prevents decreased H3K4me3 in Cpt1b KO and negates the effects on maturation and proliferation.

5. Indeed, continuing point 4, throughout the mechanistic sections of the manuscript, the authors treat isolated cardiomyocytes with various chemicals (e.g. α KG and enzyme inhibitors) and OE KDM5D. Because of off-target effects of inhibitors and chemicals, it is crucial to additionally target specific enzymes (IDHs and KDMs as well potentially KMTs for K27 and K9) using genetic KD or KO.

Additional Points:

1. To confirm the Cpt1b deletion is specific to cardiomyocytes, authors should examine other cardiac cell types to show that they maintain WT levels of Cpt1b.
2. In Figure 3J, the heat maps look nearly identical between the Cpt1bcKO and the Cpt1biKO. Could this be an accidental duplication? Further the heat maps are all at maximum increase or decrease. These results seem highly unlikely. The authors should expand the range of the Z-score.
3. Authors indicate increased levels in α KG in the Cpt1b KO is due to increased levels of IDH proteins. However, it is unclear if this is due to specific loss of Cpt1b or dysregulation of the FAO pathway. Authors should KO other enzymes within the FAO pathway to show α KG levels are increased.
4. Following point 3, authors should overexpressed IDH genes to prove the hypothesis that is indeed increased levels of these proteins that promotes high levels of α KG.
5. Authors mention there are no changes in most of the TCA metabolites measured. However, about half of them seem to be different between WT and Cpt1b KO, can the authors clarify how KO cells are producing energy?
6. Authors should show Figures 6B and 6C for the Cpt1b KO. This would support the effects that Cpt1b KO are targeted to H3K4me3 broad domains.
7. Authors should replace tracks for Cacna1g with TNi3 as functional data (Fig 7N, and S7N) focuses on TNi3.
8. Authors nicely demonstrate that the effects of α KG is blunted upon KDM inhibitor, however, it would be a good control to show this effect is not observed at genes that are changing but are independent of KDM5 (H3K4me narrow peak marked genes).
9. Authors should provide western blot images that better match the quantifications shown (Fig 6A, 7D and 7J).

Author Rebuttals to Initial Comments:

Referee #1 (Remarks to the Author):

1. *Metabolic assessment is inadequate; some speculations are based on incorrect assumptions or interpretations. Mechanism(s) leading to alpha-KG accumulation was speculative. The authors proposed that alpha-KG production exceeded clearance in KO hearts resulting in accumulation. Protein levels of ICDH and OGDH, enzymes responsible for alpha-KG production and consumption respectively, both increased in KO. However, enzymatic activity of OGDH was lower in KO. Thus, it was speculated that increased production and decreased clearance caused alpha-KG accumulation. There are several problems in this interpretation. 1) metabolites downstream of OGDH, succinyl-CoA, was also higher while other TCA cycle metabolites did not change in KO, not supporting OGDH was the bottleneck; 2) No explanation why OGDH activity was lower despite higher protein amount. It was speculated that OGDH activity was inhibited by higher BCAA catabolites in KO. This does not make sense since OGDH activity assay was performed in vitro which would not reflect the influence by metabolites in vivo unless there was co-valent modification. Further study of OGDH is required. 3) It was not shown whether only protein levels of ICDH and OGDH changed or other enzymes in the TCA cycle also changed. If all enzymes in TCA cycle are upregulated in KO, why only alpha-KG accumulates?*

Response: We thank the reviewer for raising this critical issue. To comply with the reviewer's request, we performed additional experiments.

1) Succinyl-CoA is not only an intermediate metabolite of the TCA cycle but also the end product of BCAA catabolism in mitochondria. Targeted metabolomics revealed an increase of intermediate metabolites from BCAA catabolism in *Cpt1b*-deficient hearts including initial catabolic products (branched-chain α -keto acids such as KIV, KMV) and the precursor of succinyl-CoA (Propionyl-CoA and Methylmalonyl-CoA) (Revised Fig. 3d, e). Moreover, we performed metabolic flux experiments using ^{13}C -Isoleucine. Inactivation of *Cpt1b* resulted in an increased contribution of ^{13}C -Isoleucine to the production of acetyl-CoA and propionyl-CoA compared to control animals, indicating a pronounced activation of BCAA catabolism in the absence of FAO to compensate for production of intermediate metabolites (Revised Fig. 3g). Thus, it is reasonable to assume that the higher levels of succinyl-CoA in *Cpt1b*-deficient hearts are contributed by increased BCAA catabolism. Processing of BCAA is well known to contribute to the energy metabolism of the heart and will be enhanced when other energy sources such as fatty acids are not used any longer.

2) The OGDH enzymatic activity in the original manuscript (Revised Fig. 3i) was measured by using extracts prepared from isolated primary adult cardiomyocytes (CMs) without prior cultivation. Strictly speaking, we performed an *ex vivo* but not an *in vitro* experiment. The use of lysates from freshly isolated control and *Cpt1b*-deficient adult CMs for enzymatic measurements made sure that potential inhibitory effects by metabolites from the BCAA metabolism are accurately reflected. We now directly determined increased concentrations of intermediate metabolites from the BCAA catabolism (KIV and KMV) (Revised Fig. 3d), which are known to inhibit enzymatic activity of the OGDH complex²¹⁻²⁴. Taken together we provide evidence that accumulation of KIV and KMV accounts for the suppression of OGDH enzymatic activity, as a consequence of *Cpt1b* inactivation.

In addition to the inhibition by KIV and KMV, it is indeed possible that a covalent modification contributes to the reduced enzymatic activity of OGDH after *Cpt1b* inactivation. We attempted to purify OGDH by immunoprecipitation (IP) for mass spectrometry, which would allow detection of potential covalent modifications. However, despite extensive efforts, which included the use of 3 different antibodies from

Sigma, Proteintech, and Cell signaling, we were unable to recover OGDH. None of these antibodies worked in the IP experiments (Fig. 1a for reviewer).

To further validate the causality between OGDH function and α KG accumulation in *Cpt1b* KO heart, we generated conditional *Ogdh* knockout mice, in which the OGDH enzyme was depleted in CMs 3-4 weeks after TAM injections (Revised Extended data Fig. 7f, g). We detected a strong accumulation of α KG after inactivation of *Ogdh* in adult CMs (Revised Fig. 3j). Importantly, conditional inactivation of *Ogdh* in CMs recapitulated the phenotype observed in *Cpt1b* KO animals including reduced H3K4me3 levels, resulting in reversed maturation of CMs and elevated CM proliferation (monitored by Ki67, pH3, Aurora B staining, and cell number analysis) (Revised Fig. 3k-o, Revised Extended data Fig. 7h-m, Revised Extended data Fig. 8b). The major increase of α KG concentration in CMs after conditional *Ogdh* inactivation, recapitulating the phenotype of *Cpt1b* KO animals, demonstrates that reduced OGDH activity is sufficient to increase α KG concentrations, epigenetic changes and cellular reprogramming. However, unlike *Cpt1b*-deficient animals, *Ogdh*-mutants die relatively early after gene inactivation, which makes them suitable for monitoring reduced H3K4me3 levels, reversed maturation of CMs, and elevated CMs proliferation as a consequence of increased α KG but not for cardiac regeneration studies.

3) RNA-seq data revealed that genes coding for TCA enzymes maintain regular expression levels. To comply with the reviewer's request, we also conducted western blot assays but did not detect any obvious changes in protein levels of various TCA-related enzymes including CS, ACO2, SDHA, SDHB, SDHC, MDH2 and FH (Revised Extended data Fig. 6m).

Figure 1 for reviewer: a, OGDH immunoprecipitation (IP) experiment using C2C12 lysates and antibodies from Proteintech, Cell Signaling and Sigma.

Fatty acids are the primary fuel of the heart. Deletion of CPT1b deprives mitochondria of fatty acid supplies but the KO heart presents normal contractile function. This suggested that inhibition of FAO minimally affected energy supply in KO. Metabolic adaptation to sustain energy homeostasis may be key to the reprogramming observed here, including the intriguing accumulation of alpha-KG. However, metabolic characterization was done poorly. The authors mentioned a variety of mechanisms, e.g. glycolysis, amino acids metabolism, pyruvate oxidation, but seemed uncertain which ones are contributing. Indeed, there was no evidence of increased glycolysis although the authors stressed it in the discussion. Also unclear how changes in amino acid metabolism described in the results could contribute to energy homeostasis. Ultimately, measurement of metabolic flux is required to sort out the reprogramming of fuel metabolism.

Response: We agree with the reviewer that a more comprehensive characterization of compensatory metabolic changes might be helpful for better understanding of the cellular processes induced by depletion of *Cpt1b*. Nevertheless, we would like to emphasize that the study provides compelling evidence for the critical role of KDM5B activation by α KG, leading to removal of H3K4me3 from cardiac identity genes, which results in increased proliferation of CMs. Thus, the additional metabolic changes, necessary to cope

with the absence of FAO, are primarily relevant to understand the accumulation of α KG. We demonstrate using the newly generated *Ogdh* KO model that inhibition of OGDH is sufficient to increase α KG, which causes the same effects as the increase of α KG, induced by inactivation of *Cpt1b*. Furthermore, we show that the increase of KIV and KMV (Revised Fig. 3d), which inhibit enzymatic activity of OGDH, is the result of increased BCAA catabolism. Nevertheless, we followed the reviewer's recommendations and performed seahorse experiments to study changes in energy metabolism. We found that addition of pyruvate and BCAA to medium containing low levels of FA increased energy production more strongly in *Cpt1b*-deficient adult CMs than in WT CMs. Additional supplementation of glucose further enhanced these differences, indicating that *Cpt1b*-deficient adult CMs utilize pyruvate, BCAA and glucose more efficiently than Ctrl CMs (Revised Extended data Fig. 6i, j). Of course, the more efficient utilization of pyruvate, BCAA and glucose by *Cpt1b*-deficient compared to WT CMs, does not prove definitively that *Cpt1b*-deficient CMs completely satisfy their energy needs by switching from FAO to pyruvate and BCAA catabolism but it illustrates a higher capacity to do so.

Of note, we also performed Seahorse experiments to measure the oxygen consumption rate (OCR) in control and *Cpt1b* deficient adult CMs after supplementation of palmitic acid to confirm the lack of fatty acid oxidation (FAO) in *Cpt1b* KO CMs. As expected, treatment with palmitic acid but not BSA dramatically elevated OCR in control CMs. In contrast, *Cpt1b*-deficient adult CMs did not show any significant changes of OCR in response to palmitic acid supplementation, demonstrating that *Cpt1b* deficient adult CMs are unable to utilize palmitate for energy production (Revised Extended data Fig. 6a). These new findings validate the metabolomics analysis, revealing a substantial reduction of acyl-carnitine derived from long chain fatty acids (LCFAs) in *Cpt1b* deficient CMs, indicating reduced transport of free fatty acids into mitochondria (Revised Extended data Fig. 6b, c). To further validate our findings, we performed metabolic flux assays by perfusing WT and mutant hearts with ^{13}C -Palmitate and measuring intermediate metabolites from the FAO pathway. Due to technical limitations, we monitored intermediate CoAs such as Decanoyl-CoA and Decenoyl-CoA, which represent C10-CoAs derived from the breakdown of ^{13}C -labelled palmitate. Consistent with our Seahorse data, we observed a massive reduction of both C10-CoAs in *Cpt1b*-deficient heart, indicating a lack of LCFAs oxidation (Revised Fig. 3a).

We already demonstrated in the initial version of the manuscript increased ammonia recycling and urea cycle activity in *Cpt1b* mutant CMs, which are closely related to increased amino acid catabolism (Revised Extended data Fig. 6f). We now also show increased levels of KIV and KMV, initial products of BCAA catabolism, and an increase in propionyl-CoA and methylmalonyl-CoA, which all argues for enhanced BCAA catabolism that compensates for inhibition of FAO (Revised Fig. 3d, e). Moreover, metabolic flux experiments revealed elevated production of ^{13}C -acetyl-CoA from either ^{13}C -Glucose or ^{13}C -Isoleucine after inactivation of *Cpt1b*, indicating that both BCAA catabolism and glucose/pyruvate oxidation contribute to compensate for abrogation of FAO (Revised Fig. 3f, g).

2. The I/R experiment was problematic.

Critical information, such as area at risk (AAR) and % infarct area out of AAR were missing. They are required to provide assurance that I/R caused comparable insult in all groups. Histology showed very small infarct in control I/R hearts. Have the authors aimed to produce significant I/R injury as described in the literature?

Response: We followed the reviewer's suggestion and measured additional parameters as requested. We determined the area at risk (AAR) and the infarct (IF) area 24 hours after the I/R surgery by injection of Evans Blue combined with TTC staining. The AAR induced by I/R was similar in Ctrl and mutant (*Cpt1b*^{CKO} and *Cpt1b*^{iKO}) hearts, accounting for 60-70% of the myocardium, which is comparable to published reports (Revised Fig. 2e, f, Revised Extended data Fig. 5a, b), suggesting that I/R caused equivalent damage in all groups¹⁶⁻¹⁸. The infarct area was significantly lower in the *Cpt1b*^{CKO} and *Cpt1b*^{iKO} compared to control animals, suggesting that *Cpt1b* inactivation exerts cardio-protective effects during the early response to myocardial ischemia as already pointed out in the initial submission. We have also repeated quantification of the scar areas using Trichrome staining and replaced the representative images accordingly (Revised Fig. 2a-d).

To clearly distinguish between cardio-protective and pro-regenerative effects of *Cpt1b* inactivation and to demonstrate more convincingly the ability of *Cpt1b*-deficient hearts to regenerate, we conducted a whole series of new experiments. Instead of *Cpt1b* inactivation before I/R surgery, we inactivated *Cpt1b* after I/R surgery. Importantly, we observed a strong reduction of the fibrotic scar area in *Cpt1b*^{iKO} compared to Ctrl animals (Revised Fig. 2h, i). These data unequivocally demonstrate that inactivation of *Cpt1b* facilitates cardiac regeneration by reversing maturation of CMs, leading to proliferation of adult CMs.

Many studies have shown that upregulation of HIF1a and/or switch from FAO to glucose metabolism prevent ischemia/reperfusion injury. This part of the finding is confirmatory but not novel. However, it would be interesting to ask whether a proliferating heart that at the same time unable to utilize FAO for energy supply can maintain contractile function after I/R insult. In an I/R heart with sufficient myocardial injury, the remaining myocardium is subjected to higher workload. Previous studies showed that CPT1b+/- hearts could not tolerate increased workload. How does cardiac function and survival of CPT1b KO look immediately after I/R? It has also been suggested that promoting cardiomyocytes re-entry into cell cycle would compromise contractile function in adult hearts. It will be very significant to determine whether metabolic reprogramming would remedy or exacerbate the situation.

Response: We followed the reviewer's request and assessed cardiac function by MRI at the earliest possible time point. Due to technical and regulatory issues, the first time point when MRI measurements are feasible is 48 hours after I/R surgery. 24 hours after I/R surgery, animals still suffer from pain, reduced physical activity, and are losing weight due to the anesthesia. Unfortunately, MRI measurements require additional anesthesia, causing high lethality of animals when done immediately or 24 hours after surgery, which is not compatible with the approved animal regulations. Therefore, we performed MRI measurements before and 48 hours after I/R surgery. No statistical difference in cardiac functions was evident between control and mutant mice before I/R surgery (Revised Fig. 2g). 87.5% of Ctrl and 100% mutant animals survived 24 hours after I/R surgery. 48 hours after I/R surgery, the ejection fraction (EF) of Ctrl animals was lower than 45%, while the EF of *Cpt1b*-deficient animals -remarkably- did not decrease significantly, albeit a tendency for reduction was observed (Revised Fig. 2g). We also determined the presence of detyrosinated-tubulin 24 hours after I/R surgery, which correlates with reduced cardiac contraction^{19,20}. We detected clear signals for detyrosinated-tubulin in the infarct zone and border zone of Ctrl animals, which was much lower in corresponding areas of *Cpt1b*-deficient animals. Detyrosinated-tubulin was also detected in the remote area of Ctrl mice after I/R surgery, indicating reduced contractility due to increased workload. In contrast, no detyrosinated-tubulin was present in the remote zone of *Cpt1b*-deficient animals (Revised Extended data Fig. 5d).

We already described in the initial version of the manuscript that CM-specific inactivation of *Cpt1b* does not compromise cardiac functions under baseline conditions (Revised Extended data Fig. 2g, h, Revised Extended data Fig. 2o). The new data clearly demonstrate that stimulation of CM re-entry into cell cycle after I/R injury does not compromise contractile function in damaged adult hearts. Instead, we observed improved function of *Cpt1b*-deficient hearts 48 hours after I/R injury, which we ascribe to the protective effects of *Cpt1b* inactivation. Regenerative effects are likely to contribute to improved heart function only at later stages as demonstrated by inactivation of *Cpt1b* after I/R injury. Potential adverse effects of reversed CM maturation and cell cycle entry on cardiac function, which are indeed likely, are apparently well-compensated by the cardioprotective effect of metabolic reprogramming.

3. Missing links connecting alpha-KG accumulation, epigenetics change and proliferating cardiomyocytes. The authors made compelling observations that supplementation of cell permeable alpha-KG or manipulating KDM5 activity could alter H3K4me3 and related gene expression. These actions occur outside mitochondria. How does inhibition of mitochondrial FAO lead to increased alpha-KG in the cytosol/nucleus? Metabolomics or enzyme activity assays did not distinguish the two subcellular compartments.

Response: We completely agree with the reviewer that accurate measurements of metabolites in distinct sub-cellular compartments are important to understand the connection between increased generation of α KG in mitochondria, accumulation of α KG in the cytosol/nucleus, and epigenetic changes. However, we respectfully would like to point out that it is out of the scope of our study to answer all open questions. It is still a major problem in the field how metabolites from mitochondria reach the nucleus and how the concentrations of individual metabolites in different cellular organelles look like. α KG may exit mitochondria by different means, including conversion to the α KG intermediates citrate and glutamate, which are then converted back to α KG in the cytosol or nucleus. We gathered several lines of evidence suggesting that cytosolic and locally generated nuclear α KG contribute to the high α KG accumulation in *Cpt1b*-mutant CMs, although we are unable to accurately determine α KG levels in different subcellular compartments: 1) The IDH1 enzyme, which mainly localizes in the cytoplasm to convert isocitrate to α KG, was elevated in *Cpt1b*-mutant CMs (Revised Fig. 3h, Revised Extended data Fig. 6k), suggesting increased α KG production in the cytoplasm. Cytosolic α KG might passively diffuse or actively transport into the nuclear compartment via undefined α KG transporters. 2) We found that the α KG-generating enzyme IDH3G is present in the nuclei of CMs based on analysis of FACS-purified CM nuclei, suggesting local generation of α KG from citrate in the nucleus (Fig. 2a for reviewer). 3) The level of ACL, which converts citrate to Acetyl-CoA was dramatically reduced in the nucleus after *Cpt1b* inactivation, indicating that the ratio of citrate conversion to either acetyl-CoA or α KG has shifted, making α KG the more dominant product of citrate conversion in nuclei of *Cpt1b*-deficient adult CMs (Fig. 2 b, c for reviewer). Whether/how enzymes related to α KG metabolism shuttle between cytoplasm/mitochondria and nucleus under different conditions, how metabolites get transported among different organelles, and how this relates to each other is a subject of interest that will be studied in the future.

[REDACTED]

The authors showed that iKO in adult mice could trigger cardiomyocyte proliferation. Experiments manipulating alpha-KG or KDM5 were performed in neonatal cardiomyocytes. Will similar changes be observed in adult cardiomyocytes? This is needed to address the question whether metabolic reprogramming prevents or reverse cell cycle arrest.

Response: We attempted to overexpress *Kdm5b* in CMs *in vivo* by using an AAV9 virus. Unfortunately, the insert consisting of the cTnT promoter and the coding sequence of *Kdm5b* (4.5 kb) is relatively large, preventing efficient transduction of CMs with the AAV9-*Kdm5b* virus. Instead, we treated isolated WT adult CMs with cell-permeable α KG. Treatment with α KG promoted EdU incorporation of cultured adult CMs, which was not observed in the DMSO-treated CMs. Importantly, increased incorporation of EdU in response to treatment with α KG was prevented by the KDM5 inhibitor CPI. These findings suggest that α KG-treatment promotes cell cycle re-entry of adult CMs in a KDM5-dependent manner (Revised Extended data Fig. 9k).

It is very challenging to evaluate the impact of α KG treatment on maturation of adult CMs by transcriptional profiling, since adult CMs rapidly de-differentiate in culture. To further prove the crucial role of α KG for promoting proliferation adult CMs, we therefore generated another *in vivo* model. We conditionally inactivated the *Ogdh* gene in CMs. *Ogdh* inactivation led to marked accumulation of α KG and enhanced cell cycle activity as indicated by elevated numbers of Ki67⁺, pH3⁺, and Aurora B⁺ adult CMs. These findings provide additional evidence that α KG accumulation and metabolic reprogramming is sufficient for cell cycle re-entry of adult CMs (Revised Fig. 3j-o, Revised Extended data Fig. 7f-m).

The study showed that reduction of H3K4me3 promoted the expression of immature cardiac gene. What mechanism(s) suppress the mature cardiac genes in order for adult cardiomyocytes become proliferating? Did those mechanisms also reverse the metabolic maturation, e.g. mitochondrial biogenesis or oxidative metabolism?

Response: We are sorry that we did not communicate our findings more clearly. We demonstrate that activation of the demethylase KDM5B by increased levels of α KG diminishes H3K4me3 in CMs on broad H3K4me3 peaks, which are characteristic for cell identity or maturation-related cardiac genes (e. g. sarcomere assembly or contraction-related genes). Several lines of evidence support the conclusion that accumulation of α KG after metabolic reprogramming activates KDM5 and resets the epigenome in *Cpt1b* deficient CMs: (i) Inhibition of KDM5 activity by CPI treatment prevents pro-proliferation effects of α KG (Revised Fig. 5e, f, Revised Extended data Fig. 9j, k). (ii) ChIP-qPCR analysis revealed that KDM5 binds directly to maturation-related CM genes containing broad H3K4me3 peaks. Such broad H3K4me3 peaks are characterized for cell-identity genes in various cell types (Revised Fig. 5j). (iii) Overexpression of *Kdm5b* promoted CM proliferation, and this pro-proliferative effect was further enhanced when combined with α KG supplementation (Revised Fig. 5g, h, Revised Extended data Fig. 10c, d). Thus, we do not encounter an active suppression of maturation-related cardiac genes but rather a loss of activation due to enhanced enzymatic activity of KDM5 and reduced deposition of H3K4me3, which converts adult CMs to a less mature state.

It is not clear how pro-proliferative genes including cell-cycle and HIF1-signaling-related genes are transcriptionally activated. We assume that reduced expression of maturation-related cardiac genes is the initial event, which then allows increased activity of pro-proliferative genes as a secondary effect, probably by removing inhibitory cues. We envision a crosstalk among different epigenetic markers resulting in dissociation of an interlinked negative regulatory network that balances expression of genes characteristic for either mature or immature states. However, at present we are unable to obtain further molecular insights into the machinery that links reduced expression of maturation-related genes with increased expression of genes characteristic for immature CMs.

Based on transcriptional profiling, accumulation of α KG does not alter mitochondrial biogenesis and oxidative metabolism. Key genes related to mitochondrial biogenesis and oxidative metabolism are not changed in CMs after inactivation of *Cpt1b* (Fig. 2d for reviewer). We compared the number of mitochondria in *Cpt1b* KO and Ctrl CMs by determining mtDNA copy numbers per cell for each genotype. No significant differences were evident (Revised Extended data Fig. 6n). Moreover, we observed that treatment of *Cpt1b*-deficient CMs with pyruvate, BCAA and glucose strongly elevates OCR, indicating that mitochondria in *Cpt1b*-deficient CMs are functional. Taken together, these results suggest that inactivation

of *Cpt1b* does not alter the number of mitochondria in CMs and that FAO-independent mitochondrial functions are not compromised.

4. Other points:

Some misconceptions of metabolism in the text should be corrected. a) the heart switches from glycolysis to FAO in perinatal period not one week after birth (first paragraph in Discussion). b) Oxidative metabolism of any substrates, not just FAO, generates ROS (Introduction). The fact that CPT1b KO cardiomyocytes continue oxidative metabolism and at the same time proliferate questions the role of ROS induced DDR as a major mechanism of cell cycle arrest. c) Inhibition of FAO does not increase glycolysis only, it has to increase oxidation of other substrates to maintain ATP supply of the heart (first paragraph of Discussion).

Response: We apologize for these shortcomings and have changed the text according to the reviewer's suggestions to avoid any misleading statements about cardiac metabolism.

Concerns of statistical analysis:

Many experiments have n=3. How was statistics analysis performed?

Response: The n=3 in the figures (Revised Fig. 1e-h, l-n; Fig. 3i, l; Fig. 5b, j; Revised Extended data Fig. 1d, e, g, n-o; Fig. 2. i; Fig. 3e; Fig. 4c, d; Fig. 5e; Fig. 6e, j-l; Fig. 7g; Fig. 8a, b; Fig. 9b, d, i, k; Fig. 10c) refers to the number of the hearts used for quantification. For each individual heart, hundreds of CMs (both isolated or on tissue section) were quantified. Values from each group of CMs were averaged and are presented as one sample (n=1). For experiments with neonatal CMs, values for each sample represent the results obtained from one isolation of CMs from pooled neonatal hearts. The biological significance was calculated based on at least 3 biological replicates. We describe the statistical analysis in the corresponding figure legends and added an explanation in the methods part.

For measurements taken from individual cardiomyocyte, results from all cardiomyocytes isolated from one heart should be considered n=1. Counting each cardiomyocyte from the same heart as an independent n=1 artificially inflates statistical power.

Response: As described above, we exactly followed the approach outlined by the reviewer to avoid artificial inflation of the statistical power. We have changed the revised manuscript and added the following statement to the methods part to clarify our strategy: "For other quantifications, either using isolated CMs or tissue sections, hundreds of CMs were analyzed for each individual heart. Values from each group of CMs (or sections) were averaged and are presented as one sample (n=1). For experiments with neonatal CMs, values for each sample represent the results obtained from one isolation of CMs from pooled neonatal hearts".

Referee #2 (Remarks to the Author):

Major:

1) It is interesting that KDM5 alone can have these effects. How does it function though in the absence of increase α KG? The authors conclude from the first set of studies that *Cpt1b* mediated increase in α KG the reason for the KDM5 activity. Do α KG levels increase with KDM5 gain of function? And if so, how? This is a central point in the manuscript that needs to be clarified experimentally.

Response: We assume that an increase in KDM5 levels results in a more efficient utilization of α KG, even when α KG levels are not dramatically elevated. To directly address the relation between increased *Kdm5b* expression and enhanced α KG levels, we overexpressed *Kdm5b* in neonatal cardiomyocytes (CMs) with or without α KG treatment. α KG treatment alone or *Kdm5b* OE alone enhances CMs proliferation. Importantly, the combination of α KG treatment and *Kdm5b* OE further reduces H3K4me3 levels compared to *Kdm5b* OE alone and elevates the percentage of Ki67 positive cells, indicating an additive or synergistic function of KDM5B and α KG for stimulation of CMs proliferation (Revised Fig. 5h, Revised Extended data Fig. 10c, d). We assume that α KG levels in neonatal CMs are not completely exploited by regular levels of KDM5B, so that KDM5B becomes rate-limiting. Thus, overexpression of *Kdm5b* increases the enzymatic activity and α KG levels become rate-limiting. Accordingly, combined elevation of both KDM5B and α KG further boosts the enzymatic activity.

The α KG levels did not change after *Kdm5b* OE compared to control CMs (Fig. 1a for reviewer), suggesting that α KG levels are efficiently replenished when metabolized by KDM5B, probably due to anaplerotic processes.

Figure 1 for reviewer: Quantification of α KG levels in P0-1CMs transduced with a *Kdm5b* lentivirus. Error bar represents mean \pm s.e.m. Two-tailed, unpaired Student's t-test for statistical analysis.

2) It is interesting that in Figure S3, FA metabolism is upregulated at the transcript level. This raises an important concern as to whether there is a true shift in substrate metabolism. As such, flux analysis studies are crucial to confirm the metabolic consequences of *Cpt1b* deletion.

Response: We understand the reviewer's concerns. To investigate the shift in substrate metabolism, we performed Seahorse experiments to measure the oxygen consumption rate (OCR) in control and *Cpt1b*-deficient adult CMs after supplementation of palmitic acid. Similar experiments were done using pyruvate, BCAA, and glucose. As expected, treatment with FCCP dramatically elevated OCR in control CMs when supplemented with palmitic acid but not when supplemented with BSA. In contrast, *Cpt1b*-deficient adult

CMs did not show any significant changes of OCR in response to palmitic acid compared to BSA, demonstrating that *Cpt1b*-deficient adult CMs are unable to utilize palmitate for energy production (Revised Extended data Fig. 6a). These new findings validate the metabolomics analysis, revealing a substantial reduction of acyl-carnitine derived from long chain fatty acids (LCFAs) in *Cpt1b*-deficient CMs, indicating reduced transport of free fatty acids into mitochondria (Revised Extended data Fig. 6b, c). In line with these findings, the newly performed metabolic flux analysis unveiled a reduction of ¹³C labeled intermediate CoAs derived from ¹³C-palmitate in *Cpt1b*-deficient hearts, confirming absence of LCFAs oxidation (Revised Fig. 3a).

In contrast, addition of pyruvate and BCAA to medium containing low levels of LCFAs, increased energy production more strongly in *Cpt1b*-deficient adult CMs than in WT CMs. Additional supplementation of glucose further enhanced these differences, indicating that *Cpt1b*-deficient adult CMs utilize pyruvate, BCAA and glucose more efficiently than wild-type (Revised Extended data Fig. 6i, j). Of course, the more efficient utilization of pyruvate, BCAA and glucose by *Cpt1b*-deficient compared to WT CMs does not prove definitively that *Cpt1b*-deficient CMs completely satisfy their energy needs by switching from FAO to pyruvate and BCAA catabolism but it illustrates a higher capacity to do so.

We now also demonstrate increased levels of KIV and KMV, initial products of BCAA catabolism, and an increase in propionyl-CoA and methylmalonyl-CoA, which all argues for enhanced BCAA catabolism to compensate for the inhibition of FAO (Revised Fig. 3d, e). In addition, metabolic flux analysis demonstrated elevated ¹³C-Acetyl-CoA production from ¹³C-Glucose or ¹³C-Isoleucine in *Cpt1b*-deficient hearts, indicating increased activity of BCAA catabolism and glucose oxidation to compensate for abrogated FAO (Revised Fig. 3f, g).

3) Several previous reports indicate that inactivation of FA oxidation can result in cardiomyocyte proliferation citing various mechanisms including DNA damage, and as the authors state in their text, other redox modifying interventions have similar effects. It would be important for the authors to delineate whether there is Cpt1b-specific effects, or are these effects shared by all FA and ROS modulating interventions? For example, are ROS changed in the Cpt1b hearts? Is DNA damage changed? And if so, would artificial increase in ROS reverse the observed phenotype?

Response: To answer the reviewer's question, we determined the ROS levels and measured DNA damage in Ctrl and *Cpt1b*-mutant hearts using the ROS-specific DHR123 probe and γ H2A.X antibodies, respectively. No significant difference in the mean fluorescence intensity (MFI) for DHR123 was evident between Ctrl and *Cpt1b* KO hearts, indicating that inhibition of FAO does not attenuate ROS generation (Revised Extended data Fig. 4c). However, the percentage of γ H2A.X+ cardiac nuclei dropped dramatically in *Cpt1b*-deficient compared to control hearts, suggesting that the loss of *Cpt1b* triggers mechanism reducing DNA damage, independent of ROS (Revised Extended data Fig. 4d). Several processes may account for this phenomenon. For example, the dealkylase ALKBH3 (AlkB homolog 3), which remove alkylation adducts during repair of oxidative DNA damage²⁵, uses α KG as an essential co-substrate. The increased α KG levels in *Cpt1b*-deficient CMs may stimulate the enzymatic activity of ALKBH3 and promote DNA damage repair. Of course, it is entirely possible that other α KG-dependent enzymes than ALKBH3 participate in enhanced DNA repair. However, it seems unlikely that α KG scavenges ROS, which is known since some time, since ROS levels in CMs were not affected by inactivation of *Cpt1b*²⁶. We do not think that it is within the scope of this study to delineate the effects of α KG on DNA repair in more detail. Nevertheless, we now also demonstrate that addition of α KG reduces apoptosis (monitored by

TUNEL staining) and increases cell proliferation after H₂O₂ treatment when compared to cells not subjected to α KG treatment (Fig. 2a, b for reviewer). Furthermore, we found that exposure to ROS (H₂O₂ treatment) reduced proliferation of CMs after α KG-treatment, although the effects were moderate compared to the effects of α KG. We conclude that artificial increase in ROS indeed partially reverses the observed *Cpt1b*-KO phenotype.

[REDACTED]

4) Along the lines of point #1, recent reports by the Mahmoud group indicate that manipulation of cardiac metabolites by exogenously administering these metabolites can induce heart regeneration. If the author's claim that the effect of Cpt1b is dependent on the increase in α KG, does administration of α KG affect KDM5 levels? Or cardiomyocyte maturation and proliferation? Certainly, this would be a critical therapeutic angle.

Response: RNA-seq analysis does not indicate significant differences in the expression of *Kdm5* (5a, 5b and 5c) between primary adult wild-type and *Cpt1b*-deficient CMs, despite the strong accumulation of α KG (Revised Extended data Fig. 8c). Furthermore, RT-qPCR analysis revealed that *Kdm5b* expression in neonatal CMs is not affected by α KG treatment for 96 hours (Fig. 3a for reviewer).

The group of Ahmed Mahmoud reported recently that administration of malonate promotes heart regeneration²⁷. Malonate is a well-known inhibitor of succinate dehydrogenase (SDH). However, the mechanisms underlying heart regeneration after malonate administration were not explored in the study by Bae et al. It is not clear whether inhibition of SDH by malonate is instrumental for heart regeneration and effects of malonate administration on α KG levels and KDM5B activity were not studied²⁷. Regarding systemic administration of α KG: it has been demonstrated that administration of α KG has cardiac protective

effects by stimulation of kynurenic acid production in the liver²⁸, which make it difficult to analyze the effect of α KG on CMs specifically.

To further substantiate the critical role of α KG accumulation for stimulation of KDM5B activity, depletion of H3K4me3, and induction of heart regeneration, we inactivated *Ogdh* specifically in cardiomyocytes (Revised Extended data Fig. 7f). OGDH (oxoglutarate dehydrogenase) is the enzyme that catalyzes the conversion of α KG into succinyl-CoA. We observed profound depletion of OGDH 3-4 weeks after induction of recombination by TAM injections (Revised Extended data Fig. 7g). Inactivation of *Ogdh* in CMs results in substantial accumulation of α KG as expected (Revised Fig. 3j). Importantly, conditional inactivation of *Ogdh* in CMs recapitulated the phenotype observed in *Cpt1b* KO animals including reduced H3K4me3 levels, resulting in reversed maturation of CMs, and elevated CMs proliferation (monitored by Ki67, pH3, Aurora B staining, and cell number analysis) (Revised Fig. 3k-o, Revised Extended data Fig. 7h-m, Revised Extended data Fig. 8b). Recapitulation of the *Cpt1b* phenotype in a different mouse model that also shows a major increase in α KG concentrations in CMs strongly supports our conclusions about the underlying mechanisms and the critical role of α KG in this process. From a therapeutic viewpoint it will be highly interesting to develop new methods for CMs-specific delivery of α KG and/or KDM5, identification of novel compounds increasing KDM5 enzymatic activity, or compounds that inhibit CPT1B specifically in CMs but not in skeletal muscle cells.

Fig. 3 for reviewer: a, RT-qPCR analysis of *Kdm5b* expression in neonatal CMs treated with DMSO and α KG (n=3). *36b4* expression was used as reference. Two-tailed, unpaired student t-test was performed for statistical analysis.

5) *The dichotomy of increased cell size and proliferation in the Cpt1b ko hearts is very interesting. It is typically viewed that immature cardiomyocytes are smaller, as indicated by numerous reports. How do the authors reconcile that difference? Is it possible to be immature, proliferative, and hypertrophied at the same time? Can the authors provide some examples?*

Response: We do not expect that all CMs respond in an identical manner to the increase of α KG, which is supported by the observation that only subset of *Cpt1b*-deficient CMs enter the cell cycle. We assume that differences in the properties of CMs, reflected by differential gene expression have a major impact on the response to the increase of α KG. We agree with the reviewer that concomitant hypertrophy and proliferation of the very same heart muscle cell is unlikely. To clarify this issue, we analyzed more carefully the distribution of CMs cell sizes after inactivation of *Cpt1b*. Interestingly, the CM population with the smallest cell size (<100 μm^2) increased markedly after *Cpt1b* inactivation, suggesting that smaller CMs are more prone to proliferation in response to *Cpt1b* inactivation. However, we cannot rule out increased division of large CMs, which generate smaller CMs. The percentage of medium-sized CMs (100-300 μm^2) decreased,

whereas the number of larger CMs (>300 μm^2) increased significantly, which accounts for the overall hypertrophic growth of CM when the whole CM population is assessed and not distinct classes of CMs with different sizes (Revised Fig. 1h). Our findings do not contradict previous reports that smaller immature CMs are more prone to proliferation than larger CMs. We have added the new results to the revised manuscript.

6) *It is very interesting that the drug etomoxir can induce cardiomyocyte proliferation in vitro. From a therapeutic standpoint, the authors should test this in vivo. Does etomoxir induce cardiomyocyte proliferation in vivo? Can it induce regeneration? This needs to be tested.*

Response: A previous study demonstrated that inhibition of FAO by etomoxir improves heart function of heart failure patients in clinical trials²⁹. Etomoxir is a widely used pan-inhibitor of CPT1, which inhibits CPT1 irreversibly. Concomitant inhibition of CPT1A and CPT1B will suppress FAO in numerous cell types, which has grave side effects. In fact, etomoxir is highly hepatotoxic and inhibition of CPT1B in both skeletal muscle and heart muscle cells results in lipotoxicity.

Nevertheless, in order to comply with the reviewer's request, we treated C57/BL6 WT mice for 2 weeks with etomoxir. Surprisingly, the etomoxir treatment partially recapitulated the phenotype of *Cpt1b* KO mice including elevated HW/BW ratios, elevated numbers of Ki67, pH3, and Aurora B positive CMs (Fig. 3a-d for reviewer). Unfortunately, etomoxir-treated animals showed a strong physical burden due to the known hepatotoxicity, which prevented us from performing heart regeneration experiments. We believe that development of a selective inhibitor of CPT1B that ideally targets only CMs holds great promise for future therapeutic approaches. Due to space restrictions and the controversial discussions about the use of etomoxir in *in vivo* experiments, we do not show these data in the manuscript.

[REDACTED]

7) The authors use Ki67 for proliferation studies *in vivo*. However, Ki67 is not an accurate indicator of mitosis. The standard in the field now includes pH3 and aurora B kinase +/- stereology. Some of these should be shown especially for the *in vivo* studies

Response: We extended the analysis of pH3⁺ and Aurora B kinase⁺ CMs as requested by the reviewer. We found a substantial increase of pH3⁺ and Aurora B⁺ CMs in heart tissue sections of *Cpt1b* mutant hearts (Revised Fig. 1f, g, l-m; Revised Extended data Fig. 2a-c, p-r). The new data have been included in the revised manuscript. We are well aware that even staining for pH3 and Aurora B is not unequivocal proof for division of CMs. Therefore, we strongly emphasize the increase of the overall CM numbers after *Cpt1b* and *Ogdh* inactivation in CMs, which was done by counting CMs numbers after complete digestion of the hearts, a method that has also been widely used in other studies³⁰⁻³³.

8) In Figure 4 the KO heart size in panel A is massive, even compared to the KO heart size in panel B despite being a younger mouse. Is this to scale? Regardless the use of the Cre model rather than the MCM model here is not appropriate. If in fact this constitutes a regenerative effect, deletion should occur after MI, not before. This will allow for assessment of a pro regenerative effect irrespective of a potential protective effect.

Response: Images of hearts in (Revised Fig. 2a) are at the same scale. The differences of heart sizes between Ctrl and *Cpt1b*-mutant hearts are also obvious before I/R surgery and are caused by *Cpt1b* inactivation, which increases CM proliferation and the number of CMs 1.7-fold per heart (Revised Fig. 1c, Extended data Fig. 1k). Proliferation of CMs in *Cpt1b*^{CKO} mice is a persisting process. We observed increased CM numbers 4 weeks after TAM-induced *Cpt1b* inactivation, which was further enhanced 8 weeks after TAM (Revised Fig. 1n, o). The section shown in (Revised Fig. 2c) was prepared from mice 4 weeks post-TAM, at a time point when differences in heart sizes between Ctrl and *Cpt1b* KO are only moderate.

The reviewer is obviously right to ask for a better distinction between cardio-protective and cardiac-regenerative effects. We think we accumulated sufficient evidence for both cardio-protective and cardiac-regenerative effects due to inactivation of *Cpt1b* but the initial submission did indeed not include experiments in which *Cpt1b* was inactivated after I/R injury. We have now eradicated this shortcoming. To clearly distinguish between cardio-protective and pro-regenerative effects of *Cpt1b* inactivation and to demonstrate more convincingly the ability of *Cpt1b*-deficient hearts to regenerate, we conducted a whole series of new experiments. Instead of *Cpt1b* inactivation before I/R surgery, we inactivated *Cpt1b* after I/R surgery. Importantly, we observed a strong reduction of the fibrotic scar area in *Cpt1b*^{CKO} compared to Ctrl animals (Revised Fig. 2h, i). These data unequivocally demonstrate that inactivation of *Cpt1b* facilitates cardiac regeneration by reversing maturation of CMs, leading to proliferation of adult CMs.

Minor:

1) In figure 1D, Y axis probably should be per HPF? Or plate? These are not sections

Response: We are sorry for this mistake. We used chamber slides. We have changed the label of Y-axis in (Revised Extended data Fig. 1d) accordingly.

2) *The cardiomyocyte proliferation images in vivo are too small and not clear. Please provide high resolution images of mitotic cardiomyocytes where applicable.*

Response: We prepared new samples and redid the staining and imaging with Ki67 antibodies. The old images (original Fig. 2h, Fig. 3g) have been replaced by larger images of higher quality in the (Revised Extended Fig. 2a and revised Extended Fig. 2p).

3) *I am not convinced that there is almost doubling of cardiomyocyte number as shown in figure 2L. The heart size does not reflect that. There is a moderate increase in heart size accompanied by hypertrophy (which could certainly explain the increase in heart size). An additional doubling of cardiomyocyte number would likely result in a far greater increase in heart size. Certainly, my comments are arbitrary, and I do realize that getting an accurate numerical correlation between heart size and cell number is not possible. However, perhaps the authors can show an additional parameter like stereology which would greatly enhance confidence in the data.*

Response: We are also stunned by the remarkable increase of CMs numbers, which increased by 1.7-fold to be precise, and the increase in heart size that was also evident from the increase in heart weight and the cMRI analysis. The counting was repeated several times using different animals and now also using a completely different model (CM-specific inactivation of *Ogdh*). We have replaced the images in (Revised Extended data Fig. 1k) to provide a more representative image, reflecting the increase in size. We also would like to point out that a 2-D representation of the heart size might be misleading and does not completely reflect the real size of the heart in 3 dimensions. As also pointed out by the reviewer it is very difficult to calculate changes in the size of an organ based on the number of cells in the respective organ.

4) In figure 3, it is interesting that *Hif1* and *mTOR* mRNA are upregulated. This needs to be discussed in the discussion section as a possible explanation for the observed hypertrophy and proliferation phenotype.

Response: We are grateful for this suggestion. We analyzed phosphorylation of p70 S6 Kinase, as a marker of mTOR activity. We did not detect a significant increase of p-S6K after inactivation of *Cpt1b* (Fig. 5 for reviewer). Likewise, we did not observe increased phosphorylation of mTORC1 and ULK1 (not shown). Thus, it seems not likely that increased mTOR activity contributes to hypertrophy and hyperplasia after inactivation of *Cpt1b*. We would have loved to add a paragraph to the discussion, which addresses the mechanisms leading to CM hypertrophy. Unfortunately, the space limitations of the journal do not allow us to expand on this topic. We hope this is acceptable for the reviewer.

Figure 5 for reviewer: Western Blot Analysis of phospho-p70 S6 Kinase and pan p70 S6 Kinase using adult CMs isolated from *Ctrl^{Cre}* and *Cpt1b^{cko}* animals (n=3).

Referee #3 (Remarks to the Author):

Major Points:

1. One central argument is that α KG accumulation is due to elevated levels of IDH1, IDH2, and IDH3A proteins resulting in increased α KG production (Figure 5E, 5I). First, is this elevation of IDH proteins at the RNA level or only protein level? Second, importantly, to confirm this mechanism, authors should demonstrate that inhibition/depletion of IDH proteins prevents α KG accumulation in *Cpt1b*KO, and prevents the loss of H3K4me3 levels by Western and ChIPseq.

Response: We argue that both an increase in IDH activity as well inhibition of OGDH activity via metabolites of the BCAA pathway contribute to accumulation of α KG. We did not observe a significant increase of RNA-seq reads for *Idh1*, *Idh2*, and *Idh3a* in *Cpt1b* KO CMs, but a substantial elevation of IDH1, IDH2, and IDH3A proteins, which indicates that a posttranscriptional mechanism leads to increased IDH activity. Unfortunately, no specific inhibitor of IDH3 is available. The commercially available inhibitors for IDH1/2 target the catalytic activity of mutant IDH1 or IDH2, which generate the oncometabolite (2HG), counteracting α KG. Obviously, such inhibitors are not suitable for our purposes. Moreover, it is difficult to knock down all five *Idh* genes (e.g. *Idh1*, *Idh2*, and *Idh3a*, *Idh3b* and *Idh3g*) simultaneously. Therefore, we pursued an alternative, gain-of-function approach by overexpressing *Idh3b* and *Idh3g*, two components of the IDH3 complex, in neonatal CMs. Increased expression of *Idh3b* and *Idh3g* elevated α KG and reduced H3K4me3 levels in CMs, indicating enhanced KDM5 activity (Revised Extended data Fig. 9c, d). In addition, we also analyzed H3K4me3 deposition on promoter and TSS regions of cardiac maturation genes by ChIP-qPCR. *Idh3b* and *Idh3g* overexpression reduced deposition of H3K4me3 on cardiac maturation genes with broad H3K4me3 domains, corroborating our hypothesis. In contrast, no reduction was found on narrow H3K4me3 peaks, characteristic for house-keeping genes such as *Snx19* (Revised Extended data Fig. 9e).

Elevation of α KG levels due to overexpression of *Idh3b* or *Idh3g* significantly enhances expression of genes typical for immature CMs, dramatically reduces expression of maturation-related genes, similar to the situation in *Cpt1b*-deficient CMs, and stimulates proliferation of CMs (Revised Extended data Fig. 9f-h). Importantly, the pro-proliferative effects of α KG are reversed by treatment with the KDM5 inhibitor and *Kdm5* knockdown, but enhanced by concomitant overexpression of *Kdm5b*, further confirming that the pro-proliferation effect of α KG is KDM5 dependent (Revised Extended data Fig. 9j, k; Revised Extended data Fig. 10d, f).

Moreover, we generated a completely novel mouse model to assess the effects of OGDH inhibition on α KG accumulation, KDM5 stimulation, H3K4me3 depletion and CMs proliferation. OGDH (oxoglutarate dehydrogenase) is the enzyme that catalyzes the conversion of α KG into succinyl-coA. Inactivation of *Ogdh* in CMs resulted in substantial accumulation of α KG as expected. Importantly, conditional inactivation of *Ogdh* in CMs recapitulated the phenotype observed in *Cpt1b* KO animals including reduced H3K4me3 levels, resulting in reversed maturation of CMs, and elevated CMs proliferation (monitored by Ki67, pH3, Aurora B staining, and cell number analysis) (Revised Fig. 3j-o, Revised Extended data Fig. 7f-m, Revised Extended data Fig. 8b). Recapitulation of the *Cpt1b* phenotype in *Ogdh* KO CMs demonstrates that reduction of OGDH activity is able to cause a major increase in α KG concentrations in CMs and strongly supports our conclusions about the underlying mechanisms.

2. There is a major problem in the logic in focusing on the decrease in H3K4me3 in the *Cpt1b*KO. That is because equally or even more striking is the increase in repressive histone modifications H3K9me3, H3K27me3 (Fig 6A). Since both decreased H3K4me3 and increased H3K9me3/H3K27me3 cause lowered transcription, there is no strong justification to focus exclusively on H3K4me3 decrease—beyond the α KG increase and IDH increases. Hence, first, are SAM levels changed? SAM is a major metabolite and if it is increased, then this would provide increased cofactor for KMT enzymes for H3K9me3 and H3K27me3. Second, are the KMT enzymes for K9 and K27 increased? Third, does H3K27me3 increase by ChIPseq over the cardiac cell identity genes (which show decreased H3K4me3)? What about H3K9me3?

Response: We monitored SAM levels in primary adult CMs isolated from *Ctrl*^{Cre} and *Cpt1b*^{CKO} animals. No significant differences were detected among these two groups (Revised Extended data Fig. 8d). Transcription levels of KMTs for H3K9me3 (e.g. *Suv39h1*, *Suv39h2*, *Setdb1*, *Setdb2*) and H3K27me3 (e.g. *Ezh1*, *Ezh2*) did not differ between *Cpt1b*-deficient and Ctrl CMs (Revised Extended data Fig. 8c), suggesting that KMTs do not play essential roles in the transcriptional regulation of cardiac cell identity genes after *Cpt1b* deficiency.

To elucidate whether repressive histone modifications are involved in transcriptional suppression of cardiac cell identity genes, we performed ChIP-qPCR experiments for H3K9me3 and H3K27me3. No significant differences in H3K9me3 deposition were observed at promoter regions of key maturation-related genes (Revised Extended data Fig. 8j). We did not detect enrichment of H3K27me3 at the 5'-regulatory regions of the cardiac maturation-related genes *Myocd* (TSS), *Tnni3* (promoter), *Tnni3* (TSS), *Cacna1g* (TSS), *Actc1* (promoter), and *Actc1* (TSS) (Fig. 1a for reviewer). H3K27me3 is enriched at 5'-regulatory regions of *Myocd* (promoter), *Mylk3* (TSS), *Mylk3* (promoter), and *Cacna1g* (promoter) but no significant differences between control and *Cpt1b* KO CMs are present (Revised Extended data Fig. 8k). Thus, we conclude that the elevated global levels of repressive histone modifications such as H3K9me3 and H3K27me3 do not directly contribute to the transcriptional repression of cardiac cell identity genes in *Cpt1b* KO CMs.

To identify genes that show enrichment of H3K9me3 and H3K27me3 in *Cpt1b* KO CMs, we analyzed the deposition of H3K9me3 and H3K27me3 on genes that show transcriptional repression after inactivation of *Cpt1b* (based on RNA-seq data) but no changes in H3K4me3 content (based on ChIP-seq data). Interestingly, H3K9me3 was elevated at the promoter of *Bcl2l11*, *Bnip3* and H3K27me3 at the promoters of *Bcl2l11*, *Cited2* and *Pink1* (Fig. 1b, c for reviewer). These results suggest that the H3K9me3 and H3K27me3 contribute to the transcriptional repression of non-cardiac identity genes, probably as a secondary effect.

In addition to these arguments, we would like to draw the reviewer's attention to the KDM5 inhibition and overexpression experiments, which in our view unequivocally demonstrate the crucial role of H3K4me3 depletion for stimulation of CM proliferation. We demonstrate in numerous different experiments that inhibition of KDM5 and also *Kdm5b* knockdown (see below) efficiently antagonize pro-proliferative effects of α KG on CMs, fully restore reduced H3K4me3 levels instigated by α KG treatment, and increase expression of several key genes associated with CM maturation including *Tnni3* and *Mylk3* (Revised Fig. 5e, f, i; Revised Extended data Fig. 9i-k, Revised Extended data Fig. 10e-h). Likewise, overexpression of *Kdm5b* promotes CM proliferation, and this pro-proliferative effect is further enhanced when combined with α KG supplementation (Revised Fig. 5g, h; Revised Extended data Fig. 10b-d). We do not want to

exclude additional effects of increased H3K9me3, H3K27me3 formation but the decline of H3K4me3 is obviously decisive for the effects of *Cpt1b* (and *Ogdh*) inactivation in CM.

[REDACTED]

3. A third major finding is that the effects of *Cpt1b* mutations occurred more strongly on broad H3K4me3 peaks as compared to narrow. This is likely dependent on the localization of KDM proteins at broad vs narrow sites (Fig 7N) using ChIP-qPCR. To better demonstrate this key point, authors should perform genome-wide analysis of KDM5B to show this preferential pattern.

Response: We extensively tried to acquire new KDM5B ChIP-seq data but did not succeed. The ChIP-grade antibodies (Abcam: ab50958; Santa Cruz: sc-67035) that were used successfully in published studies are no longer available. We used the antibody from Cell Signaling, which worked nicely for ChIP-qPCR experiments but did not work for ChIP-seq. In the revised manuscript, we extended the ChIP analysis by determining enrichment of KDM5B at genomic regions of additional cardiac identity genes with broad H3K4me3 domains (e.g. promoter, TSS, intron 1, exon 3 of *Actc1*; promoter and exon1 of *Cacnalg*). We observe a remarkable enrichment of KDM5B in these regions in *Cpt1b* KO compared to control CMs (Revised Fig. 5j).

4. The authors indicate that the accumulation of α KG leads to increased expression of the cardiac proliferating genes through increased activity of KDM5. The authors introduce α KG into neonatal cardiomyocytes (CM) and show increase cardiomyocytes and relevant genes by RT-qPCR; to show specificity, the authors should carry out RNA-seq. In addition, the authors examine histone methylation by Western and see decrease in H3K4me3 but should also examine H3K27me3 for an increase (see above). The authors performed overexpression of KDM5B to promote cardiac maturation and proliferation (Fig 7M). Although the authors used chemical inhibition of KDM5B (Fig 7H – 7K) the authors should also demonstrate that depletion/deletion of KDM5B prevents decreased H3K4me3 in *Cpt1b* KO and negates the effects on maturation and proliferation.

Response: We followed the reviewer's suggestion and performed RNA-seq using RNA from neonatal CMs treated with DMSO or α KG. GO term enrichment analysis revealed transcriptional up-regulation of genes mainly associated with GO terms related to proliferation such as cell cycle, cell division, chromosome segregation and so on (Revised Fig. 5c), which fits nicely to the increased numbers of Ki67⁺, pH3⁺ and Aurora B⁺ CMs in *Cpt1b* KO hearts. Accordingly, genes that were down-regulated after α KG treatment belonged to biological processes such as cell differentiation, supporting the idea that *Cpt1b* inactivation drives CMs into a more immature state (Revised Fig. 5c).

We did not detect significant change of H3K27me3 levels in α KG-treated compared to DMSO mock-treated neonatal CMs (Revised Fig. 5d). We reason that in this short-term *in vitro* experiment (α KG-treatment for 3 days), presumably secondary effects such as the increase of H3K27me3 and H3K9me3 do not happen, which are increased in CMs 8 weeks after *Cpt1b* depletion.

We also performed *Kdm5b* knockdown experiments as requested. siRNA treatment efficiently reduced *Kdm5b* expression, resulting in marked elevation of H3K4me3 in neonatal CMs (Revised Extended data Fig. 10 e, g). Knockdown of *Kdm5b* prevents α KG-induced lowering of H3K4me3 and attenuates pro-proliferative effects of α KG (Revised Fig. 5i, Revised Extended data Fig. 10f). Consistent with these findings, *Kdm5b* knockdown prevents α KG-induced up-regulation of genes related to CM immaturity (e.g. *Nppb*, *Myh7*), and down-regulation of maturation-related genes (*Tnni3*, and *Mylk3*) (Revised Extended data Fig. 10g). No transcriptional changes were observed for genes containing narrow H3K4me3 peaks (*Dull1*, *Lsm6*, and *Snx19*) (Revised Extended data Fig. 10h). Taken together, the knockdown experiments confirmed the essential role KDM5B for regulation of CM maturation and proliferation.

5. Indeed, continuing point 4, throughout the mechanistic sections of the manuscript, the authors treat isolated cardiomyocytes with various chemicals (e.g. α KG and enzyme inhibitors) and OE KDM5D. Because of off-target effects of inhibitors and chemicals, it is crucial to additionally target specific enzymes (IDHs and KDMs as well potentially KMTs for K27 and K9) using genetic KD or KO.

Response: To address this concern we overexpressed *Idh3b* and *Idh3g* in neonatal CM via lentiviral transduction, which increases α KG levels and reduces H3K4me3 at promoter and TSS regions of cardiac maturation genes with broad H3K4me3 domains such as *Tnni3*, *Mylk3*, *Cacna1g*, *Actc1* etc. In contrast, no reduction of H3K4me3 is seen on genes (e.g. *Snx19*) with narrow H3K4me3 peaks. These results are consistent with the findings in *Cpt1b*-deficient CMs (Revised Extended data Fig. 9 c-h).

Furthermore, we knocked down *Kdm5b* expression by siRNA, which results in marked elevation of H3K4me3. Knockdown of *Kdm5b* also prevents α KG-induced reduction of H3K4me3 level and α KG-induced proliferation of CM. Moreover, we found that knockdown of *Kdm5b* prevents α KG-induced up-regulation of genes related to CM immaturity (e.g. *Nppb*, *Myh7*) but increases expression of maturity-related genes (e.g. *Tnni3*, and *Mylk3*), which are suppressed by α KG treatment (Revised Fig 5i, Revised Extended data Fig. 10e-h). Thus, we conclude that the increase of H3K9me3 and H3K27me3 observed in *Cpt1b*-deficient CMs are secondary effects, which only happen after prolonged metabolic reprogramming of CMs, such as 8 weeks after *Cpt1b* depletion. We think that a detailed molecular analysis of secondary events causing the increase of H3K9me3 and H3K27me3 is well beyond the scope of this study. Therefore, we did not attempt to manipulate expression of KMTs in CMs, which is a project in its own right.

Additional Points:

1. To confirm the *Cpt1b* deletion is specific to cardiomyocytes, authors should examine other cardiac cell types to show that they maintain WT levels of *Cpt1b*.

Response: We monitored expression of *Cpt1a* and *Cpt1b* in non-CMs isolated from Ctrl and *Cpt1b*-deficient animals by RT-qPCR analysis. *Cpt1a* and *Cpt1b* mRNA levels are not altered in non-CMs of *Cpt1b* KO animals compared to the controls, indicating specific inactivation of *Cpt1b* in CM (Fig. 2a for reviewer).

Fig2 for reviewer: a, RT-qPCR analysis of *Cpt1a* and *Cpt1b* expression in non-CMs isolated from 10 weeks old *Ctrl^{Cre}* and *Cpt1b^{CKO}* mice (n=4). *36b4* expression was used as reference. Two-tailed, unpaired student t-tests were performed for statistical analysis.

2. In Figure 3J, the heat maps look nearly identical between the *Cpt1b*CKO and the *Cpt1b*iKO. Could this be an accidental duplication? Further the heat maps are all at maximum increase or decrease. These results seem highly unlikely. The authors should expand the range of the Z-score.

Response: The heat maps are not erroneously duplicated. Nearly identical changes occur in *Cpt1b^{CKO}* and *Cpt1b^{iKO}* animals. We have expanded the range of the Z-score and now show the heat map with log₂ (expression) as requested by the reviewer (Revised extended data Fig. 3c). We also compared RNA-seq data from *Cpt1b^{CKO}* and *Cpt1b^{iKO}* animals to identify genes that were up- and downregulated in both lines (Venn diagrams in (Revised extended data Fig. 3f)).

3. Authors indicate increased levels in α KG in the *Cpt1b* KO is due to increased levels of IDH proteins. However, it is unclear if this is due to specific loss of *Cpt1b* or dysregulation of the FAO pathway. Authors should KO other enzymes within the FAO pathway to show α KG levels are increased.

Response: The suggestion of a knockdown of enzymes involved in FAO is interesting. However, most of enzymes involved in fatty acid oxidation also play essential roles in BCAA catabolism. It would be very difficult to block FAO completely while avoiding disturbances of amino acid metabolism. We demonstrate increased activity of BCAA pathway, whose metabolites inhibit OGDH activity. Thus, manipulation of enzymes of FAO seems contra-productive. CPT1 is the rate-limiting enzyme for fatty acid beta-oxidation. Inactivation of *Cpt1b* prevents transport of fatty acids into mitochondria for further oxidation, resulting in the complete blockage of the FAO pathway without changing expression of enzymes that are involved in BCAA metabolism.

Our data indicate that excessive α KG accumulation after inhibition of FAO ramps up cardio-protection and enables heart regeneration. To further validate this hypothesis, we generated a conditional *Ogdh* knockout mouse line. Inactivation of *Ogdh* in CMs results in substantial accumulation of α KG as expected. Importantly, conditional inactivation of *Ogdh* in CMs recapitulates the phenotype observed in *Cpt1b* KO animals including reduced H3K4me3 levels, resulting in reversed maturation of CMs and elevated CMs proliferation (monitored by Ki67, pH3, Aurora B staining, and cell number analysis) (Revised Fig. 3j-o, Revised Extended data Fig. 7f-m, Revised Extended data Fig. 8b). Recapitulation of the *Cpt1b* phenotype in a different mouse model that also shows a major increase in α KG concentration in CMs strongly supports our conclusions about the underlying mechanisms and the critical role of α KG in this process.

4. Following point 3, authors should overexpressed IDH genes to prove the hypothesis that is indeed increased levels of these proteins that promotes high levels of α KG.

Response: We followed the reviewer's advice and overexpressed *Idh3b* and *Idh3g* in CMs, which increases α KG levels, activates KDM5, and reduces H3K4me3 deposition on cardiac maturation-related genes, resulting in increased CM proliferation. Importantly, these effects are reversed by administration of the KDM5 inhibitor, which confirms that the pro-proliferative effects of α KG accumulation are KDM5-dependent (Revised Extended data Fig. 9c-f). Please see also our responses to main comment #1.

5. Authors mention there are no changes in most of the TCA metabolites measured. However, about half of them seem to be different between WT and *Cpt1b* KO, can the authors clarify how KO cells are producing energy?

Response: We observed an increase of pyruvate but a decline of lactate levels in *Cpt1b*-deficient CMs. Pyruvate is metabolized to oxaloacetate and Acetyl-CoA in mitochondria. We assume that enhanced utilization of glucose leading to increased pyruvate production partially compensates for suppression of FAO, allowing production of TCA cycle metabolites in *Cpt1b*-deficient CMs (Revised Fig. 3c). In addition, increased breakdown of BCAAs may contribute to the production of numerous metabolites of the TCA cycle, including acetyl-CoA and succinyl-CoA (Revised Fig. 3d, e). These conclusions are not only supported by the metabolic analysis presented in the initial version of manuscript but also by two new lines of evidence: 1) new seahorse experiments demonstrate that energy production in *Cpt1b*-deficient adult CMs is restored to a comparable or even higher level than in WT CMs when *Cpt1b*-deficient adult CMs are fed with pyruvate, BCAA, and glucose in the presence of low levels of palmitic acid (Revised Extended data Fig. 6i, j); 2) new metabolic flux experiments reveal that generation of ^{13}C -labeled intermediate CoAs from ^{13}C -palmitate is strongly reduced in *Cpt1b*-deficient hearts. In contrast, generation of ^{13}C -labeled acetyl-CoA and propionyl-CoA is increased, which argues for compensatory usage of alternative energy substrates (Revised Fig. 3f, g). We conclude that *Cpt1b*-deficient CMs mainly utilize pyruvate, glucose, and amino acids to produce energy needed for cardiac contraction. Our findings are consistent with published studies, demonstrating that amino acid catabolism compensates for energy production in ischemic hearts³⁴.

6. Authors should show Figures 6B and 6C for the *Cpt1b* KO. This would support the effects that *Cpt1b* KO are targeted to H3K4me3 broad domains.

Response: To comply with the reviewer's request, we generated ChIP-seq coverage profiles of H3K4me3 for *Cpt1b* KO CMs and corresponding expression boxplots of different gene groups following the same categorization (width of H3K4me3 peaks) as in (Revised Fig.4c). No obvious differences were apparent in *Cpt1b* KO CMs compared to control CMs (Fig. 3a, b for reviewer). We assume that the difference of H3K4me3 enrichment in mutant CMs is masked, since only 945 (478 genes with down-regulated peaks; 467 genes with up-regulated peaks) out of more than ten thousand of genes contain differential H3K4me3 peaks. To highlight the differences of H3K4me3 deposition in genes that respond to α KG accumulation in *Cpt1b*-deficient CMs, we compared H3K4me3 coverage profiles of 151 genes, for which mRNA levels and H3K4me3 enrichment are reduced after *Cpt1b* inactivation (Revised Fig.4d), in control and *Cpt1b* KO CMs. We observed a pronounced attenuation of H3K4me3 deposition within gene bodies compared to TSS regions, indicating the broad H3K4me3 domains, which extend into gene bodies, are prone to be erased by activated KDM5 after *Cpt1b* inactivation (Revised Extended data Fig. 8h).

Figure 3 for reviewer: **a**, Coverage plots of H3K4me3 ChIP-seq signals in *Cpt1b*^{cKO} CMs of genes categorized into three different groups based on H3K4me3 peak breadth: The broad group contains the top 25% genes with broadest peaks. The narrow group contains the 25% genes with narrowest peaks. The medium group contains all other genes (25-75%). **b**, Box plots showing DESeq-normalized expression levels of genes with broad, medium and narrow H3K4me3 peaks in *Cpt1b*^{cKO} CMs. Error bars represent mean \pm s.e.m. One-way ANOVA analysis with correction of multiple testing by controlling the FDR with the two-stage step-up method of Benjamini, Krieger and Yekutieli.

7. Authors should replace tracks for *Cacna1g* with *TNI3* as functional data (Fig 7N, and S7N) focuses on *TNI3*.

Response: We appreciate this comment. However, the ChIP-seq analysis of H3K4me3 on *Tnni3* only uncovers a tendency for reduction of H3K4me3 in *Cpt1b* KO CMs but does not show statistically significant differences due to an outlier (Fig. 4 for reviewer). To cope with this shortcoming, we performed additional H3K4me3 ChIP-qPCR experiments using higher n-numbers, which reveal a clear reduction of H3K4me3 deposition over *Tnni3* (Revised Extended data Fig. 8i).

Fig4 for reviewer: a, Deseq2 normalized counts of identified peaks for *Tnni3* using a H3K4me3 ChIP-seq dataset comparing *Ctrl^{Cre}* and *Cpt1b^{ckO}* samples.

8. Authors nicely demonstrate that the effects of α KG is blunted upon KDM inhibitor, however, it would be a good control to show this effect is not observed at genes that are changing but are independent of KDM5 (H3K4me narrow peak marked genes).

Response: Thank you for this suggestion. We have performed additional RNA-seq experiments using RNA from CMs treated with either DMSO or α KG. We did not observe significant differences in the expression levels of genes with narrow H3K4me3 peak such as *Dus11*, *Snx19* and *Lsm6* (Fig. 5a for reviewer) after α KG treatment compared to mock-treated CMs, indicating the broad H3K4me3 marked genes are more sensitive to α KG accumulation.

Fig5 for reviewer: a, Deseq2 normalized counts indicating gene expression for genes with narrow peaks in DMSO and α KG treated neonatal CMs.

9. *Authors should provide western blot images that better match the quantifications shown (Fig 6A, 7D and 7J).*

Response: The reviewer is right. We have replaced the respective images in (Revised Fig. 4a, Fig. 5d, e) with more representative western blots.

References

- 1 Chabowski, A., Gorski, J., Glatz, J. F., JJ, P. L. & Bonen, A. Protein-mediated Fatty Acid Uptake in the Heart. *Curr Cardiol Rev* **4**, 12-21, doi:10.2174/157340308783565429 (2008).
- 2 Angelini, A. *et al.* PHDs/CPT1B/VDAC1 axis regulates long-chain fatty acid oxidation in cardiomyocytes. *Cell Rep* **37**, 109767, doi:10.1016/j.celrep.2021.109767 (2021).
- 3 Rech, M. *et al.* Assessing fatty acid oxidation flux in rodent cardiomyocyte models. *Sci Rep* **8**, 1505, doi:10.1038/s41598-018-19478-9 (2018).
- 4 Bartelds, B. *et al.* Myocardial carnitine palmitoyltransferase I expression and long-chain fatty acid oxidation in fetal and newborn lambs. *Am J Physiol Heart Circ Physiol* **286**, H2243-2248, doi:10.1152/ajpheart.00864.2003 (2004).
- 5 Ascuitto, R. J., Ross-Ascuitto, N. T., Chen, V. & Downing, S. E. Ventricular function and fatty acid metabolism in neonatal piglet heart. *Am J Physiol* **256**, H9-15, doi:10.1152/ajpheart.1989.256.1.H9 (1989).
- 6 Makinde, A. O., Gamble, J. & Lopaschuk, G. D. Upregulation of 5'-AMP-activated protein kinase is responsible for the increase in myocardial fatty acid oxidation rates following birth in the newborn rabbit. *Circ Res* **80**, 482-489, doi:10.1161/01.res.80.4.482 (1997).
- 7 Zahabi, A. & Deschepper, C. F. Long-chain fatty acids modify hypertrophic responses of cultured primary neonatal cardiomyocytes. *J Lipid Res* **42**, 1325-1330 (2001).
- 8 Lopaschuk, G. D., Belke, D. D., Gamble, J., Itoi, T. & Schonekess, B. O. Regulation of fatty acid oxidation in the mammalian heart in health and disease. *Biochim Biophys Acta* **1213**, 263-276, doi:10.1016/0005-2760(94)00082-4 (1994).
- 9 van der Vusse, G. J., Glatz, J. F., Stam, H. C. & Reneman, R. S. Fatty acid homeostasis in the normoxic and ischemic heart. *Physiol Rev* **72**, 881-940, doi:10.1152/physrev.1992.72.4.881 (1992).
- 10 Schonfeld, P. & Wojtczak, L. Short- and medium-chain fatty acids in energy metabolism: the cellular perspective. *J Lipid Res* **57**, 943-954, doi:10.1194/jlr.R067629 (2016).
- 11 Silva, Y. P., Bernardi, A. & Frozza, R. L. The Role of Short-Chain Fatty Acids From Gut Microbiota in Gut-Brain Communication. *Front Endocrinol (Lausanne)* **11**, 25, doi:10.3389/fendo.2020.00025 (2020).
- 12 Demigne, C., Yacoub, C. & Remesy, C. Effects of absorption of large amounts of volatile fatty acids on rat liver metabolism. *J Nutr* **116**, 77-86, doi:10.1093/jn/116.1.77 (1986).
- 13 Labarthe, F., Gelinas, R. & Des Rosiers, C. Medium-chain fatty acids as metabolic therapy in cardiac disease. *Cardiovasc Drugs Ther* **22**, 97-106, doi:10.1007/s10557-008-6084-0 (2008).
- 14 Papamandjaris, A. A., MacDougall, D. E. & Jones, P. J. Medium chain fatty acid metabolism and energy expenditure: obesity treatment implications. *Life Sci* **62**, 1203-1215, doi:10.1016/s0024-3205(97)01143-0 (1998).
- 15 Bach, A. C. & Babayan, V. K. Medium-chain triglycerides: an update. *Am J Clin Nutr* **36**, 950-962, doi:10.1093/ajcn/36.5.950 (1982).
- 16 Nagaoka, K. *et al.* A New Therapeutic Modality for Acute Myocardial Infarction: Nanoparticle-Mediated Delivery of Pitavastatin Induces Cardioprotection from Ischemia-Reperfusion Injury via Activation of PI3K/Akt Pathway and Anti-Inflammation in a Rat Model. *PLoS One* **10**, e0132451, doi:10.1371/journal.pone.0132451 (2015).
- 17 Dongworth, R. K. *et al.* Quantifying the area-at-risk of myocardial infarction in-vivo using arterial spin labeling cardiac magnetic resonance. *Sci Rep* **7**, 2271, doi:10.1038/s41598-017-02544-z (2017).
- 18 Evans, S. *et al.* Ischemia reperfusion injury provokes adverse left ventricular remodeling in dysferlin-deficient hearts through a pathway that involves TIRAP dependent signaling. *Sci Rep* **10**, 14129, doi:10.1038/s41598-020-71079-7 (2020).

- 19 Robison, P. *et al.* Detyrosinated microtubules buckle and bear load in contracting cardiomyocytes. *Science* **352**, aaf0659, doi:10.1126/science.aaf0659 (2016).
- 20 Schuldt, M. *et al.* Proteomic and Functional Studies Reveal Detyrosinated Tubulin as Treatment Target in Sarcomere Mutation-Induced Hypertrophic Cardiomyopathy. *Circ Heart Fail* **14**, e007022, doi:10.1161/CIRCHEARTFAILURE.120.007022 (2021).
- 21 Patel, M. S. Inhibition by the branched-chain 2-oxo acids of the 2-oxoglutarate dehydrogenase complex in developing rat and human brain. *Biochem J* **144**, 91-97, doi:10.1042/bj1440091 (1974).
- 22 Gibson, G. E. & Blass, J. P. Inhibition of acetylcholine synthesis and of carbohydrate utilization by maple-syrup-urine disease metabolites. *J Neurochem* **26**, 1073-1078, doi:10.1111/j.1471-4159.1976.tb06988.x (1976).
- 23 Oldham, W. M., Clish, C. B., Yang, Y. & Loscalzo, J. Hypoxia-Mediated Increases in L-2-hydroxyglutarate Coordinate the Metabolic Response to Reductive Stress. *Cell Metab* **22**, 291-303, doi:10.1016/j.cmet.2015.06.021 (2015).
- 24 Andrade, J. *et al.* Control of endothelial quiescence by FOXO-regulated metabolites. *Nat Cell Biol* **23**, 413-423, doi:10.1038/s41556-021-00637-6 (2021).
- 25 Sundheim, O. *et al.* Human ABH3 structure and key residues for oxidative demethylation to reverse DNA/RNA damage. *EMBO J* **25**, 3389-3397, doi:10.1038/sj.emboj.7601219 (2006).
- 26 Mailloux, R. J. *et al.* The tricarboxylic acid cycle, an ancient metabolic network with a novel twist. *PLoS One* **2**, e690, doi:10.1371/journal.pone.0000690 (2007).
- 27 Bae, J. *et al.* Malonate Promotes Adult Cardiomyocyte Proliferation and Heart Regeneration. *Circulation* **143**, 1973-1986, doi:10.1161/CIRCULATIONAHA.120.049952 (2021).
- 28 Olenchok, B. A. *et al.* EGLN1 Inhibition and Rerouting of alpha-Ketoglutarate Suffice for Remote Ischemic Protection. *Cell* **164**, 884-895, doi:10.1016/j.cell.2016.02.006 (2016).
- 29 Holubarsch, C. J. *et al.* A double-blind randomized multicentre clinical trial to evaluate the efficacy and safety of two doses of etomoxir in comparison with placebo in patients with moderate congestive heart failure: the ERGO (etomoxir for the recovery of glucose oxidation) study. *Clin Sci (Lond)* **113**, 205-212, doi:10.1042/CS20060307 (2007).
- 30 Mahmoud, A. I. *et al.* Meis1 regulates postnatal cardiomyocyte cell cycle arrest. *Nature* **497**, 249-253, doi:10.1038/nature12054 (2013).
- 31 Nakada, Y. *et al.* Hypoxia induces heart regeneration in adult mice. *Nature*, doi:10.1038/nature20173 (2016).
- 32 Cardoso, A. C. *et al.* Mitochondrial substrate utilization regulates cardiomyocyte cell-cycle progression. *Nature Metabolism* **2**, 167-178, doi:10.1038/s42255-020-0169-x (2020).
- 33 Chen, Y. *et al.* Reversible reprogramming of cardiomyocytes to a fetal state drives heart regeneration in mice. *Science* **373**, 1537-1540, doi:10.1126/science.abg5159 (2021).
- 34 Drake, K. J., Sidorov, V. Y., McGuinness, O. P., Wasserman, D. H. & Wikswo, J. P. Amino acids as metabolic substrates during cardiac ischemia. *Exp Biol Med (Maywood)* **237**, 1369-1378, doi:10.1258/ebm.2012.012025 (2012).

Reviewer Reports on the First Revision:

Referees' comments:

Referee #1 (Remarks to the Author):

The authors have provided more data in this revision. Overall, the results support the conclusion that increasing alpha-KG enhances KDM5 activity and stimulates cardiomyocytes proliferation. A few points need clarification though.

It is still fuzzy how CPT1b KO increases alpha-KG level. The schematics in Fig S10i seemed to suggest that increased citrate export from mitochondria was responsible for a greater production of alpha-KG in cytosol/nucleus. What are the evidence for increased citrate export? It was not made clear in the text. Is this also the case for Ogdh KO?

Increased CM proliferation in Ogdh KO heart supports the alpha-KG hypothesis. However, Ogdh KO mice died early thus presented a worse phenotype than CPT1b KO. This should be clearly stated in the text (not just in the response to reviewers) to distinguish the two models.

The authors proposed that H3K4me3 demethylation reduced expression of mature cardiac contractile proteins and, through unclear mechanism, triggered proliferation of CM. Curiously, reduced maturation of sarcomeres were not associated with any changes in the expression of metabolism genes, and contractile function of the heart was maintained. A discussion of how such a scenario is achieved should be included.

Referee #2 (Remarks to the Author):

This is a very well written and executed manuscript. The authors have done a good job in addressing the reviewers' comments. The main remaining issue in my view is the injury model as explained below. It is not possible to assess the pro-regenerative effect with the data provided.

Major:

1) The main issue in the revised manuscript is that the cardiac function in the injury model is not performed correctly. Ejection fraction data are lacking clarity. Figure 2g shows evidence of cardioprotection 2 days after injury. However, ejection fraction after the most relevant model, which is deletion a few days after injury, is not done. The model where the authors induce deletion one day after injury does not completely preclude cardioprotection. In my view it would be important to space out the induction of deletion a few days after injury. This is in my view the most relevant study to evaluate the potential for regeneration. The other models test protection rather than regeneration. The authors need to perform temporal assessment of systolic function before and shortly after injury and then at later timepoints. It is important to determine that both groups started out with a similar degree of injury. It has become standard in the field now to perform an echocardiogram a few days after injury before initiation of the deletion to make sure that the experimental group does not start at a higher ejection fraction

2) It is interesting that the authors did not observe lipotoxicity as seen in previous studies (Circulation. 2012 Oct 2; 126(14): 1705–1716.) and I am satisfied with the author's explanation. However, the issue of potential hypertrophic effect remains especially that the authors note an increase in cardiomyocyte size. This is true especially for the drug studies. The authors state that the global ko had lipotoxicity compared to the conditional ko. Does the drug lead to lipotoxicity? Given that it would affect skeletal muscles and the heart?

3) The number of mitotic events is quite impressive. 15 aurora B cardiomyocyte per section and over 20 pH3(ser10) cardiomyocyte per section is highly unusual based on published reports. It would be important for the authors to show histological sections demonstrating this impressive phenotype. The images that are provided are not very impressive. For example, they show in figure 1K that there are over 40 myocytes that are ki67+. However the limited images provided only show one positive myocyte in a relatively modest magnification field. Please provide additional images.

Minor:

1) There is discrepancies in cardiomyocyte numbers. In figure 1n myocyte number in control mice is less than 700k, which is unusually low. While in panel o the number in control mice is close to 1 million. Please elaborate

2) The authors need to show full panels of histological sections (base to apex) in the injury models so that the reader can appreciate the extent of difference between control and KO groups.

3) It is quite interesting that there marked mitotic indices 4 weeks after tamoxifen injection. Did the authors follow these mice for longer periods? Do they eventually die of cardiomegaly?

Referee #4 (Remarks to the Author):

The manuscript of Li et al. describes studies to demonstrate that the inhibition of fatty acid oxidation stimulates KDM5-mediated H3K4me3 demethylation and 2 enables heart regeneration in adult mice. The revised version of the manuscript contains substantial metabolomic data that supports the findings of the study and I have been asked to comment specifically on the quality and suitability of those data and their interpretation.

I found the metabolomics data to be of high quality. It is well described using appropriate, targeted methodology with stable isotope standards or calibration curves for quantitation.

The interpretation of these data is also consistent with the proposed mechanism of the switch from FAO to other forms of energy metabolism including BCAA and glucose in the Cpt1b deficient CMs. I found the responses to the Reviewers comments with this new metabolomic data supported the underlying hypotheses of the study concerning the downstream metabolic consequences of Cpt1b inhibition in cardiomyocytes.

Author Rebuttals to First Revision:

Reviewers' comments:

Referee #1 (Remarks to the Author):

The authors have provided more data in this revision. Overall, the results support the conclusion that increasing alpha-KG enhances KDM5 activity and stimulates cardiomyocytes proliferation. A few points need clarification though.

Response: We sincerely appreciate the positive comments of the reviewer.

It is still fuzzy how CPT1b KO increases alpha-KG level. The schematics in Fig S10i seemed to suggest that increased citrate export from mitochondria was responsible for a greater production of alpha-KG in cytosol/nucleus. What are the evidence for increased citrate export? It was not made clear in the text. Is this also the case for Ogdh KO?

Response: We apologize for the misleading scheme in Fig S10i, which depicts the export of citrate from mitochondria and import into nuclei. The scheme was only meant to illustrate one possible route how α KG may reach the nucleus to activate KDM5. α KG is able to freely diffuse through channels (such as voltage-dependent anion channels) in the outer mitochondrial membrane, and it is transported across the inner mitochondrial membrane through the oxoglutarate carrier, also known as an oxoglutarate/malate antiporter. α KG may also exit the mitochondria after conversion to citrate via the mitochondrial citrate transporter (SLC25A1). Excessive levels of α KG in Cpt1b-mutant CMs may be generated in mitochondria by IDH2/3 followed by transport (directly or via citrate) into the cytoplasm, or in the cytosol by IDH1. Uptake of α KG into nuclei may happen by passive diffusion or by active transport via unknown transporters. Moreover, α KG might also be produced directly within nuclei, since various studies report the nuclear localization of several mitochondrial enzymes including ACO2 and IDH3 converting citrate to α KG (Nagaraj et al., Cell, 2017; Kafkia et al., Sci. Adv. 2022).

Our analysis of FACS-purified CM nuclei revealed that the α KG-generating enzyme IDH3G is present in the nuclei of CMs, supporting the possibility that nuclear IDH3 generates α KG from citrate in nuclei of cardiomyocytes. However, we did not directly examine the different routes, by which α KG reaches the nucleus, neither in *Cpt1b*- nor in *Ogdh*-mutant CMs, since this has not been the focus of our study. Moreover, it is still a major problem in the field to accurately measure metabolites in distinct sub-cellular compartments, which would be essential to determine exactly where α KG is produced. To avoid any confusion, we have modified the scheme in Fig S10i. We have added an arrow that represents the direct mitochondrial export of α KG into cytosol, next to the route via citrate. To emphasize that the exact mechanisms by which α KG and citrate are imported into the nucleus are not entirely clear, the corresponding arrows are dashed, and a remark was added to the legend that “The mechanisms underlying the transport of α KG and citrate (dashed lines) into the nucleus are not fully understood”.

We explain the increased accumulation of α KG in *Cpt1b*-mutant CMs by the enhanced expression of all IDH enzymes, which convert of citrate/isocitrate into α KG, and the reduced OGDH activity, converting α KG to succinyl-CoA. The reduction of the enzymatic activity of OGDH is most likely caused by the well-known inhibitory effects of intermediate metabolites from BCAA catabolism, which are increased in the *Cpt1b*-mutants.

Increased CM proliferation in Ogdh KO heart supports the alpha-KG hypothesis. However, Ogdh KO mice died early thus presented a worse phenotype than CPT1b KO. This should be clearly stated in the text (not just in the response to reviewers) to distinguish the two models.

Response: Of course, the reviewer is right. We apologize for this omission. We now describe in more detail the phenotype of *Ogdh*-mutants in the revised text.

The authors proposed that H3K4me3 demethylation reduced expression of mature cardiac contractile proteins and, through unclear mechanism, triggered proliferation of CM. Curiously, reduced maturation of sarcomeres were not associated with any changes in the expression of metabolism genes, and contractile function of the heart was maintained. A discussion of how such a scenario is achieved should be included.

Response: We thank the reviewer for this comment. However, it is not completely correct that we did not see changes in the expression of metabolism genes. As shown in (Extended data Figure 3), we observed multiple changes in GO-terms for “fatty acid metabolic process”, “metabolic process”, “lipid metabolic process”, “fatty acid beta oxidation”, “regulation of cholesterol metabolic process” and others. Several of the genes that contribute to the enrichment of GO-terms are also involved in other metabolic processes, which makes it very hard to pinpoint their specific role in the reprogramming CMs to a more immature proliferative state. Importantly, we discovered substantial changes in the concentrations or activities of different enzymes that play decisive roles in metabolic regulation. For example, the

concentration of IDH1, IDH2, IDH3A and OGDH were strongly increased, but the enzymatic activity of OGDH was much lower. Apparently, metabolic reprogramming after *Cpt1b* inactivation is mainly achieved at the posttranscriptional or even posttranslational level.

Based on morphological criteria obtained by EM analysis adult *Cpt1b*-deficient CMs resemble a more fetal, neonatal-like stage with more loosely packed sarcomeres. Neonatal hearts are fully functional and contractile but still allow division of cardiomyocytes to occur. We think that *Cpt1b*-deficient hearts have acquired a similar state, allowing CM divisions while maintaining regular contractile functions. We also cannot exclude that, following metabolic reprogramming, only a distinct subset of CMs enters cytokinesis, which show a more severe disassembly of sarcomeres but cannot be easily identified on tissues sections. We are intensively working on this possibility and are confident to come up with new insights in the future. It may be also feasible that CMs only temporarily undergo more extensive disassembly of sarcomeres, which would also explain why we did not observe a more widespread sarcomere disassembly within the myocardium. We now provide additional quantitative data to describe the reduction in sarcomere density in adult *Cpt1b*-deficient cardiomyocytes, which is a characteristic for CMs at the neonatal stage (New Extended data Fig. 3e). We also discuss the different scenarios in the revised manuscript as requested.

Referee #2 (Remarks to the Author):

This is a very well written and executed manuscript. The authors have done a good job in addressing the reviewers' comments. The main remaining issue in my view is the injury model as explained below. It is not possible to assess the pro-regenerative effect with the data provided.

Response: We appreciate the reviewer's positive view of the revised manuscript. We agree that it is important to distinguish between pro-regenerative and cardioprotective effects after inactivation of *Cpt1b*. To specifically address this concern, we now provide new data, demonstrating that depletion of CPT1B protein, initiated by treatment with TAM one day after ischemia/reperfusion injury, occurs 5-10 days later, well beyond the time period relevant for cardioprotection. With other words, CPT1B is still present during the time when cardioprotection may occur and thus cannot contribute to this process. Moreover, we performed additional cMRI experiments to demonstrate that heart functions of drop shortly after ischemia/reperfusion injury, followed by an improvement in *Cpt1b*-deficient mice but not in control mice.

Major:

1) The main issue in the revised manuscript is that the cardiac function in the injury model is not performed correctly. Ejection fraction data are lacking clarity. Figure 2g shows evidence of cardioprotection 2 days after injury. However, ejection fraction after the most relevant model, which is deletion a few days after injury, is not done. The model where the authors induce deletion one day after injury does not completely preclude cardioprotection. In my view it would be important to space out the induction of deletion a few days after injury. This is in my view the most relevant study to evaluate the potential for regeneration. The other models test protection rather than regeneration. The authors need to perform temporal assessment of systolic function before and shortly after injury and then at later timepoints. It is important to determine that both groups started out with a similar degree of injury. It has become standard in the field now to perform an echocardiogram a few days after injury before initiation of the deletion to make sure that the experimental group does not start at a higher ejection fraction

Response: We understand the reviewer's concern about the injury model used to evaluate the potential for regeneration. We also agree with the reviewer that (new Extended data Fig. 5e-g) shows evidence of cardioprotection two days after injury. In fact, we argued that the improved heart function in *Cpt1b*-deficient mice (deletion of *Cpt1b* before injury) versus control mice after ischemia/reperfusion injury is the result of both cardiac regeneration and cardioprotection. To unequivocally distinguish between cardiac protection and cardiac regeneration, we focused on the model in which inactivation of *Cpt1b* is initiated one day after I/R surgery (post-I/R inactivation of *Cpt1b*), as recommended by the reviewer and therefore moved the old (Figure 2g) to the extended figures, putting more weight on the new measurements. To prove that the strong reduction of the fibrotic scar area in *Cpt1b*^{IKO} compared to Ctrl animals is not caused by cardioprotection but by cardiac regeneration, we analyzed the concentration of CPT1B protein at different time points after initiation of TAM treatment. We found that CPT1B protein levels only showed a minor decline five days after initiation of TAM treatment (i.e. six days after I/R injury) but are markedly reduced after ten days (new Extended data Fig. 5h). Thus, CPT1B is still present during the time period when cardiac protection may happen and therefore cannot contribute to cardioprotection, indicating that absence of scar formation in the post-I/R *Cpt1b*-inactivation cohort is caused by cardiac regeneration.

Furthermore, we complied with the reviewer's request and conducted additional longitudinal MRI measurements using the post-I/R *Cpt1b*-inactivation cohort. As expected, we observed a similar decline of the ejection fraction one week after I/R in control as well as in *Cpt1b*-flox mutant mice, in which CPT1B protein levels are not yet markedly reduced, verifying that cardioprotection has no impact in the

post-I/R *Cpt1b* inactivation cohort. One week later (two weeks after I/R injury), we found a further decline in the post-I/R control cohort, but not in the mutant group that by now had lost CPT1B. Importantly, after additional two weeks (four weeks after I/R injury), *Cpt1b* mutants show a remarkable recovery of cardiac function, nearly reaching 90% of the ejection fraction seen before the I/R injury (new Fig.2i). We think, these new results unequivocally demonstrate that suppression of fatty acid oxidation and the subsequent metabolic and epigenetic changes enable heart regeneration. We have changed the text in revised manuscript accordingly.

2) *It is interesting that the authors did not observe lipotoxicity as seen in previous studies (Circulation. 2012 Oct 2; 126(14): 1705–1716.). and I am satisfied with the author’s explanation. However, the issue of potential hypertrophic effect remains especially that the authors note an increase in cardiomyocyte size. This is true especially for the drug studies. The authors state that the global ko had lipotoxicity compared to the conditional ko. Does the drug lead to lipotoxicity? Given that it would affect skeletal muscles and the heart?*

Response: As explained in the response letter for the first revision, “we treated C57/BL6 WT mice for 2 weeks with etomoxir. Surprisingly, the etomoxir treatment partially recapitulated the phenotype of *Cpt1b* KO mice including elevated HW/BW ratios, elevated numbers of Ki67, pH3, and Aurora B positive CMs. Unfortunately, etomoxir-treated animals showed a strong physical burden due to the known hepatotoxicity, which prevented us from performing heart regeneration experiments. We believe that development of a selective inhibitor of CPT1B that ideally targets only CMs holds great promise for future therapeutic approaches. Due to space restrictions and the controversial discussions about the use of etomoxir in *in vivo* experiments, we do not show these data in the manuscript”. Nevertheless, to satisfy the reviewer’s curiosity, we stained heart sections of mice treated with etomoxir for formation of lipid droplets. Indeed, we observed formation of lipid droplets in cardiomyocytes of some mice treated with we etomoxir but never in control-treated animals (Fig. 1 for reviewers). We sincerely think that future work should focus on compounds that specifically inhibits CPT1B in cardiomyocytes.

Figure 1 for reviewers: a, Lipid drop detection with LipidTox Green, counterstained with DAPI on heart tissue section from Ctrl and Etomoxir treated mice. Scale bar: 50µm.

3) *The number of mitotic events is quite impressive. 15 aurora B cardiomyocyte per section and over 20 pH3(ser10) cardiomyocyte per section is highly unusual based on published reports. It would be important for the authors to show histological sections demonstrating this impressive phenotype. The images that are provided are not very impressive. For example, they show in figure 1K that there are over 40 myocytes that are ki67+. However the limited images provided only show one positive myocyte in a relatively modest magnification field. Please provide additional images.*

Response: We were also impressed by the relatively high number of mitotic events. However, the high number of mitotic events fits very well to the nearly doubled number of CMs in *Cpt1b*^{ckO} heart and the continuous increase of CM numbers in *Cpt1b*^{ikO} mutant hearts between 4 and 8 weeks after TAM

injection. We do not think that it makes sense to show a low magnification image of a single section, since the low magnification prevents detection of the signals. Therefore, we now provide more images from a SINGLE section of *Cpt1b^{iKO}* mutant hearts, in which Ki67, Aurora B⁺ or pH3(ser10)⁺ cardiomyocytes are indicated and enlarged (Fig. 2 for reviewers). Due to space restrictions, we only provide the figure for the reviewers. We have also replaced individual stainings in (Extended data Fig. 2p) by more representative examples and indicated an additional pH3⁺ CM by inserting an arrow in (Extended data Fig. 2b, two pH3⁺ CMs instead of one in this image).

Figure 2 for reviewers: a-c, Immunofluorescence staining for Ki67 (a), pH3 (b), Aurora B (c), and sarc-actinin on heart sections from *Cpt1b^{iKO}* mice. Scale bar: 50 μ m. All the 10 images were taken from the same heart section.

Minor:

1) *There is discrepancies in cardiomyocyte numbers. In figure 1n myocyte number in control mice is less than 700k, which is unusually low. While in panel o the number in control mice is close to 1 million. Please elaborate.*

Response: We thank the reviewer for careful inspection of our data. (Fig.1o) was meant to identify a potential relative increase of cardiomyocyte numbers between two different time points after TAM treatment in *Cpt1b*-mutant hearts. In fact, the CM numbers in *Cpt1b*-mutant hearts 4 weeks after TAM treatment (Fig.1n and o) are similar, i.e. around 0.9×10^6 . To determine the numbers of CMs per individual heart, all CMs isolated by enzymatic digestion from a single heart, were pooled and plated into a Sedgewick rafter chamber for counting. Since exactly the same protocol was used for all control and mutant hearts, we assume that our experimental approach reliably reflects relative changes in the abundance of CMs. We have most certainly lost some CMs during the isolation procedure but this systematic error equally affects control and mutant hearts and thus should not affect assessment of relative changes.

2) *The authors need to show full panels of histological sections (base to apex) in the injury models so that the reader can appreciate the extent of difference between control and KO groups.*

Response: We thank the reviewer for this suggestion. Due to the limited space in the main text, we only showed corresponding sections of control and mutant hearts from between the apex and the ligation site. Now, we provide a broader range, covering the heart from the base to the apex as requested. Images of layer 2 (from ligation to apex) are shown in the (new Fig.2 a, c, g). Full panels are provided in the (Supplementary Information, Fig. 6).

3) *It is quite interesting that there marked mitotic indices 4 weeks after tamoxifen injection. Did the authors follow these mice for longer periods? Do they eventually die of cardiomegaly?*

Response: We followed the *Cpt1b^{iKO}* mice for up to 8 weeks after tamoxifen injection and observed continuous increase of heart weight/body weight ratios and an increase of cardiomyocyte numbers between 4 and 8 weeks after TAM injection (Fig.1j, o). We did not systematically study *Cpt1b^{iKO}* animals after 8 weeks, but some *Cpt1b^{iKO}* mice survived for up to 50 weeks after TAM treatment (60 weeks-old), suggesting that inactivation of *Cpt1b* in cardiomyocytes is tolerated for a rather long time period without necessarily leading to lethal cardiomegaly. It will be interesting to systematically investigate the fate of aged *Cpt1b^{iKO}* animals in future studies.

Referee #4 (Remarks to the Author):

The manuscript of Li et al. describes studies to demonstrate that the inhibition of fatty acid oxidation stimulates KDM5-mediated H3K4me3 demethylation and 2 enables heart regeneration in adult mice. The revised version of the manuscript contains substantial metabolomic data that supports the findings of the study and I have been asked to comment specifically on the quality and suitability of those data and their interpretation.

I found the metabolomics data to be of high quality. It is well described using appropriate, targeted methodology with stable isotope standards or calibration curves for quantitation.

The interpretation of these data is also consistent with the proposed mechanism of the switch from FAO to other forms of energy metabolism including BCAA and glucose in the Cpt1b deficient CMs. I found the responses to the Reviewers comments with this new metabolomic data supported the underlying hypotheses of the study concerning the downstream metabolic consequences of Cpt1b inhibition in cardiomyocytes.

Response: We are delighted by the positive comments of the reviewer, who appreciated our efforts for the careful evaluation of metabolic changes after inactivation of *Cpt1b* in cardiomyocytes. Thank you.

Reviewer Reports on the Second Revision:

Referees' comments:

Referee #1 (Remarks to the Author):

The authors have satisfactorily addressed my questions.

Referee #2 (Remarks to the Author):

I am a bit confused as to why the authors did not perform the suggested experiment which is straight forward.

The argument that induction one day after injury instead of much longer is sufficient is in my view not accurate. The authors base this argument on levels of the protein at 5 day after days (Suppl fig 5), however we dont see what the levels are like after 1 or 2 days and there is a clear drop after 5 days - which is still within the reperfusion period. Cells continue to die after MI for a few days after injury and as you know the post injury inflammatory response is also part of what is called reperfusion injury.

The authors also dont fully address the two apparent phenotypes appropriately - both hyperplasia and hypertrophy. This is an unusual phenotype and it is unclear how much an increase in cell number contributes to the increase in heart size. I think this should be clearly discussed in the discussion.

Author Rebuttals to Second Revision:

Response to reviewers

Reviewer #1 (Remarks to the Author):

The authors have satisfactorily addressed my questions.

Response: We are delighted about the positive reception of the revised paper.

Reviewer #2 (Remarks to the Author):

I am a bit confused as to why the authors did not perform the suggested experiment which is straight forward. The argument that induction one day after injury instead of much longer is sufficient is in my view not accurate. The authors base this argument on levels of the protein at 5 day after days (Suppl fig 5), however we dont see what the levels are like after 1 or 2 days and there is a clear drop after 5 days - which is still within the reperfusion period. Cells continue to die after MI for a few days after injury and as you know the post injury inflammatory response is also part of what is called reperfusion injury.

Response: We do understand the reviewer's confusion, although we thought that we addressed the issue before. We agree with the reviewer that it is necessary to deplete CPT1B well after the ischemia reperfusion injury in order to claim cardiac regeneration and exclude cardioprotective effects. We also agree that a complete loss of CPT1B 1 day after ischemia reperfusion injury would not make the point, since cardiomyocytes continue to die for a few days after I/R. We specifically designed the experiment to achieve depletion of the CPT1B enzyme well after I/R injury, according to the timing of CPT1B disappearance after initiation of *Cpt1b* gene inactivation. Unlike transcription factors, mitochondrial enzymes have relatively long half-lives, in particular when present in the mitochondrial membrane as CPT1B, where turnover is much slower than in the mitochondria matrix. We assessed CPT1B concentrations in the heart at different time points after initiation of *Cpt1b* gene inactivation and found no decrease 3 days after I/R (2 days after initiation of tamoxifen treatment) and 6 days after I/R (5 days after initiation of tamoxifen treatment). Only 11 days after I/R (10 days after initiation of tamoxifen treatment) we observed a significant decline of CPT1B concentrations. We fully complied with the reviewer's request and measured CPT1B concentrations 3 days after I/R (2 days after initiation of tamoxifen treatment) by western blot analysis. Furthermore, we quantified and statistically evaluated the western blot analysis. We hope these additional data convince the reviewer that loss of CPT1B in cardiomyocytes indeed enables heart regeneration.

The authors also dont fully address the two apparent phenotypes appropriately - both hyperplasia and hypertrophy. This is an unusual phenotype and it is unclear how much an increase in cell number contributes to the increase in heart size. I think this should be clearly discussed in the discussion.

Response: We agree that the simultaneous occurrence of hyperplasia and hypertrophy is unusual. However, hypertrophic growth of cardiomyocytes was minor compared to the massive hyperplasia and did not encompass all cardiomyocytes as detailed in the main text. When describing the content of (Fig. 1), we wrote 'In addition, we noted a relatively modest increase of CM surface area in *Cpt1b*^{ckO} mice, suggesting combined hyperplastic and hypertrophic growth of the myocardium. A more detailed morphometric evaluation revealed a marked increase of the CM population with the smallest cell size (<100 μm²), suggesting formation of new CMs in response to *Cpt1b* inactivation. The percentage of medium-sized CMs

(100-300 μm^2) decreased, whereas the number of larger CMs ($>300 \mu\text{m}^2$) increased significantly, reflecting the overall increase of CM surface area (Fig. 1h).’

Due to the space limitations, we cannot discuss all aspects and potential reasons of the combined hyperplasia and hypertrophy in in *Cpt1b^{CKO}* mice in detail. However, as requested, we inserted a sentence in the discussion to cover this topic. The sentence reads: ‘Increased HIF1 signaling may also contribute to the relatively modest increase of CM surface area in *Cpt1b^{CKO}* mice (doi:10.1111/j.1582-4934.2011.01497.x), which was limited to larger CMs ($>300 \mu\text{m}^2$).’ We also added an additional sentence to the first part of the results section (‘Hyperplastic growth of *Cpt1b*-deficient hearts’), which reads ‘The relatively modest increase of CM surface area, in contrast to the massive increase of CM numbers in *Cpt1b^{CKO}* hearts, suggests that the increased heart size is mainly caused by hyperplastic growth’. We are confident that these additions adequately explain the phenotype.

Reviewer Reports on the Third Revision:

Referees' comments:

Referee #2 (Remarks to the Author):

The authors have satisfactorily responded to my concerns